# Learning What to Remember for Non-Markovian Reinforcement Learning

## Abstract

Recent success in developing increasingly general purpose agents based on sequence models has led to increased focus on the problem of deploying computationally limited agents within the vastly more complex real-world. A key challenge experienced in these more realistic domains is highly non-Markovian dependencies with respect to the agent's observations, which are less common in small controlled domains. The predominant approach for dealing with this in the literature is to stack together a window of the most recent observations (*Frame Stacking*), but this window size must grow with the degree of non-Markovian dependencies, which results in prohibitive computational and memory requirements for both action inference and learning. In this paper, we are motivated by the insight that in many environments that are highly non-Markovian with respect to time, the environment only causally depends on a relatively small number of observations over that time-scale. A natural direction would then be to consider meta-algorithms that maintain relatively small adaptive stacks of memories such that it is possible to express highly non-Markovian dependencies with respect to time while considering fewer observations at each step and thus experience substantial savings in both compute and memory requirements. Hence, we propose a meta-algorithm (*Adaptive Stacking*) for achieving exactly that with convergence guarantees and quantify the reduced computation and memory constraints for MLP, LSTM, and Transformer-based agents. Our experiments utilize popular memory tasks, which give us control over the degree of non-Markovian dependencies in the environment. This allows us to demonstrate that an appropriate meta-algorithm can learn the removal of memories not predictive of future rewards and achieve convergence in the stack management policy without excessive removal of important experiences.

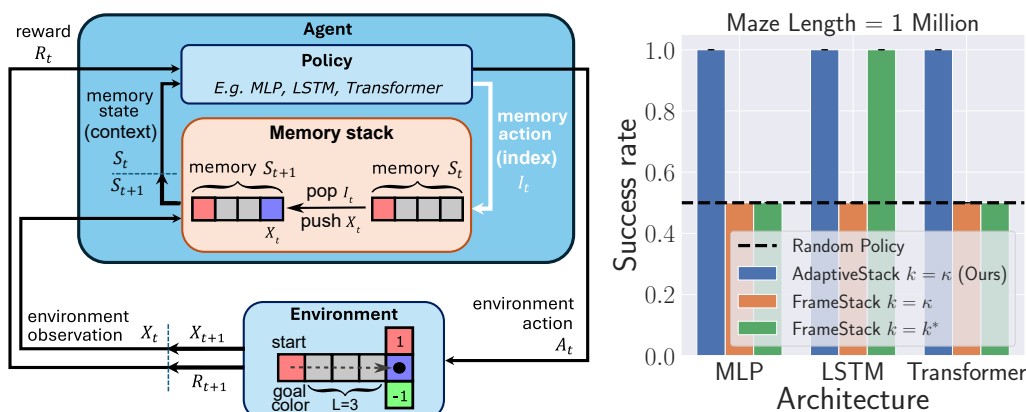

Figure 1: Learning what to remember using Adaptive Stacking. **(left)** Modification to the standard RL loop. **(right)** Performance of PPO on a Passive-TMaze when trained for *one million* timesteps on random maze lengths in $[2, 16]$ and tested on a maze of length *one million*. The episode lengths are equal to the maze lengths ($L + 2$). The stack size (context window) $k$ needed for FrameStack is $k^* = 16$, which not only requires *oracle* knowledge of the environment (the largest maze length), but also scales with the maze length (making it struggle when tested on the larger maze). In contrast, only $\kappa = 2$ is needed irrespective of maze length and architecture.

# 1 INTRODUCTION

Reinforcement learning (RL) agents are typically formulated under the Markov assumption: the agent's current observation contains all information needed for optimal decision-making (Puterman, 2014). In practice, however, real-world environments are often partially observable – the agent's immediate observation is an incomplete snapshot of the true state. This leads to non-Markovian dependencies over time, where past observations contain critical context for future decisions. Notably, Abel et al. (2021) proved that there exist certain tasks (for example expressed as desired behaviour specifications) that cannot be captured by any Markovian reward function. In other words, no memoryless reward can incentivise the correct behaviour for those tasks and agents must rely on histories of observations to infer hidden state information to resolve non-Markovian dependencies. This theoretical insight underlines that non-Markovian tasks are not just harder, but sometimes fundamentally require memory beyond the scope of standard Markov formulations. We are interested in such settings in big worlds (Javed & Sutton, 2024), where only a relatively small subset of past observations are relevant for optimal decision-making, but they are separated by large spans of time.

While RL has shown great success in a variety domains (Arulkumaran et al., 2017; Cao et al., 2024), handling such temporal dependencies remains a challenge especially for computationally limited agents operating in big worlds (Javed & Sutton, 2024). In practice, the most common approach to address this problem is *Frame Stacking* (FS), which is a FIFO short-term memory wherein a fixed context window of the most recent $k^*$ observations (and actions) are concatenated. This is then used directly as policy input, or first used to infer hidden states typically using active inference (Friston, 2009; Sajid et al., 2021) or sequence models like recurrent neural networks (Hochreiter & Schmidhuber, 1997; Hausknecht & Stone, 2015), Transformers (Vaswani et al., 2017; Chen et al., 2021), and state space models (Gu et al., 2021; Samsami et al., 2024). Given knowledge of the nature of the temporal dependencies, for example when they are expressible as reward machines (Icarte et al., 2022; Bester et al., 2023), prior works also use such histories of observations and program synthesis to learn abstract state machines that compactly represent the memory and temporal dependencies (Toro Icarte et al., 2019; Hasanbeig et al., 2024). While such approaches based on FS are very effective in domains with short-term dependencies, such as in Atari games (Mnih et al., 2013) where 4 frames are enough to capture the motion of objects, they quickly become impractical in domains where relevant information may have occurred in an *unknown large* number of steps (Ni et al., 2023). Importantly, increasing $k^*$ causes an exponential increase in the dimensionality of the observation space, leading to both a severe increase in compute and storage, and potentially poor sample efficiency and generalisation.

However, many tasks may not actually require remembering everything. Often only a sparse subset of past observations is truly relevant for making optimal decisions. This insight aligns with findings in cognitive neuroscience: working memory in humans is known to have limited capacity and is thought to employ a selective gating mechanism that retains task-relevant information while filtering out irrelevant inputs (Unger et al., 2016). For example, a driver listening to a traffic report will update only the few road incidents relevant to her route into memory and ignore other trivial reports. Similarly, an RL agent with constrained memory should learn what to remember and what to forget. If the agent can identify which observations carry information critical for future reward, it could store just those and safely discard others, drastically reducing the burden on its memory and computation. Ideally, this is possible without sacrificing performance, but instead while actually improving generalisation.

Driven by this insight, we make the following main contributions: 1. **Adaptive Stacking**: We propose *Adaptive Stacking* (AS), a general meta-algorithm that learns to selectively retain observations in a working memory of fixed size $\kappa$ (Figure 1). When $\kappa \ll k^*$, this significantly improves compute and memory efficiency. It also leads to an exponential reduction in the size of the search space, which has implications for sample efficiency and generalisation. 2. **Theoretical analysis**: We then prove that agents using this approach are guaranteed to converge to an optimal policy in general when using unbiased value estimates, and in particular when using TD-learning under general assumptions. This enables practical trade-offs under the same resource constraints, such as the use of smaller memory to enable larger policy networks and the use of partial, instead of full, observations for better generalisation. 3. **Empirical analysis**: We run comprehensive experiments on memory intensive tasks using standard algorithms like Q-learning and PPO. Results demonstrate that AS generally leads to better memory management and sample efficiency than FS with $\kappa$ memory (when $k^*$ is unknown), while having comparable sample efficiency to FS with $k^*$ memory (when $k^*$ is given by an oracle).

## 2 PROBLEM SETTING

**The Environment.** We are interested in non-Markovian environments, which can be modelled as a Non-Markovian Decision Process (NMDP). Here, an agent interacts in an environment receiving observations $x_t \in \mathcal{X}$ at each step $t \in \{0, 1, ..., T\}$ and producing action $a_t \in \mathcal{A}$, where $T$ is the length of an episode (or the lifetime of the agent in non-episodic settings). The agent's action causes the environment to transition to a new observation $x_{t+1} \in \mathcal{X}$ and also provides the agent with a scalar reward $r_{t+1} \in \mathbb{R}$. The environment is $k^*$-order Markovian (i.e. a $k^*$-order Markov Decision Process (Puterman, 2014)), meaning that $k^* \in \mathbb{N}$ is the smallest number such that the probability function $Pr(x_{t+1}, r_{t+1}|x_{t:t-k^*}, a_t)$ is stationary regardless of the agent's policy, where $x_{t:t-k^*}$ includes the last $k^*$ observations.[1] If $k^* = 1$, then this is a standard Markov Decision Processes (MDP). We are interested in designing realistic computationally limited agents that can perform in environments where $k^*$ is very large. Note that our setting closely mirrors that of partially observable Markov decision processes (POMDP) (Kaelbling et al., 1998) where the last $k^*$ observations constitute a sufficient statistic of the state of the environment. In our work, discussion of the environment state is not necessary as we make no attempt to build a formal belief state as is commonly done in POMDPs. The notion of a memory state that we focus on building can be far more compact at scale.

**The Agent.** The agent acts in the environment using a policy $\pi(a_t|x_{t:t-k^*})$, which can be characterised by a value function $V^\pi(x_{t:t-k^*}) := \mathbb{E}_{a_t \sim \pi, (x_{t+1}, r_{t+1}) \sim Pr} \left[ \sum_{t=0}^{\infty} \gamma^t r_{t+1}|x_{t:t-k^*} \right]$. The agent's objective is to learn an optimal policy $\pi^*$ that maximizes their long-term accumulated reward, characterised by the optimal value function $V^*(x_{t:t-k^*}) = \max_\pi V^\pi(x_{t:t-k^*})$. However, the agent must learn $\pi^*$ with finite computational resources including a working memory $w$ (i.e. RAM) of finite capacity (in bits) $|w| \leq |w|^*$, and computational resources $c$ of finite capacity (in allowable floating point operations per environment step) $|c| \leq |c|^*$ split across both inference and learning. The size and architecture of the agents parameters $\theta$ must be chosen such that the two resource limits are always respected. Most recent progress in AI has been driven by sequence models (e.g. Transformers or RNNs), which in our setting would learn a policy of the form $\pi_\theta(a_t|x_{t:t-k})$. A fully differentiable sequence model has at least a linear dependence with respect to the sequence length $k$ for the working memory size i.e. $|w| \in \Omega(k)$ and computation i.e. $|c| \in \Omega(k)$ during inference and learning.[2]

**The Problem.** For a fully differentiable sequence model to learn in environments with large $k^*$, we must then correspondingly decrease the model size $|\theta|$ so that we can accommodate for the agent's limitations in terms of working memory $|w|^*$ and computational resources $|c|^*$. However, in many environments with high $k^*$, only $\kappa \ll k^*$ observations are actually needed to predict the environment dynamics. Thus $k^*$ is only large because the relevant observations are spaced apart by long temporal distances, not because there are many relevant observations to consider. So then if we learn to maintain a memory of size $k^* \geq k \geq \kappa$ with RL, we can potentially improve the efficiency of computation and working memory by a factor of $\Omega(k^*/\kappa)$ and increase $|\theta|$ at the same resource budget. Additionally, such an abstraction will induce a policy search space reduction of $\mathcal{O}(|\mathcal{X}|^{k^*-\kappa})$, which could lead to improvements in sample efficiency and generalisation for a policy using it. In this work, we consider approaches for achieving this goal with deep sequence models.

## 3 RELATED WORK

**Agents without working memory.** Foundational work has shown that settings that violate the Markov property introduce substantial complexity. For example, Singh et al. (1994) and Talvitie & Singh (2011) demonstrated that applying standard TD-learning in POMDPs leads to biased value estimates. Classical solutions attempt to address this problem by maintaining a belief state (distribution over states) as a sufficient statistic of the history (Kaelbling et al., 1998; Friston, 2009; Sajid et al., 2021). However, exact belief-state planning is intractable for complex environments, so modern RL agents rely on learned memory or state representations without full state estimation.

**RNN-based Agents.** To address the input dimensionality explosion of Frame Stacking, recurrent neural networks (RNNs) such as Long Short Term Memories (LSTMs) and Gated Recurrent Units

---

[1]For clarity and without loss of generality, we only consider the history of observations and not the history of actions and rewards, since these can always be included in the observations as well.

[2]For the popular Transformer architecture, it is actually even worse $|c| \in \Omega(k^2)$.

(GRUs) have been employed (Hausknecht & Stone, 2015), offering a learned internal state representation. Yet, these architectures often struggle in long-horizon tasks due to: vanishing gradients during back-propagation through time (BPTT) over full histories, limited capacity, and sensitivity to training dynamics (Singh et al., 1994; Ni et al., 2021; Javed et al., 2023; 2024).

**Transformer-based Agents.** In a separate line of work, Transformers have been increasingly applied to RL settings due to their success in natural language processing (NLP) (Vaswani et al., 2017). Self-attention allows these models to learn to focus on relevant past events and scale to longer memory horizons. For example Parisotto et al. (2020) proposed GTrXL, demonstrating improved stability over LSTMs. Chen et al. (2021) later introduced the Decision Transformer, a sequence model for offline RL. Recent works have also proposed ways of making Transformers more efficient (Pramanik et al., 2023), and even combining them with RNNs (Cherepanov et al., 2023). Most relevant to this work, Ni et al. (2023) rigorously studied the separation of memory length and credit assignment. They showed that Transformers can remember cues over a relatively large number of steps in synthetic T-Maze tasks, but struggle with long-term credit assignment. These works still depend on maintaining a memory stack of length $k^*$ using FS.

**Agents Agnostic to Sequence Models.** Several works attempt to bypass the exponential blow-up in agent states by learning compact, predictive memory representations for arbitrary sequence models. Allen et al. (2024) introduced $\lambda$-*discrepancy*, a measure of the deviation between TD targets with and without bootstrapping. They prove that this discrepancy is zero in fully observed MDPs and positive in POMDPs, offering a diagnostic and learning signal for memory sufficiency. Alternative strategies include learning which observations are worth remembering, such as the *Act-Then-Measure* framework (Krale et al., 2023) which lets agents actively choose when to observe their state, balancing the cost of memory against its value. Most closely related to our work is the line of works that provide the agent with additional actions to explicitly manage the memory stack (Peshkin et al., 1999; Demir, 2023). Demir (2023) is probably the most directly relevant work to our paper as the memory actions they consider are *push* (add an element to the top of the stack) and *skip* (do nothing), which they learn using intrinsic motivation (see Appendix E.4 for an expanded discussion).

However, all of these works still use Frame Stacking when the stack is full, always removing the last observation from the stack when space is needed for a new observation. Hence their baselines are generally other sequence models (Lu et al., 2023; Allen et al., 2024) or even RL algorithms (Krale et al., 2023; Demir, 2023) all using FS. In contrast, our focus in this work is investigating Adaptive Stacking as an alternative to Frame Stacking, by having an agent learn *which* observations to remove from its stack irrespective of the base RL algorithm or sequence model being used (Figure 1).

## 4 ADAPTIVE STACKING

We propose *Adaptive Stacking* as a general-purpose memory abstraction for reinforcement learning in partially observable environments. Adaptive Stacking extends the common frame stacking heuristic by endowing the agent with control over which past observations to retain in a bounded memory stack of size $k$. Rather than passively retaining the most recent $k$ observations, the agent *actively decides* which observation to discard, including the current observation. This transforms memory management into a decision-making problem aligned with maximizing reward.

**Motivating Examples.** Consider the TMaze environment (Figures 1,3), a canonical memory task from neuroscience (O'Keefe & Dostrovsky, 1971) which we adapt similarly to prior work in the field of RL (Bakker, 2001; Osband et al., 2019; Hung et al., 2019; Ni et al., 2023): 1. **Passive-TMaze task:** The agent begins in a corridor with a color-coded goal indicator (green ▨ or red ▨), then proceeds through a long grey corridor ▨ to the blue junction ▨ where the correct turning direction depends on the goal shown at the start. Here, the agent only needs to remember the goal cue in order to pick the correct goal at the junction cue, and doesn't need to learn to navigate in the maze. 2. **Active-TMaze task:** Here, the agent must both learn to navigate to find the goal color and remember it to navigate to the corresponding goal location. While the necessary memory length is bounded for Frame Stacking in the Passive-TMaze ($k^* = L + 2$), this memory threshold only holds for the optimal policy in the Active-TMaze. Indeed, the memory requirement can grow indefinitely ($k^* = \infty$) since a sub-optimal policy could stay arbitrarily long in some cells, while the true memory requirement to solve the tasks remains unchanged ($\kappa = 2$). See a more detailed description of both tasks in Appendix C.

### 4.1 RL WITH INTERNAL MEMORY DECISIONS

Formally, Adaptive Stacking induces a new decision process where the agent at each timestep $t$ receives an observation $x_t \in \mathcal{X}$ and maintains a memory stack $s_t = [x_{i_1}, \ldots, x_{i_k}]$ containing $k$ selected past observations indexed by their relative timesteps in which the last element is always $x_{i_k} = x_t$. We will refer to this stack $s_t$ as the *memory* or *agent state* (Dong et al., 2022). Upon receiving $x_{t+1}$, the agent executes two actions: an *environment action* $a_t \in \mathcal{A}$, and a *memory action* $i_t \in \{1, \ldots, k\}$ selecting which observation to pop. The agent state is then updated as:

$$s_{t+1} = \text{push}(\text{pop}(s_t, i_t), x_{t+1}). \tag{1}$$

In general, this process induces a new POMDP $\mathcal{M}_k = \langle \mathcal{M}, \mathcal{S}, \mathcal{I}, u \rangle$ where $\mathcal{M}$ is the original POMDP, $\mathcal{S}$ is the set of agent states, $\mathcal{I}$ is the set of *memory management actions*, and $u : \mathcal{S} \times \mathcal{I} \times \mathcal{X} \to \mathcal{S}$ is a *memory update function* (such as Equation 1). Hence, the approach is also compatible with modern architectures such as Transformers by simply defining $\mathcal{S}$, $\mathcal{I}$, and $u$ appropriately. The agent's policy is now $\pi_k(a_t, i_t | s_t)$, which can be characterised by a value function $V_k^{\pi_k}(s_t) := \mathbb{E}_{(a_t, i_t) \sim \pi_k, (x_{t+1}, r_{t+1}) \sim Pr} \left[ \sum_{t=0}^{\infty} \gamma^t r_{t+1} | u(s_t, i_t, x_{t+1}) \right]$. Its objective is now to learn an optimal policy $\pi_k^*$ that maximizes its long-term accumulated reward, characterised by the optimal value function $V_k^*(s_t) = V_k^{\pi_k^*}(s_t) = \max_{\pi_k} V^{\pi_k}(s_t)$.

Consider for example the Passive-TMaze example with corridor length $L = 3$, so that $k^* = L+2 = 5$. Figure 2 contrasts Frame Stacking and Adaptive Stacking with $k = 4$. FS, due to its FIFO nature ($i_t = 0$ for all $t$), forgets the goal signal when the maze is longer than the context window ($k < k^*$). In contrast, the optimal AS policy $\pi_k^*$ learns to retain only the green goal indicator and discard irrelevant grey observations (thereby solving the task with a much smaller memory budget). Importantly, notice that discarding those irrelevant grey observations makes the problem non-Markovian and changes the agent's perceived optimal values. For example at timestep $t = 1$, the memory state is $s_t = $ ▮▮▯▯. However, multiple latent histories $x_{t:t-k^*}$ are compatible with this state: $x_{t:t-k^*} \in \{$▮▮▮▯▯, ▮▮▯▮▯, ▮▯▮▯▯$\}$. This gives:

$$V_2^{\pi_2^*}(\text{▮▮▯▯}) = \frac{1}{3}V^{\pi_2^*}(\text{▮▮▮▯▯}) + \frac{1}{3}V^{\pi_2^*}(\text{▮▮▯▮▯}) + \frac{1}{3}V^{\pi_2^*}(\text{▮▯▮▯▯}) = \frac{1}{3}(\gamma^3 + \gamma^2 + \gamma).$$

However, the actual latent history at time $t = 1$ is $x_{t:t-k^*} = $ ▮▮▮▯▯, and the true optimal value is: $V^*(x_{t:t-k^*}) = \gamma^3$. This induces a value gap $|V^*(x_{t:t-k^*}) - V_2^{\pi_2^*}(s_t)| > 0$, but $\pi_2^*$ is still optimal since $V^{\pi_2^*}(x_{t:t-k^*}) = V^*(x_{t:t-k^*})$, even though $s_t$ is not a sufficient statistic of the $k^*$-history $x_{t:t-k^*}$. Figure 7b shows the value gap for varying T-Maze lengths. This illustrates a crucial point:

**Remark 1** *Uncertainty in history may harm value expectations, $|V^*(x_{t:t-k^*}) - V_k^{\pi_k^*}(s_t)| > 0$, but it does not necessarily harm policy optimality as long as the uncertain differences are irrelevant for optimal decision making: $V^*(x_{t:t-k^*}) = V^{\pi_k^*}(x_{t:t-k^*})$.*

By integrating the memory update into the RL loop (see Figure 1), Adaptive Stacking fits cleanly into existing learning pipelines similarly to Frame Stacking (for example Q-learning as shown in Algorithm 1) and can be trained end-to-end. In this view, Adaptive Stacking transforms memory selection into a sequential decision-making problem aligned with the agent's reward signal. This stands in contrast to passive memory mechanisms based on Frame Stacking, which indiscriminately process all inputs.

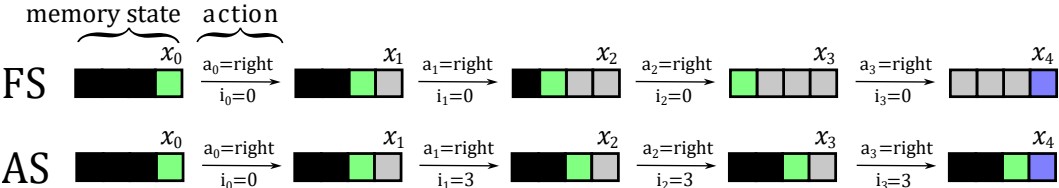

Figure 2: Illustration of Frame Stacking (FS) and Adaptive Stacking (AS) with $k = 4$ in the passive-TMaze with $k^* = L + 2 = 5$. In order to free up space to push the new observation $x_t$, the FS agent always pops the last observation ($i_t = 0$) while the AS agent chooses which observation to pop ($i_t \in \{1, \ldots, k\}$). In this figure, ▮ corresponds to an empty memory slot.

This also aligns with cognitive models of working memory in humans, where attention-gated memory buffers retain only task-relevant cues while filtering distractors (Unger et al., 2016). However, this raises an important question: *How does selective forgetting affect the standard theoretical guarantees established for the convergence of RL agents such as value function and policy optimality?*

## 4.2 Monte Carlo Value Function Estimates

Recent work for reasoning with large language models has highlighted the effectiveness of Monte Carlo estimates of the value function (Shao et al., 2024) as originally pioneered by the REINFORCE policy gradient algorithm (Williams, 1992). An advantage of this kind of algorithm is that its estimates of the value function are unbiased as they are formed based on rolling out the policy for sufficiently long in the environment itself. This greatly simplifies convergence to optimality in the limited memory setting (Allen et al., 2024). We can then consider a notion of the minimal sufficient memory length.

**Definition 1** *Define $\kappa$ to be the smallest memory length such that there exists a policy $\pi_\kappa^*$ satisfying $V^{\pi_k^*}(x_{t:t-k^*}) = V^*(x_{t:t-k^*})$ for all $t$.*

This characterises the minimal task-relevant context size needed to act optimally in environments with large $k^*$, and motivates the central promise of Adaptive Stacking: optimal memory management via reward-guided memory decisions. $\kappa$ always exists since in the simplest case we can have $\kappa = k^*$ (Proposition 1), as shown in Figure 7a when the maze length is 2 (when $L = 0$). Hence, any unbiased RL algorithm that is guaranteed to converge to optimal policies under Frame Stacking with $k = k^*$ is also guaranteed to converge to optimal policies under Adaptive Stacking with $k = \kappa$ (Theorem 1).

**Proposition 1** *If $k = k^*$, then there exists a $\pi_k^*$ such that $V^{\pi_k^*}(s_t) = V^*(x_{t:t-k^*})$ for all $s_t \in \mathcal{S}$.*

**Theorem 1** *Let $\mathbb{A}$ be an RL algorithm that converges under Frame Stacking with $k \geq k^*$. If $\mathbb{A}$ uses unbiased value estimates to learn optimal policies, then it also converges under Adaptive Stacking with $k \geq \kappa$ observations, assuming the policy class is sufficiently expressive.*

See Appendix B for all the proofs. This implies that algorithms that leverage Monte Carlo return based value functions for policy gradient updates can be shown to converge to the optimal policy with standard conditions regarding exploration and the policy parameterization. We also prove convergence for linear policies in the episodic setting and the standard continuing average reward setting. In line with prior work on RL, non-linear policies in general can only be shown to converge to local optima. This has significant implications for compute and memory efficiency when using sequence models like Transformers. Transformers incur compute costs of $\Omega(k^2)$ and working memory costs of $\Omega(k)$ due to self-attention over long contexts (Narayanan et al., 2021; Anthony et al., 2023). Adaptive Stacking reduces these to $\Omega(\kappa^2)$ and $\Omega(\kappa)$ respectively, by retaining only reward-relevant observations of length $\kappa \ll k^*$, thereby yielding substantial efficiency gains in both inference and training shown in Table 1 (see Appendix F for derivations).

**The need for bootstrapped value estimates.** It is also important to note that there are scaling issues regarding using Monte Carlo returns for value function estimates in continuing environments as in Appendix B.5. For true continual RL environments in big worlds, the amount of steps needed for these unbiased rollouts becomes unwieldy (Riemer et al., 2022; Khetarpal et al., 2022). Additionally, Riemer et al. (2024) demonstrated that this amount of steps increases with the agent's memory size. Thus is it will eventually be necessary to use truncated returns with bias inserted from bootstrapped value estimates to tackle the challenging futuristic environments that our paper is inspired by.

| Architecture | Memory Type | $\|c\|_{a \sim \pi_\theta}$ | $\|c\|_{\text{TD}}$ | $\|w\|_{a \sim \pi_\theta}$ | $\|w\|_{\text{TD}}$ |
|---|---|---|---|---|---|
| MLP or LSTM | Frame Stack | $\Omega(k^*)$ | $\Omega(k^*)$ | $\Omega(k^*)$ | $\Omega(k^*)$ |
| MLP or LSTM | Adaptive Stack | $\Omega(\kappa)$ | $\Omega(\kappa)$ | $\Omega(\kappa)$ | $\Omega(\kappa)$ |
| Transformer | Frame Stack | $\Omega(k^{*2})$ | $\Omega(k^*)$ | $\Omega(k^{*2})$ | $\Omega(k^*)$ |
| Transformer | Adaptive Stack | $\Omega(\kappa^2)$ | $\Omega(\kappa)$ | $\Omega(\kappa^2)$ | $\Omega(\kappa)$ |

Table 1: Compute $|c|$ and memory $|w|$ requirements for computing actions $a \sim \pi_\theta$ and TD updates.

### 4.3 ADAPTIVE STACKING AS A FORM OF STATE ABSTRACTION

While Adaptive Stacking is designed to learn which past observations to retain, a key theoretical question is how this compression affects the ability of RL agents to preserve optimal behaviour. Specifically, we want to understand how the value function under Adaptive Stacking relates to the value function under full-history policies. We first observe that there is a general relationship between the adaptive stack value function and the underlying full-history value function: $V_k^{\pi_k}(s_t) = \sum_{x_{t:t-k^*}} Pr(x_{t:t-k^*}|s_t, \pi_k)V^{\pi_k}(x_{t:t-k^*})$ for all $s_t \in \mathcal{S}$, where $Pr(x_{t:t-k^*}|s_t, \pi_k)$ is the asymptotic probability amortized over time that the environment $k^*$-history is $x_{t:t-k^*}$ when the agent state is $s_t$ under policy $\pi_k$ (Singh et al., 1994). This equation shows that the agent's value under compressed memory is an expectation over possible latent histories. When the memory stack discards critical observations, this conditional distribution becomes broader, increasing uncertainty. As such, it is clear that Adaptive Stacking can be seen as a form of state abstraction in which multiple histories are compressed together in the estimate of the value function.

**Model-equivalent abstractions.** One of the most popular form of state abstractions are based on the idea that a state abstraction itself should be able to reconstruct the rewards received in the environment and state transitions in its own abstract space. A number of methods for learning these kinds of abstractions have been proposed (Zhang et al., 2019; 2020; Tomar et al., 2021) – often called "bisimulation" abstractions. As discussed by Li et al. (2006) this class of abstractions allows for convergent TD learning, but results in the least compression among popular techniques.

**Value-equivalent abstractions.** A more ambitious kind of abstraction that results in more compression while still ensuring convergent TD learning is based on the value equivalence principle (Li et al., 2006; Abel, 2022). This class of abstractions requires that, given a policy (or the optimal policy), its value function conditioned on the true history is equal to that conditioned on the state abstraction. While even more compressed state abstractions exist that preserve the optimal policy, it cannot be shown that they lead to convergent TD learning in the general case (Li et al., 2006).

**An even more powerful class of abstractions.** While it is not possible to show this convergence in general, it is possible to exploit the fact that Adaptive Stacking is a very particular form of structured state abstraction in that it maintains actual observations from the environment. Indeed, The **T-Maze** counter example in Section 4.1 shows how the optimal policy can be learned from a form of state abstraction that does not preserve the standard value-equivalence property and thus results in even more compression. In general we can show that in tasks where uncertainty in the full history is not relevant for value prediction (Assumption 4.1), Adaptive Stacking preserves the relative ordering between policies (Theorem 2), which is a sufficient condition for TD convergence.

**Assumption 4.1 (Value-Consistency)** *Let $\pi_k$ be an Adaptive Stacking policy over memory states $s_t \in \mathcal{S}_k$. We say the memory representation is* value-consistent *with respect to $\pi_k$ if, for all full histories $x_{t:t-k^*}$ and $x'_{t:t-k^*}$ such that both $Pr(x_{t:t-k^*} \mid s_t, \pi_k) > 0$ and $Pr(x'_{t:t-k^*} \mid s_t, \pi_k) > 0$, it holds that $V^{\pi_k}(x_{t:t-k^*}) = V^{\pi_k}(x'_{t:t-k^*})$ for all $s_t \in \mathcal{S}$.*

**Theorem 2 (Partial-order Preserving)** *Consider an agent with a value-consistent memory stack of arbitrary length $k \in \mathbb{N}$. Let $\pi_k^1$ and $\pi_k^2$ be two arbitrary Adaptive Stacking policies such that $V_k^{\pi_k^1}(s_t) \leq V_k^{\pi_k^2}(s_t)$ for all $t$. Then $V^{\pi_k^1}(x_{t:t-k^*}) \leq V^{\pi_k^2}(x_{t:t-k^*})$ for all $t$.*

This key result allows us to extend convergence results from traditional RL to our setting. That is, any RL agorithm that converges to the optimal policy under Adaptive Stacking simultaneously converges to the optimal policy over the underlying history when $k \geq \kappa$. For example, Singh et al. (1994) (in Theorem 1) show that in POMDPs, policy evaluation of a policy $\pi_k$ using TD(0) under standard assumptions converges to a fixed point value function $V_{TD}^{\pi_k}(s_t)$ that is generally *lower* than the true expected return $V_k^{\pi_k}(s_t)$, due to uncertainty over hidden state. Hence TD-learning also preserves partial-ordering. Similary, we can show that Q-learning still converges to optimal policies (Theorem 5). While these results rely on Assumption 4.1, it happens to always hold for a wide range of tasks of interest in RL, such as goal-reaching tasks with non-zero rewards only for reaching goal states. See Appendix C for a list of relevant benchmarks for which this assumption holds. As this is the case for the popular environments we consider in Section 5, we do not need to consider any form of explicit state abstraction supervision in our experiments.

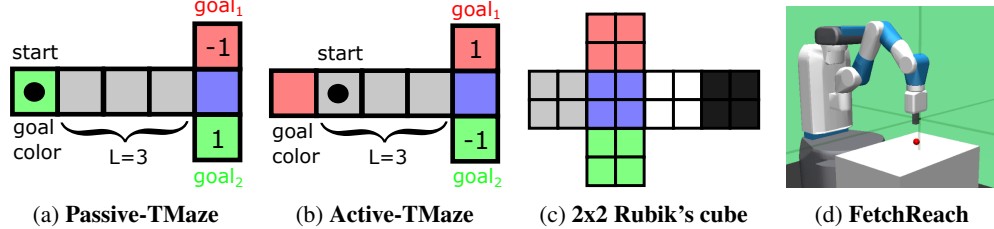

(a) **Passive-TMaze**    (b) **Active-TMaze**    (c) **2x2 Rubik's cube**    (d) **FetchReach**

Figure 3: Experimental domains ($10^6$ training steps for all). (a-b) are as described in Section 4. (c) Episodic ($T = 100$) where the agent only sees one cube face at a time. (d) Continual with hidden joint velocities. The goal position also changes every 50 steps and is only observable for the first 2 steps after each change (Assumption 4.1 does not hold). See Appendix D.1 for detailed descriptions.

## 5 EXPERIMENTS

We evaluate Adaptive Stacking on a variety of challenging memory tasks (Figure 3) to assess both learning performance, memory management, and generalisation. We compare against two baselines: FrameStack with $k = \kappa$ (insufficient memory) and $k = k^*$ (oracle memory), and report four key metrics here: (1) **Returns**: cumulative discounted rewards, (2) **Memory regret**: number of steps when the goal cue is absent from memory, (3) **Active memory regret**: steps where the goal cue is seen but not added to memory, and (4) **Passive memory regret**: steps where the goal cue is removed from memory. All error bars represent one standard deviation across a number of random seeds ($N_{rs}$).

**Continual TMaze with Q-learning.** We first evaluate in a continual Passive and Active TMazes, where episodes do not terminate, and rewards are only given at goal transitions. This stresses the agent's ability to persist and discard information appropriately. Results in Figure 4 show that Adaptive Stacking achieves high returns and low reward regret, consistent with theoretical predictions. When $\kappa = k^*$ (maze length 2), all methods perform similarly. But when $\kappa < k^*$, AS($\kappa$) retains significantly lower passive memory regret than FS($\kappa$), learning to preserve goal cues over long delays. Note that FS($k^*$), even in the Passive-Tmaze, still incures some total memory regret for not having the goal cue in memory each time the agent re-spawns at the start location. See Appendix E for the learning curves.

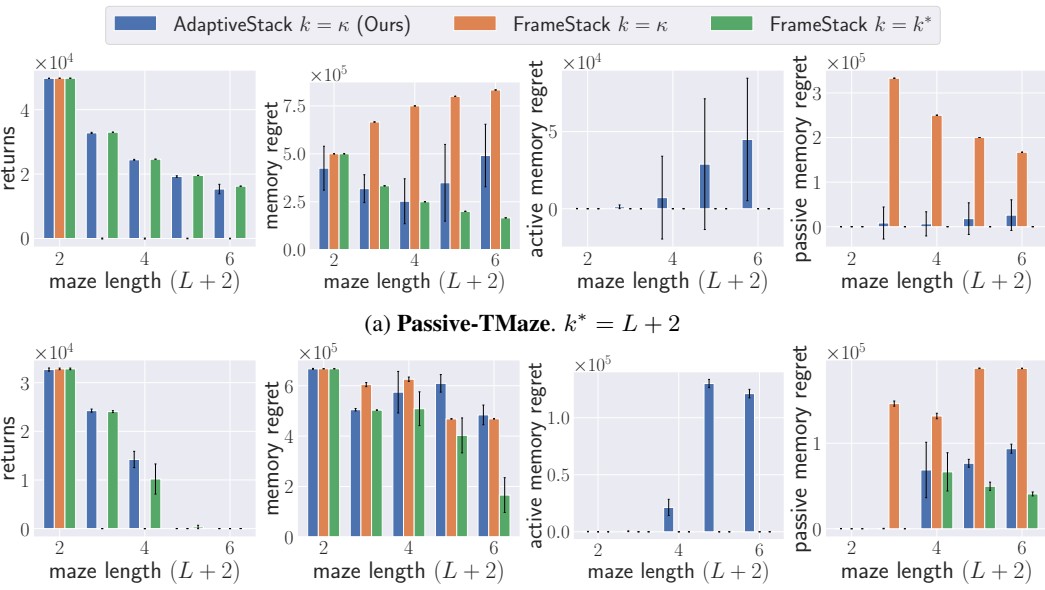

(a) **Passive-TMaze**. $k^* = L + 2$

(b) **Active-TMaze** $k^* = \infty$ theoretically, but for practicality we use $k = L + 2$ for the oracle Framestack.

Figure 4: Continual TMazes ($T = 10^6$) with Q-learning ($N_{rs} = 20$). AS matches the oracle FS($k^*$) in returns and memory usage, while outperforming FS($\kappa$) especially for long-term dependencies.

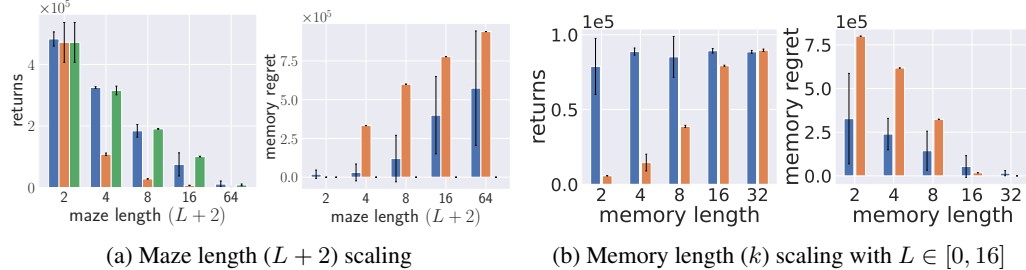

(a) Maze length ($L + 2$) scaling   (b) Memory length ($k$) scaling with $L \in [0, 16]$

Figure 5: Episodic **Passive-TMaze** ($k^* = T = L + 2$) with PPO using an MLP ($N_{rs} = 10$). AS learns to retain the goal cue regardless of maze and memory lengths, unlike FS with the same $k$.

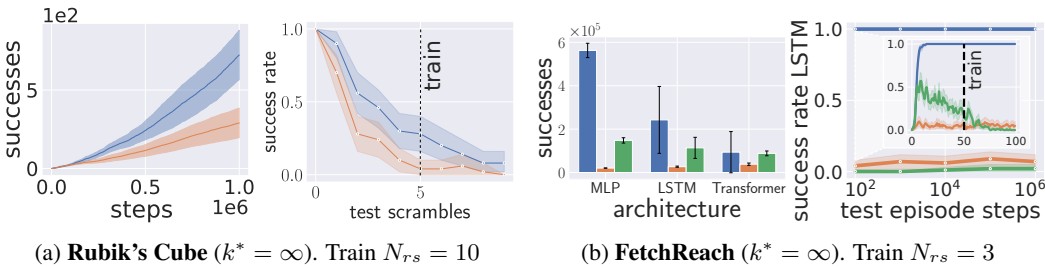

(a) **Rubik's Cube** ($k^* = \infty$). Train $N_{rs} = 10$   (b) **FetchReach** ($k^* = \infty$). Train $N_{rs} = 3$

Figure 6: Generalisation results with PPO (test $N_{rs} = 100$). We show successes during training (left) and test success rates (right). The shaded regions show 95% confidence intervals. (a) uses an MLP.

**Episodic TMaze with PPO.** We further evaluate AS in episodic Passive-TMaze using PPO in variable maze and memory stack lengths (Figure 5). Similarly to the tabular case, we observe that AS($\kappa$) still significantly outperforms FS($\kappa$) and matches the oracle $k^*$ baseline. Importantly, we also observe that these results are consistent regardless of whether the architecture is an MLP, LSTM, or even a Transformer (Figures 10-11). Additionally, when testing on a much larger maze of length $10^6$, AS($\kappa$) still solves the task with 100% accuracy, regardless of architecture (Figures 1). In contrast, the oracle FS previously trained with $k = 16$ now fails for all architectures except for the LSTM. This because the LSTM (using AS or FS) only maintains a recurrent hidden state during testing. This difference in generalisation may also be attributed to the compact agent state representations learned by AS (Figures 24-28), since observations irrelevant for reward maximisation are discarded.

**Generalisation to other representative domains.** Finally, we investigate the performance of AS on significantly more complex tasks: The **Rubik's cube** trained with $k = 10$ for both FS and AS; The **FetchReach** trained with $k = 50$ for FS($k^*$) for practicality, and $k = \kappa = 4$ for both FS($\kappa$) and AS($\kappa$)) (Figure 6). AS achieves significantly higher training successes than FS with the same $k$ in both domains, and even higher than the oracle FS($k^*$) in **FetchReach**. This demonstrates the inability of FS to learn under limited memory, and it's sample inefficiency when learning from full histories (consistent with results from Ni et al. (2023)). Finally, AS with only $k = 4$ memory even generalises from remembering the goal after 50 steps to still remembering it after $10^6$ steps. In contrast, the oracle degrades quickly to 0% success rates irrespective of architectures (Figure 29).

## 6    CONCLUSION

We have introduced **Adaptive Stacking**, a general-purpose meta-algorithm for learning to manage memory in partially observable environments. Unlike standard Frame Stacking, which blindly retains recent observations, Adaptive Stacking allows agents to learn which observations to remember or discard via reinforcement learning. We showed that this yields theoretical guarantees on policy optimality under both unbiased optimization and TD-based learning, even when using a significantly smaller memory than required for full observability. Experiments across multiple TMaze tasks confirm that Adaptive Stacking matches the performance of oracle memory agents while using far less memory, and substantially outperforms naive baselines under tight memory budgets. This offers a promising path toward scalable, memory-efficient RL in large, partially observable environments.

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

## A    ADAPTIVE STACKING ALGORITHM

---

**Algorithm 1:** Q-Learning: Adaptive Stacking

---

**Input**      : discounting $\gamma = 0.99$, learning rate $\alpha = 0.01$, exploration $\epsilon = 0$, memory length $k$
**Initialise :** value function $Q(s, \langle a, i \rangle) = R_{\text{MAX}}$
**foreach** *episode* **do**
    Get initial observation $x_0 \in \mathcal{X}$
    Initialise observation stack $s_0 \leftarrow [x_0]_k$  // e.g.   $s_0 = [x_0, x_0]$ if $k = 2$
    **foreach** *timestep $t = 0, 1, ..., T$ while episode is not done* **do**
$$\langle a_t, i_t \rangle \leftarrow \begin{cases} \arg\max_{\langle a,i \rangle} Q(s_t, \langle a, i \rangle) & \text{w.p. } 1 - \varepsilon \\ \text{a random action} & \text{w.p. } \varepsilon \end{cases}$$
        Execute $a_t$, get reward $r_{t+1}$ and next observation $x_{t+1}$
        Remove observation from stack $s_{t+1} \leftarrow pop(s_t, i_t)$
        Push observation into stack $s_{t+1} \leftarrow push(s_{t+1}, x_{t+1})$
        $Q(s_t, \langle a_t, i_t \rangle) \xleftarrow{\alpha} \left( r_{t+1} + \gamma \max_{\langle a,i \rangle} Q(s_{t+1}, \langle a, i \rangle) \right) - Q(s_t, \langle a_t, i_t \rangle)$

---

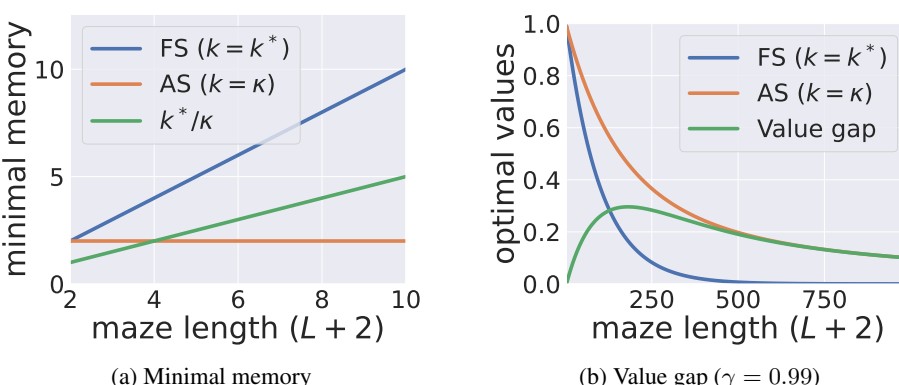

(a) Minimal memory          (b) Value gap ($\gamma = 0.99$)

Figure 7: **TMaze** Memory and value scaling in the **TMaze** environments. (a) Minimal memory stack required when using Frame Stacking vs Adaptive Stacking in either task. (b) The value gap between the optimal Frame Stacking values ($V^*$) and Adaptive Stacking values ($V_k^{\pi_k^*}$) when an agent has observed ▦ then ▦ under their respective optimal policies.

## B    THEORETICAL RESULTS

We begin by restating the key definitions and then give precise statements and proofs for all the theoretical results in the main paper.

### B.1    PRELIMINARIES AND NOTATION

Let the underlying non-Markovian environment be a $k^*$-order MDP over observations $x_t \in \mathcal{X}$, with full-history value

$$V^*(x_{t:t-k^*}) = \max_{\pi} \mathbb{E}\left[ \sum_{h=0}^{\infty} \gamma^h r_{t+h+1} \mid x_{t:t-k^*}, \pi \right].$$

Under Adaptive Stacking (AS) with memory size $k$, the agent memory state is $s_t = [x_{i_1}, \ldots, x_{i_k}]$ and its value under policy $\pi_k$ is

$$V_k^{\pi_k}(s_t) = \mathbb{E}\left[ \sum_{h=0}^{\infty} \gamma^h r_{t+h+1} \mid s_t, \pi_k \right] = \sum_{x_{t:t-k^*}} \Pr\left( x_{t:t-k^*} \mid s_t, \pi_k \right) V^{\pi_k}(x_{t:t-k^*}), \quad (2)$$

where $V^{\pi_k}(x_{t:t-k^*})$ is the full-history value of $\pi_k$ using Frame Stacking (FS).

We define
$$\kappa \;=\; \min\big\{k \in \mathbb{N} : \exists\, \pi_k^* \text{ with } V^{\pi_k^*}(x_{t:t-k^*}) = V^*(x_{t:t-k^*})\ \forall t\big\}.$$

## B.2 Proof of Proposition 1

**Proposition 1** *If $k = k^*$, then there exists a policy $\pi_k^*$ such that $V^{\pi_k^*}(s_t) = V^*(x_{t:t-k^*})$ for all $t$.*

**Proof** When $k = k^*$, the Adaptive Stacking agent can simply retain the last $k^*$ observations in order, equivalently to Frame Stacking. Thus, no important information is discarded, and the agent can follow an optimal full-history policy $\pi_k^*$ on $s_t = [x_t, x_{t-1}, \ldots, x_{t-k^*}]$. Hence,

$$\Pr\big(x_{t:t-k^*} \mid s_t, \pi_k^*\big) = \begin{cases} 1 & \text{if } s_t = x_{t:t-k^*}, \\ 0 & \text{otherwise}, \end{cases}$$

implying $V_k^{\pi_k^*}(s_t) = V^{\pi_k^*}(s_t) = V^{\pi_k^*}(x_{t:t-k^*}) = V^*(x_{t:t-k^*})$. ∎

## B.3 Proof for Theorem 1

Theorem 1 in the main paper stated that traditional RL algorithms that converge under Frame Stacking with $k \geq k^*$ also converge under Adaptive Stacking, provided they use unbiased value estimates to learn optimal policies. We first formally state this unbiased convergence assumption:

**Assumption B.1 (Unbiased Convergence)** *Let $\mathbb{A}$ be an RL algorithm that converges under Frame Stacking with $k \geq k^*$. Assume that for any memory length $k \in \mathbb{N}$, $\mathbb{A}$ also converges to a $k$-order policy*

$$\pi_k^*(x_{t:t-k}) \;=\; \arg\max_{\pi_k} \hat{V}^{\pi_k}(x_{t:t-k}) \quad \forall t,$$

*where $\hat{V}^{\pi_k}(x_{t:t-k})$ is an unbiased estimator of the true return:*

$$\mathbb{E}\left[\hat{V}^{\pi_k}(x_{t:t-k})\right] = V^{\pi_k}(x_{t:t-k}).$$

We now restate the result more formally and provide a detailed proof.

**Theorem 1** *Let $\mathbb{A}$ be an RL algorithm that satisfies Assumption B.1. Then for any $k \geq \kappa$, $\mathbb{A}$ converges to an optimal Adaptive Stacking policy $\pi_k^*$ such that:*

$$V^{\pi_k^*}(x_{t:t-k^*}) = \max_{\pi} V^{\pi}(x_{t:t-k^*}) \quad \forall t.$$

**Proof** Adaptive Stacking with memory size $k$ induces a new decision process $\mathcal{M}_k$ in which the agent new observation is the memory state $s_t \in \mathcal{S}_k$, a stack of $k$ underlying environment observations. This process can be treated as a POMDP, where the true underlying state is the latent history $x_{t:t-k^*}$.

By definition of $\kappa$, there exists at least one $k$-order policy $\pi_k$ with $k \geq \kappa$ that achieves the optimal value on all underlying latent histories:

$$V^{\pi_k}(x_{t:t-k^*}) = V^*(x_{t:t-k^*}) \quad \text{for all } t.$$

Since $\pi_k$ acts on $s_t \in \mathcal{S}_k$ and implicitly induces a distribution over latent histories $x_{t:t-k^*}$, its value in the induced process is as shown in Equation 2. By construction of $\pi_k$, we have $V^{\pi_k}(x_{t:t-k^*}) = V^*(x_{t:t-k^*})$, so:

$$V_k^{\pi_k}(s_t) = \sum_{x_{t:t-k^*}} \Pr(x_{t:t-k^*} \mid s_t, \pi_k)\, V^*(x_{t:t-k^*}) = \mathbb{E}_{x_{t:t-k^*}}[V^*(x_{t:t-k^*}) \mid s_t].$$

This implies that the policy $\pi_k$ achieves the best possible value in the induced process $\mathcal{M}_k$ given that it is optimal over latent histories.

Now, because $\mathbb{A}$ uses unbiased estimates of $V^{\pi_k}(x_{t:t-k})$ and converges to the policy that maximizes expected return under such estimates (by Assumption B.1), and since $k \geq \kappa$ implies such a policy exists, it follows that $\mathbb{A}$ converges to $\pi_k^*$ that satisfies:

$$V^{\pi_k^*}(x_{t:t-k^*}) = V^*(x_{t:t-k^*}) \quad \forall t.$$

∎

The critical observation from Theorem 1 is that convergence to an optimal policy is not limited to $k \geq k^*$, but to any $k \geq \kappa$, where $\kappa$ is the minimal sufficient memory required to disambiguate value-relevant latent histories. The key assumption for this result is that $\mathbb{A}$ optimizes return estimates that are unbiased *with respect to the true value under the full history*, for example as achieved by Monte Carlo policy gradient methods like REINFORCE (Sutton & Barto, 2018).

We emphasize that this result does not extend to TD-based methods (which use biased targets), and is handled separately in Appendix B.7.

### B.4 CONVERGENCE OF EPISODIC MONTE CARLO LEARNING FOR OPTIMAL $\kappa$-MEMORY POLICIES

**Corollary 2 (Convergence of Episodic Monte Carlo Policy Gradient)** *Consider the episodic setting with finite horizon $T < \infty$. Let policies be softmax-linear:*

$$\pi_\theta(b \mid s) \;=\; \frac{\exp(\theta^\top \phi(s,b))}{\sum_{b'} \exp(\theta^\top \phi(s,b'))},$$

*where $\phi(s,b) \in \mathbb{R}^d$ are bounded features and parameter vector $\theta$ is constrained to a compact convex set $\Theta$. Let the learning algorithm $\mathbb{A}$ be Monte Carlo policy-gradient (REINFORCE) with entropy regularization (weight $\beta > 0$) and projected gradient updates (projection onto $\Theta$). Assume the entropy coefficient $\beta$ may be annealed to zero slowly so as to ensure persistent exploration during learning.*

*Then, under Assumption B.1, for any memory size $k \geq \kappa$ Adaptive Stacking trained with $\mathbb{A}$ converges to an optimal $\kappa$-sufficient Adaptive Stacking policy $\pi_k^*$ (i.e. it attains the full-history optimal value $V^*$ on all latent histories).*

**Proof** We show that REINFORCE with the stated parameterization satisfies the unbiasedness requirement of Assumption B.1 when estimating policy returns on the induced stack-process $\mathcal{M}_k$. Once unbiasedness is shown, Theorem 1 implies the corollary.

**1. Monte Carlo returns are unbiased in episodic setting.** Fix a stack-policy $\pi_k$ and an initial stack $s_t$. Let $\tau = (s_t, b_t, r_{t+1}, \ldots, s_{t+T})$ denote a finite-horizon trajectory generated by executing $\pi_k$ in the true environment. The Monte Carlo return

$$\hat{V}^{\pi_k}(s_t) \;=\; \sum_{n=0}^{T-1} \gamma^n r_{t+n+1}$$

is an unbiased estimator of the true expected return

$$V_k^{\pi_k}(s_t) \;=\; \mathbb{E}_{\tau \sim \pi_k}\Big[ \sum_{n=0}^{T-1} \gamma^n r_{t+n+1} \mid s_t \Big],$$

because expectation and sample-average commute (law of the unconscious statistician). This unbiasedness is purely a sampling fact and does not require the augmented-state to be Markov; it only requires that rollouts are generated according to the policy $\pi_k$ and the true environment dynamics.

**2. REINFORCE gradient estimator uses unbiased returns.** The REINFORCE policy-gradient estimator uses samples $\hat{V}^{\pi_k}(s_t)$ (possibly with a baseline) multiplied by $\nabla_\theta \log \pi_\theta(b_t \mid s_t)$. Since

$\hat{V}^{\pi_k}(s_t)$ is an unbiased estimator of the true return, the REINFORCE estimator is an unbiased estimator of the true policy gradient:

$$\mathbb{E}\Big[\nabla_\theta \log \pi_\theta(b_t \mid s_t)\, \hat{V}^{\pi_k}(s_t)\Big] \;=\; \nabla_\theta J(\theta),$$

where $J(\theta)$ is the (entropy-regularized) episodic objective. The softmax-linear parameterization with bounded $\phi$ and compact $\Theta$ ensures $\nabla_\theta \log \pi_\theta$ is bounded; projection onto $\Theta$ guarantees iterates remain bounded.

**3. Exploration via entropy regularization.** The entropy regularizer (with slowly annealed $\beta$) ensures the policy stays sufficiently stochastic during learning so that the sampling procedure visits the relevant stack-states and joint actions; this condition is part of the standard assumptions required for policy-gradient convergence in the episodic Monte Carlo setting (and is compatible with Assumption B.1).

**4. Apply Theorem 1.** Because REINFORCE provides unbiased estimates $\hat{V}^{\pi_k}(s_t)$ of $V_k^{\pi_k}(s_t)$ for any stack-policy $\pi_k$, the Unbiased Convergence Assumption is satisfied. Hence by Theorem 1, when $k \geq \kappa$ the algorithm $\mathbb{A}$ converges to an optimal Adaptive Stacking policy $\pi_k^*$ achieving the full-history optimum $V^*$. This completes the proof. ∎

### B.5 CONVERGENCE OF MEMORY-AUGMENTED MONTE CARLO POLICY GRADIENT IN UNICHAIN AVERAGE-REWARD NMDPS

**Corollary 3 (Convergence of Average-Reward Monte Carlo Policy Gradient)** *Consider the infinite horizon unichain average-reward setting: assume every stationary policy on the induced stack-process $\mathcal{M}_k$ yields a unichain Markov chain (single recurrent class) and the chain is aperiodic. Let the policy class be softmax-linear $\pi_\theta(b \mid s)$ as in Corollary 2 with bounded features and $\theta \in \Theta$ compact. Use entropy regularization (weight $\beta > 0$) and projected updates. Let $\mathbb{A}$ be Monte Carlo policy-gradient that estimates the average reward via long trajectory averages (or regeneration-based sampling) of length $L$ on the order of the chain's mixing time; assume $L$ is chosen (or scheduled) so that estimators are asymptotically unbiased in the SA sense described below.*

*Then, under Assumption B.1 and the unichain assumption above, for any $k \geq \kappa$ Adaptive Stacking trained with $\mathbb{A}$ converges (in the average-reward sense) to an entropy-regularized optimal Adaptive Stacking policy $\pi_k^*$ maximizing long-run average reward.*

**Proof** We organize the proof into the following steps: (i) show how to construct asymptotically unbiased average-reward estimators using long trajectories or regenerative sampling under the unichain assumption, (ii) note that with those estimators $\mathbb{A}$ satisfies the Unbiased Convergence Assumption, and (iii) apply Theorem 1.

**1. Unichain ergodicity $\Rightarrow$ time-average convergence.** Under the unichain and aperiodicity assumption for the induced stack-process $\mathcal{M}_k$, the Markov chain induced by any stationary policy $\pi_k$ has a unique stationary distribution $\mu^{\pi_k}$. By the ergodic theorem for Markov chains, for a single long trajectory $(s_0, b_0, r_1, s_1, b_1, r_2, \dots)$ generated under $\pi_k$ we have almost surely

$$\frac{1}{L}\sum_{t=0}^{L-1} r_{t+1} \xrightarrow[L\to\infty]{a.s.} \rho(\pi_k) \;=\; \sum_s \mu^{\pi_k}(s)\sum_b \pi_k(b \mid s) r(s,b),$$

the long-run average reward (gain). Thus the empirical average over a sufficiently long trajectory is an asymptotically unbiased estimator of $\rho(\pi_k)$. Alternatively, if regenerative sampling is available (returns to a recurrent state), one can form i.i.d. regenerative cycles and obtain unbiased cycle-averages; both approaches are standard ways to estimate average reward unbiasedly on unichain chains.

**2. Constructing an (approximately) unbiased differential-return estimator for gradients.** Policy-gradient formulas in the average-reward setting require estimating $\nabla_\theta \rho(\theta)$, which can be

written in terms of stationary expectations involving the differential value function (Poisson solution). A practical Monte Carlo estimator uses centered finite-horizon partial returns with empirical centering by the block average, e.g. for a block of length $L$:

$$\widehat{U}_t \;=\; \sum_{k=0}^{L-1-t} \big(r_{t+k+1} - \bar{r}^{(L)}\big), \quad \bar{r}^{(L)} := \frac{1}{L}\sum_{k=0}^{L-1} r_{t+k+1}.$$

Under unichain ergodicity, as $L \to \infty$ these centered finite-horizon returns yield consistent (asymptotically unbiased) estimators of the differential/action-value $Q_{\mathrm{diff}}^{\pi_k}$ that appears in the average-reward policy-gradient identity. See standard references on average-reward policy gradient estimators (e.g., Konda & Tsitsiklis (2002), Marbach & Tsitsiklis (2001)) for the detailed derivation.

**3. Practical sampling schedule and asymptotic unbiasedness.** To use these estimators in stochastic approximation, one chooses a schedule of block lengths $L_n$ and batch sizes $N_n$ that grows so that the estimator bias due to finite $L_n$ is controlled relative to the step sizes $\alpha_n$ (standard SA condition: $\sum_n \alpha_n \varepsilon_n < \infty$ where $\varepsilon_n$ is the bias at iteration $n$). Concretely, if the chain mixes geometrically with mixing time $\tau_{\mathrm{mix}}$ (uniform in a neighbourhood of the iterates), choosing $L_n = C \log(1/\alpha_n)$ (or larger) plus sufficient burn-in ensures the bias $\varepsilon_n = O(\alpha_n)$ or better, which is summable when multiplied by $\alpha_n$. Under the unichain assumption this scheduling is feasible in principle; in practice one picks $L_n$ large enough (or uses regenerative sampling) to make bias negligible.

**4. Boundedness, exploration and gradient boundedness.** With softmax-linear parameterization and bounded features, $\nabla_\theta \log \pi_\theta$ is uniformly bounded on the compact parameter set $\Theta$. Entropy regularization keeps policies stochastic during learning and avoids vanishing exploration. Projection of $\theta$ onto $\Theta$ ensures iterates stay bounded.

**5. Satisfying the Unbiased Convergence Assumption.** Putting (1)–(4) together, the Monte Carlo average-reward gradient estimator (constructed from long blocks or regenerative cycles) yields asymptotically unbiased estimates of the average-reward policy gradient; equivalently one can produce (asymptotically) unbiased estimates $\hat{V}^{\pi_k}$ of the relevant value-like quantities required by $\mathbb{A}$. Hence the Unbiased Convergence Assumption (Assumption B.1) is satisfied in the asymptotic sense required for SA convergence.

**6. Apply Theorem 1.** By Assumption B.1, any algorithm that converges under Frame Stacking with unbiased value estimates will converge to the optimal $k$-order policy. Therefore, with the asymptotically unbiased average-reward estimates constructed above and $k \geq \kappa$, $\mathbb{A}$ converges to an Adaptive Stacking policy $\pi_k^*$ that maximizes the long-run average reward $\rho(\pi)$. This completes the proof. ∎

**Remarks**

- The episodic corollary is straightforward because finite-horizon Monte Carlo returns are exactly unbiased. The linear softmax parameterization + compactness + entropy + projection assumptions are standard to ensure bounded gradients and persistent exploration, and to make the SA theory applicable.

- The unichain average-reward corollary requires the extra ergodicity/unichain assumption to justify long-run averages as consistent estimators of $\rho(\pi)$ (or regenerative sampling to provide unbiased cycle averages). It also requires an explicit sampling schedule (block lengths $L_n$ growing appropriately) so that estimator bias is negligible in the SA limit; I sketched the standard way to satisfy this condition (pick blocks scaling with mixing time / $\log(1/\alpha_n)$).

- In both corollaries the key bridge to Theorem 1 is verifying that the practical Monte Carlo estimators produce (asymptotically) unbiased estimates of the target value quantities. Once that is established, the theorem implies convergence under Adaptive Stacking for $k \geq \kappa$.

### B.6 Proof for Theorem 2

We now prove that if two policies have an ordering over value functions in the induced memory POMDP $\mathcal{M}_k$, and the memory representation is value-consistent, then the same ordering holds over the original latent histories.

**Assumption B.2 (Value-Consistency)** *Let $\pi_k$ be an Adaptive Stacking policy over memory states $s_t \in \mathcal{S}_k$. We say the memory representation is* value-consistent *with respect to $\pi_k$ if for any $s_t \in \mathcal{S}_k$ and any two latent histories $x_{t:t-k^*}, x'_{t:t-k^*}$ such that*

$$\Pr(x_{t:t-k^*} \mid s_t, \pi_k) > 0 \quad \text{and} \quad \Pr(x'_{t:t-k^*} \mid s_t, \pi_k) > 0,$$

*it holds that:*

$$V^{\pi_k}(x_{t:t-k^*}) = V^{\pi_k}(x'_{t:t-k^*}).$$

**Theorem 4 (Partial-order Preserving)** *Let $k \in \mathbb{N}$ and let $\pi_k^1, \pi_k^2$ be two policies under Adaptive Stacking such that for all memory states $s_t \in \mathcal{S}_k$:*

$$V_k^{\pi_k^1}(s_t) \leq V_k^{\pi_k^2}(s_t).$$

*If both policies induce value-consistent memory representations (Assumption B.2), then for all latent histories $x_{t:t-k^*}$:*

$$V^{\pi_k^1}(x_{t:t-k^*}) \leq V^{\pi_k^2}(x_{t:t-k^*}).$$

**Proof** By Equation 2, the expected return under $\pi_k^1$ in the induced memory process is:

$$V_k^{\pi_k^i}(s_t) = \sum_{x_{t:t-k^*}} \Pr(x_{t:t-k^*} \mid s_t, \pi_k^i) V^{\pi_k^i}(x_{t:t-k^*}).$$

Under Assumption B.2, for each $i \in \{1, 2\}$, all latent histories $x_{t:t-k^*}$ consistent with a memory state $s_t$ have equal value:

$$V^{\pi_k^i}(x_{t:t-k^*}) = c_i(s_t), \quad \text{a constant.}$$

Hence, the above expectation reduces to:

$$V_k^{\pi_k^i}(s_t) = c_i(s_t).$$

Therefore, the ordering assumption implies:

$$c_1(s_t) = V^{\pi_k^1}(x_{t:t-k^*}) \leq V^{\pi_k^2}(x_{t:t-k^*}) = c_2(s_t),$$

for all $x_{t:t-k^*}$ such that $\Pr(x_{t:t-k^*} \mid s_t, \pi_k^i) > 0$.

Thus, the partial ordering $V_k^{\pi_k^1}(s_t) \leq V_k^{\pi_k^2}(s_t)$ implies:

$$V^{\pi_k^1}(x_{t:t-k^*}) \leq V^{\pi_k^2}(x_{t:t-k^*}) \quad \text{for all } x_{t:t-k^*}. \qquad \blacksquare$$

### B.7 Proof for Theorem 5

We now prove that Temporal Difference (TD) learning converges to the optimal policy under Adaptive Stacking, provided that $k \geq \kappa$ and the memory representation is value-consistent.

**Theorem 5** *Let $k \geq \kappa$, and suppose Q-learning under standard learning assumptions (Robbins & Monro, 1951) is applied to the induced decision process $\mathcal{M}_k$ under a fixed exploratory policy that ensures persistent exploration. If policies in $\mathcal{M}_k$ are value-consitent, then:*

1. *The Q-function $Q(s, a, i)$ converges with probability 1 to a fixed point $\hat{Q}(s, a, i)$.*

2. *The greedy policy with respect to $\hat{Q}$ is optimal. That is, $\pi_k^*(s_t) \in \arg\max_{(a,i)} \hat{Q}(s_t, a, i)$ achieves the optimal value $V^*(x_{t:t-k^*})$.*

**Proof** Since the agent operates over the induced process $\mathcal{M}_k$, its effective state is $s_t \in \mathcal{S}_k$. The Q-learning update rule is:

$$Q_{t+1}(s_t, a_t, i_t) \leftarrow Q_t(s_t, a_t, i_t) + \alpha_t \left[ r_{t+1} + \gamma \max_{(a', i')} Q_t(s_{t+1}, a', i') - Q_t(s_t, a_t, i_t) \right],$$

where $s_{t+1} = \text{push}(\text{pop}(s_t, i_t), x_{t+1})$ is the updated memory stack, and $\alpha_t$ is a learning rate satisfying the standard conditions:

$$\sum_t \alpha_t = \infty, \quad \sum_t \alpha_t^2 < \infty.$$

Under the assumption that all $(s, a, i)$ tuples are visited infinitely often, and rewards are bounded, Theorem 2 of Singh et al. (1994) guarantees that $Q(s, a, i)$ converges to the fixed point $\hat{Q}(s, a, i)$.

Since $k \geq \kappa$, by definition of $\kappa$, there exists a policy $\pi_k^*$ such that for all latent histories $x_{t:t-k^*}$:

$$V^{\pi_k^*}(x_{t:t-k^*}) = V^*(x_{t:t-k^*}).$$

Because memory length $k$ is sufficient to represent all task-relevant distinctions (the disambiguation required for value prediction), we know from Theorem 2 that under the value-consistency assumption, the policy $\pi_k^*$ that is greedy with respect to $\hat{Q}$ in the induced process $\mathcal{M}_k$ will also be optimal in the underlying latent space:

$$\pi_k^*(s_t) \in \arg\max_{(a,i)} \hat{Q}(s_t, a, i) \quad \Rightarrow \quad V^{\pi_k^*}(x_{t:t-k^*}) = V^*(x_{t:t-k^*}) \quad \forall t.$$

Thus, Q-learning in the Adaptive Stacking process not only converges, but yields an optimal policy over the original environment when $k \geq \kappa$. ∎

## C  VALUE-CONSISTENCY ASSUMPTION IN POPULAR BENCHMARKS

In this section, we analyze common RL benchmarks to determine when our Value-Consistency (VC) Assumption 4.1 holds. Recall that this assumption requires that all full histories $x_{t:t-k^*}$ mapping to the same agent memory state $s_t$ under policy $\pi_k$ must share the same expected return $V^{\pi_k}(x_{t:t-k^*})$. This often holds in goal-reaching or sparse-reward settings, but can be violated in tasks with dense or history-sensitive rewards (such as unobservable reward machines).

Table 2 summarizes our analysis, and we provide justification for each task below.

**T-Maze (Classic) (Bakker, 2001):** The agent observes a goal cue at the start, traverses a corridor, and makes a binary decision at a junction. Here, $k^* = T = 70$, since full observability only comes from the initial and final steps. However, $\kappa = 2$ suffices: the initial cue and position are enough to act optimally. VC holds since all consistent histories that lead to the same stack (for example, seeing "green") yield the same value.

**TMaze Long (Beck et al., 2020):** Structurally identical to Classic T-Maze but with longer horizon $T = 100$. Again, $k^* = T$, $\kappa = 2$, and VC holds for the same reason.

**Passive Visual Match (Hung et al., 2018):** The goal color is observed passively at the start. The main reward depends only on whether the agent chooses the matching color at the end (plus intermediate rewards from collecting apples). $k^* = T = 600$, but $\kappa = T$. VC holds since the goal cue and nearby apples fully determines return.

**MiniGrid-Memory (Chevalier-Boisvert et al., 2018):** To plan efficiently, the agent must memorize a cue seen early and traverse a grid. The worst-case $k^* \leq 51$ and $\kappa = 2$ for simple cue-based planning. VC holds because position and cue suffice. In practice, if the position is not given, it can be estimated using path intergration.

| Task | T | $k^*$ | $\kappa$ | Assumption 4.1 (VC) Holds? |
|---|---|---|---|---|
| T-Maze (Classic) (Bakker, 2001) | 70 | T | 2 | ✓: Only goal cue matters |
| TMaze Long (Beck et al., 2020) | 100 | T | 2 | ✓: Only goal cue matters |
| Passive Visual Match (Hung et al., 2018) | 600 | $T$ | Long | ✓: Only goal cue and apples affects return |
| MiniGrid-Memory (Chevalier-Boisvert et al., 2018) | 1445 | $\leq 51$ | 2 | ✓: Position suffices after goal cue |
| Memory Length (Osband et al., 2020) | 100 | $T$ | 2 | ✓: Observation i.i.d. per timestep |
| Memory Maze (Pasukonis et al., 2022) | 4000 | Long | Long | ✓: Only current transition affect rewards |
| PsychLab (Fortunato et al., 2019) | 600 | $T$ | Long | ✓: Passive episodic recall |
| HeavenHell (Esslinger et al., 2022) | 20 | $T$ | 2 | ✓: Only goal cue matters |
| Memory Cards (Esslinger et al., 2022) | 50 | 2 | Long | ✓: Only current card pairs affect rewards |
| Ballet (Lampinen et al., 2021) | 1024 | $\geq 464$ | $\geq 464$ | ✓: Rewards unaffected by previous actions |
| Mortar Mayhem (Pleines et al., 2023) | 135 | $T$ | $T$ | ✓: Rewards unaffected by previous actions |
| Numpad (Parisotto et al., 2020) | 500 | $T$ | $N^2$ | ✓: Rewards unaffected by previous actions |
| Reacher-POMDP (Ni et al., 2021) | 50 | Long | 2 | ✓: Only goal cue matters |
| Repeat First (Morad et al., 2023) | 832 | 2 | 2 | ✓: Only previous optimal action matters |
| Autoencode (Morad et al., 2023) | 312 | 312 | 156 | ✓: Rewards unaffected by previous actions |
| POPGym CartPole (Morad et al., 2023) | 600 | 2 | 2 | ✓: Only previous observation matters |
| Reward Machines (Icarte et al., 2022) | 1000 | $T$ | $T$ | ✗: Rewards affected by previous actions |
| **Passive T-Maze (Episodic) (Ours)** | $10^6$ | T | 2 | ✓: Only goal cue matters |
| **Passive T-Maze (Continual) (Ours)** | $10^6$ | T | 2 | ✓: Only goal cue matters |
| **Active T-Maze (Episodic) (Ours)** | 100 | $\infty$ | 2 | ✓: Only goal cue matters |
| **Active T-Maze (Continual) (Ours)** | $10^6$ | $\infty$ | 2 | ✓: Only goal cue matters |
| **XorMaze (Ours)** | 100 | $\infty$ | 3 | ✓: Only goal cues matters |
| **Rubik's Cube (Ours)** | 100 | $\infty$ | Long | ✓: Episodic, single goal, sparse rewards |
| **FetchReach (Ours)** | $10^6$ | $\infty$ | 4 | ✗: Continual, multiple goals, dense rewards |

Table 2: Evaluation of Value-Consistency (VC) assumption across popular RL benchmark tasks. T is the maximum episode horizon or total training steps (for continual settings). $k^*$ is the memory length required to make the environment Markov; Long means a relatively large proportion of the episode must be remembered to make optimal value predictions. $\kappa$ is the minimal memory length required to achieve optimal return. Finally, VC Holds states whether Value-Consistency is satisfied.

**Memory Length (Osband et al., 2020):** Observations are i.i.d. at each step. $k^* = T = 100$, but optimality requires only $\kappa = 2$. VC holds since memory state compresses all relevant statistics.

**Memory Maze (Pasukonis et al., 2022):** Agent must collect colored balls in order. The reward depends only on the current pickup. $k^* = $ Long, $\kappa = $ Long. VC holds since rewards depend only on present state and target.

**HeavenHell (Esslinger et al., 2022):** The agent visits an oracle early in the episode which defines the correct terminal target. The memory requirement is $k^* = T = 20$, but once the cue is retained, $\kappa = 2$ ensures optimality. VC holds because different paths to the same cue yield identical future returns.

**Memory Cards (Esslinger et al., 2022):** The agent must match cards based on values seen in earlier steps. $k^* = 2$, but $\kappa = $ Long due to potential card permutations. VC holds because matching decisions are memory-conditional, not trajectory-sensitive.

**PsychLab (Fortunato et al., 2019):** Involves passive image memorization, typically from the beginning of an episode. $k^* = T = 600$, but memorizing the image is sufficient ($\kappa = $ Long). VC holds due to deterministic mapping from memory state to return.

**Ballet (Lampinen et al., 2021):** Agent observes sequences of dances and selects a correct dancer. Though the reward is episodic, the agent actions occur only post-observation. $k^* \geq 464$, $\kappa \geq 464$. VC holds because the same memory state determines the post-dance plan.

**Mortar Mayhem (Pleines et al., 2023):** Memorizing a command sequence and executing it. $k^* = T = 135$, $\kappa = T$. VC holds due to value depending solely on correctly recalling the command sequence.

**Numpad (Parisotto et al., 2020):** Agent must press a sequence of pads. $k^* = T = 500$, $\kappa = N^2$. VC holds: as long as the memory contains the correct order, the actual transition path is irrelevant.

**Reacher-POMDP (Yang & Nguyen, 2021):** The goal is revealed only at the first step, so $k^*$ must capture that first observation. Any policy only needs to retain that goal and act accordingly, so $\kappa = 2$ suffices. VC holds since differing histories that preserve the same goal state will yield the same value estimate.

**Repeat First (Morad et al., 2023):** Rewards depend on repeating the first action. $k^* = T$, but $\kappa = 2$ suffices by retaining just the first action. VC holds since the memory state is value-determining.

**Autoencode (Morad et al., 2023):** Agent reproduces observed sequence in reverse. $k^* = 311$, $\kappa = 156$ (half the trajectory). VC holds since the value depends only on accuracy of reproduction.

**POPGym CartPole (Morad et al., 2023) (Figure 19):** The Stateless VelocityOnlyCartPoleHard task from the POPGym Benchmark (Morad et al., 2023), which is representative of domains that are complex in dynamics and continuous but actually simple in memory requirements. This environment occludes the velocity component, but full observability is achieved after two steps (velocity and estimated position). Thus, $k^* = 2 = \kappa$. VC holds as only immediate transitions affect return. Hence is similar in memory requirements to the TMaze task with $L = 0$.

### C.1 UNOBSERVABLE REWARD MACHINES COUNTER-EXAMPLE

While the Value-Consistency Assumption holds in many benchmark settings (Table 2), it fails in environments where the true reward function depends not just on environment observations, but on dynamic latent trajectory properties such as event sequences which change based on the agent policy. This is most notably the case in environments that use *reward machines* (Icarte et al., 2018; Vaezipoor et al., 2021; Icarte et al., 2022; Tasse et al., 2024) – finite state automata over temporal logic formulae that determine rewards or sub-goals based on the sequence of states visited.

For example, consider the task *"Deliver coffee to the office without breaking decorations"* in a the office grid-world environment (Icarte et al., 2022). The task is encoded as a reward machine over three atomic propositions: $p_{\text{coffee}}$ (the agent visits the coffee location), $p_{\text{office}}$ (the agent visits the office location), $p_{\text{decor}}$ (the agent steps on any decoration tile). The agent starts at some initial location and must: visit the coffee location *first*, then visit the office location, without ever triggering $p_{\text{decor}}$. A reward of $+1$ is given only if the full trajectory satisfies the temporal formula:

$$(F(p_{\text{coffee}} \wedge X(F p_{\text{office}}))) \wedge (G \neg p_{\text{decor}}).$$

**Why VC Fails.** In the native environment, the agent's observations are just its $(x, y)$ location. There is no explicit record of whether the coffee has been visited, or if a decoration tile was stepped on. Consequently, two different trajectories can lead to the same agent observation $s_t = (x, y)$ and memory stack $s_t = [x_{i_1}, \ldots, x_{i_k}]$. Yet these trajectories may differ in *reward-relevant history*, for example, one might have stepped on a decoration earlier while another didn't. Since the reward for reaching the office depends on whether the coffee was collected *and* no decorations were touched in the past, which is unobservable from $s_t$ alone, the condition:

$$V^{\pi_k}(x_{t:t-k^*}) = V^{\pi_k}(x'_{t:t-k^*})$$

does *not* hold for histories $x_{t:t-k^*}, x'_{t:t-k^*}$ that lead to the same agent state $s_t$. Therefore, Assumption 4.1 is violated. Other common temporal logic tasks that violate VC include:

- *"Collect key A before key B, then go to door"*: reward depends on the *order* of events, not the final state.

- *"Don't revisit any state"*: any policy that loops violates the reward constraint, but the current memory may not capture visit counts.

- *"Eventually visit both goal zones A and B, but never touch lava"*: again, whether lava was touched can be lost under memory compression.

The VC assumption breaks because environment-level memory states $s_t$ are not sufficient statistics for the reward machine's state. The true reward depends on a latent automaton state that evolves with trajectory-dependent triggers. This is equivalent to acting in a *cross-product MDP* over $(x, y) \times u$, where $u$ is the internal automaton state.

**Can the Failure Be Benign?** Despite the theoretical violation, practical agents can still learn to behave correctly using Adaptive Stacking when: The reward machine state can be inferred from a small set of key observations; The agent learns to preserve these key triggers (for example, the first visit to coffee or decoration tiles); The failure to preserve value consistency leads to pessimistic value estimates, but not incorrect action selection.

Hence, reward machine tasks represent a natural and important class of environments where the VC assumption breaks due to latent trajectory-dependent semantics. This distinction is useful for future work aiming to blend Adaptive Stacking with automaton inference, or for delineating the boundaries of where value-consistent abstraction is theoretically sound.

## D EXPERIMENTAL DETAILS

All agents were implemented using PyTorch (and Gymnasium for the environments), and trained for $10^6$ steps. Tabular Q-learning used in-memory arrays, and PPO used Stable-Baselines3. Finally, all experiments were ran on CPU only Linux servers.

### D.1 EXPERIMENTAL DOMAINS

**Our Passive and Active T-Maze (in Episodic and Continual settings) (Figures 3a,3b):** There are only 4 observations here, corresponding to the color of the grid cell the agent is in: $red$ ▧ for $goal_1$, $green$ ▧ for $goal_2$, $blue$ ▧ for the maze junction, and $grey$ ▧ for the maze corridor. The given goal (red or green sampled uniformly) is only shown at the tail end of the maze, and the corridor has length $L$. The agent is represented by the black dot and has four cardinal actions for navigation.

1. **Passive-TMaze**. The agent starts at the tail end of the maze. It then takes one step to the right at every time step regardless of it's action, until the junction location where the top and right actions achieve $goal_1$ while the down and left actions achieve $goal_2$.

2. **Active-TMaze**. The agent starts one step to the right of the tail end of the maze. It then moves in the cardinal direction corresponding to its action at every time step, or stays still if the action hits the maze walls (for example taking the up or down actions in the corridor and the right action at the junction).

In all our TMaze variants, the goal cue is shown at the tail of the maze and the return depends only on whether the goal is reached. In the continual setting, the memory state is unchanged after the agent reaches a goal (unlike the episodic setting where the memory is reset). Even in the training loop, there is no oracle done signal and the agent is automatically placed back into the starting position once it reaches a goal. Hence the agent here needs to learn to replace the goal cue it previously memorized. Thus $\kappa = 2$ and $k^*$ matches the maze traversal length ($L + 2$), which is $10^6$ in our longest evaluation (Figure 1 right). For the **Active-TMaze**, $k^* = \infty$ since the agent can stay arbitrarily long in the grey corridor. VC holds even under stochastic start states or corridor lengths.

**Our XorMaze (Figure 31):** The **XorMaze** is representative of simple domains with complex memory requirements. It is similar to the TMaze (with the same observations) but has two corridors: one vertical and one horizontal. These corridors are crossed in the middle (forming the + symbol), and the agent starts at their intersection (also the junction location). The horizontal and vertical corners are a single step from the center. At the corners of the horizontal corridor, there are goal cues randomly choosen between red and green. In the vertical axis, we have the red goal at the top ($goal_1$) and a green goal at the bottom ($goal_2$). The task is to observe the values in the horizontal axis, and the agent has to go to the cell in the vertical axis that is the result of an XOR. For example, if the horizontal values are red and green it should go to the top location ($goal_1$), but if they are red and red (or green and green) it should go to the bottom location ($goal_2$). Hence $\kappa = 3$ and $k^* = \infty$. VC holds here similarly to the TMaze.

**Our Rubik's Cube (Figure 3c):** The 2x2 Rubik's cube task is representative of complex domains with complex memory requirements. The agent here only sees one of the six cube faces at a time. Hence the state is a $24$ dimensional vector and the agent only observes a $4$ dimensional slice of it. The agent has 16 total actions: 12 default actions for rotating the cube, plus 4 additional actions for 90 degrees rotations of the camera across each 3D axis (to see an adjacent face). The goal of the agent is to start from a randomly scrambled cube and reach the solved state (the unique correct colour for each face). Hence $\kappa \geq 6$ and $k^* = \infty$ since the transitions depend on an arbitrarily long history of past actions. VC holds here since the environment is deterministic, goal reaching with sparse rewards (one for reaching the goal and zero otherwise), and there's a single goal state.

**Our FetchReach (Figure 3d):** We modify the FetchReachDense task from the Gymnasium Robotics benchmark suite (Plappert et al., 2018) to be representative of continuous control domains that are complex in both dynamics and memory requirements (requiring both long-term and short-term memory). The default environment is continual, fully observable, has dense distance based rewards, and has a 3D goal position which changes randomly every 50 steps. The goal of the agent (Fetch robot) is to always move it's end effectors to the 3D goal position and stay there forever (until the goal position changes again). Precisely:

1. We occlude the velocity component at all time steps (similarly to **POPGym Cartpole**)

2. We also occlude that goal position after the first 2 steps after each change. So this *goal cue* is only observable for the first 2 steps after each change (similarly to the **TMaze** tasks).

Hence the agent must simultaneously: remember the goal cue for all steps until the goal position changes (demonstrating long-term memory), and for each of those steps it must remember the joint positions at the previous step in order to estimate the occluded joint velocities (demonstrating short-term memory). Thus, $k^* = \infty$ and $\kappa = 4$. VC does not hold since the environment is not only continual, but also has random goals (and random start states) and dense rewards.

## D.2 RECORDED METRICS

Every 100 environment steps we log:

1. *Return*: cumulative discounted reward.

2. *Reward regret*: $V^*(x_{t:t-k^*}) - V^\pi(x_{t:t-k^*})$.

3. *Memory regret*: fraction of steps where the goal cue is absent from the memory stack.

4. *Active memory regret*: steps when the goal cue is observed but not stored.

5. *Passive memory regret*: steps when the goal cue is in memory but then discarded.

Plots report mean and 1 standard deviation over $N_{rs}$ independent seeds.

## D.3 TABULAR Q-LEARNING (CONTINUAL AND EPISODIC)

We run a standard $\varepsilon$-greedy tabular Q-learning agent in both Passive and Active T-Maze, under continual or episodic modes. Hyperparameters are listed in Table 3.

Table 3: Q-Learning hyperparameters

| Parameter | Value |
|---|---|
| Discount factor $\gamma$ | 0.99 |
| Learning rate $\alpha$ | 0.1 |
| Exploration $\varepsilon$ | fixed 0.01 |
| Total steps | $10^6$ |
| Memory configurations | FS($\kappa$), FS($k^*$), AS($\kappa$) |
| Random seeds $N_{rs}$ | 20 |
| Logging frequency | every 100 steps |

## D.4 PROXIMAL POLICY OPTIMIZATION (EPISODIC AND CONTINUAL)

We evaluate PPO with MLP, CNN, LSTM and Transformer policies in both Passive and Active T-Maze, under episodic or continual modes. Table 4 details the optimizer settings.

Table 4: PPO hyperparameters

| Parameter | Value |
|---|---|
| Total timesteps | $10^6$ |
| Discount factor $\gamma$ | 0.99 |
| GAE $\lambda$ | 0.95 |
| Rollout length $n\_steps$ | 128 |
| Minibatch size | 128 |
| Epochs per update | 10 |
| Learning rate | $3 \times 10^{-4}$ |
| Clip range | 0.2 |
| Entropy coefficient | 0.0 (default) |
| Value loss coefficient | 0.5 (default) |
| Random seeds $N_{rs}$ | 10 |
| Logging frequency | every 100 steps |

Each policy network receives the $k$-length memory stack as input and outputs two probability distributions: one over environment actions and one over memory-slot indices. The final policy is obtained by sampling each head independently.

**MLP**

1. *Input*: one-hot encoding of each of the $k$ observations, concatenated into a vector.

2. *Hidden layers*: three fully-connected layers of 128 units.

3. *Outputs*:

    (a) **Env-action head**: linear layer to $|\mathcal{A}|$ logits.

    (b) **Memory-action head**: linear layer to $k$ logits.

**LSTM**

1. *Input embedding*: each observation is embedded into a 128-dim vector.

2. *Sequence model*: single-layer LSTM with 128 hidden units processes the $k$ embeddings.

3. *Readout*: final hidden state of size 128.

4. *Outputs*: two linear heads (as above) mapping the 128-dim readout to action logits.

**Transformer**

1. *Input embedding*: each observation is embedded into 128-dim, plus learned positional embeddings for positions $1, \ldots, k$.

2. *Transformer decoder stack*: two layers, model dimension 128, 4 attention heads, feed-forward dimension 256.

3. *Readout*: the representation at the final time step.

4. *Outputs*: two linear heads mapping the 128-dim readout to environment logits and memory-slot logits.

## E SUPPLEMENTARY EXPERIMENTS

### E.1 LEARNING AREA UNDER THE CURVE BAR PLOTS

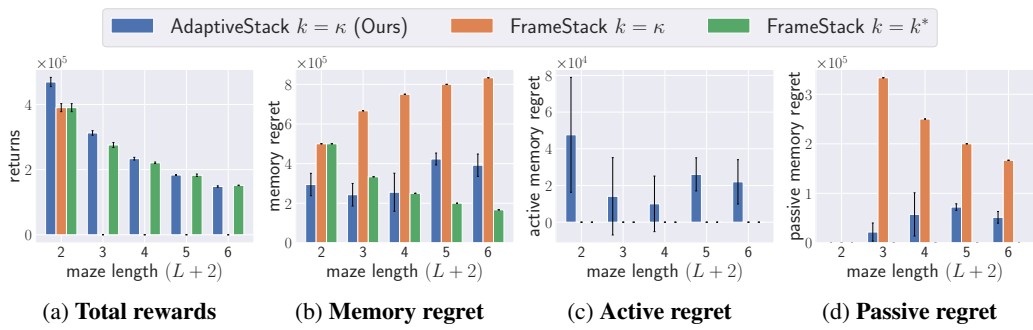

(a) **Total rewards**  (b) **Memory regret**  (c) **Active regret**  (d) **Passive regret**

Figure 8: Episodic **Passive-TMaze** (with corridor lengths per episode fixed to max length) with PPO and MLP policy ($N_{rs} = 10$).

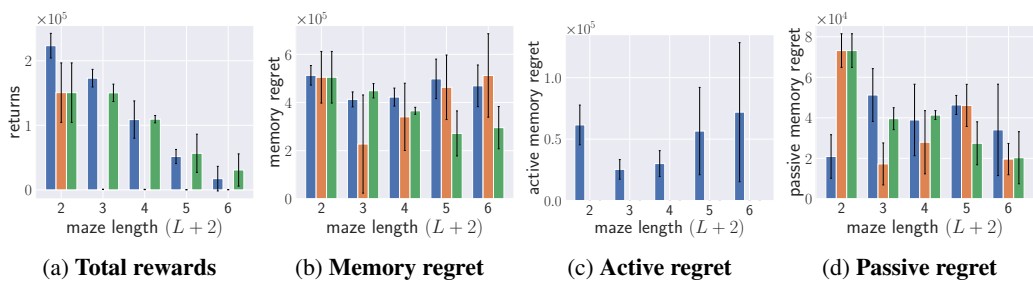

(a) **Total rewards**  (b) **Memory regret**  (c) **Active regret**  (d) **Passive regret**

Figure 9: Episodic **Active-TMaze** (with corridor lengths per episode fixed to max length) with PPO and MLP policy ($N_{rs} = 10$).

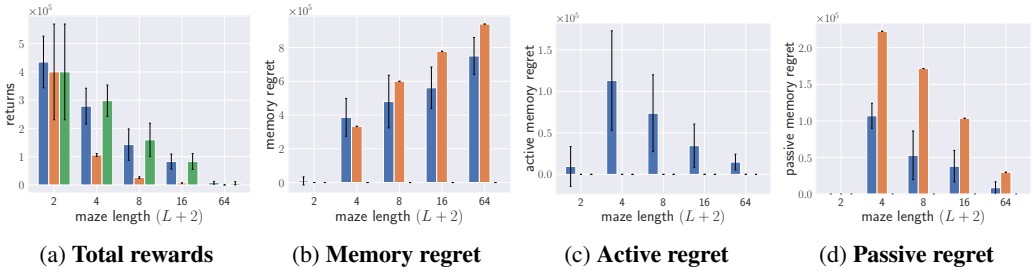

(a) **Total rewards**  (b) **Memory regret**  (c) **Active regret**  (d) **Passive regret**

Figure 10: Episodic **Passive-TMaze** with PPO and LSTM policy ($N_{rs} = 10$).

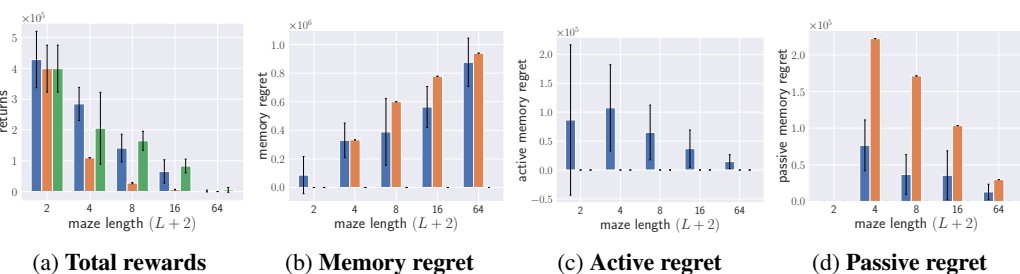

(a) **Total rewards**  (b) **Memory regret**  (c) **Active regret**  (d) **Passive regret**

Figure 11: Episodic **Passive-TMaze** with PPO and Transformer policy ($N_{rs} = 10$).).

## E.2 LEARNING CURVES

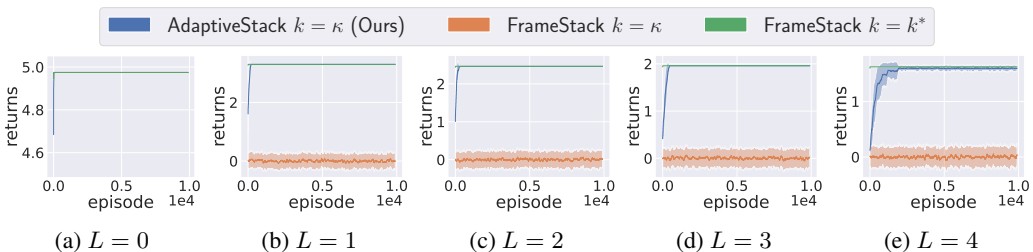

(a) $L = 0$     (b) $L = 1$     (c) $L = 2$     (d) $L = 3$     (e) $L = 4$

Figure 12: Returns in Continual **Passive-TMaze** with Q-learning ($N_{rs} = 20$) for varying maze lengths ($L + 2$). AS quickly matches the oracle FS($k^*$) in returns, while outperforming FS($\kappa$).

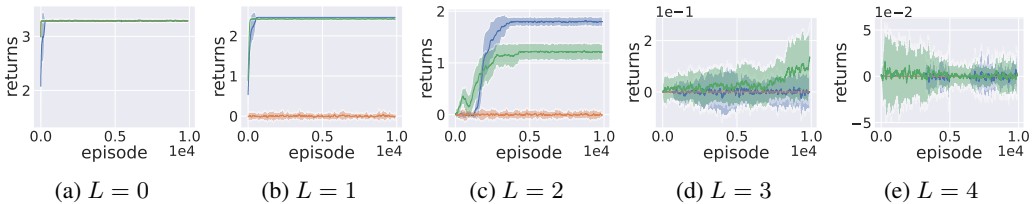

(a) $L = 0$     (b) $L = 1$     (c) $L = 2$     (d) $L = 3$     (e) $L = 4$

Figure 13: Returns in Continual **Active-TMaze** with Q-learning ($N_{rs} = 20$) for varying maze lengths ($L + 2$). AS quickly matches or exceeds the oracle FS($k^*$) in returns, while outperforming FS($\kappa$).

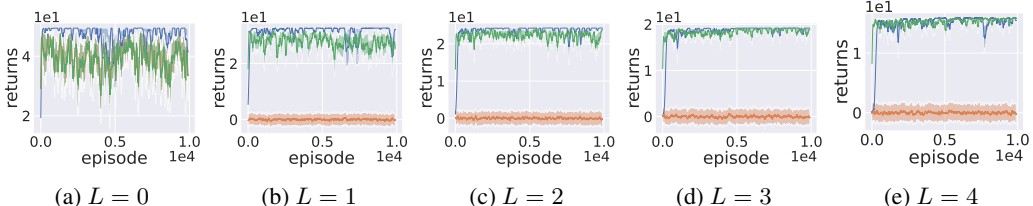

(a) $L = 0$     (b) $L = 1$     (c) $L = 2$     (d) $L = 3$     (e) $L = 4$

Figure 14: Returns in Episodic **Passive-TMaze** using PPO with an MLP ($N_{rs} = 10$) for varying maze lengths ($L + 2$). The corridor lengths per episode are fixed to the max length. AS quickly matches the oracle FS($k^*$) in returns, while outperforming FS($\kappa$, orange) especially for long-term dependencies.

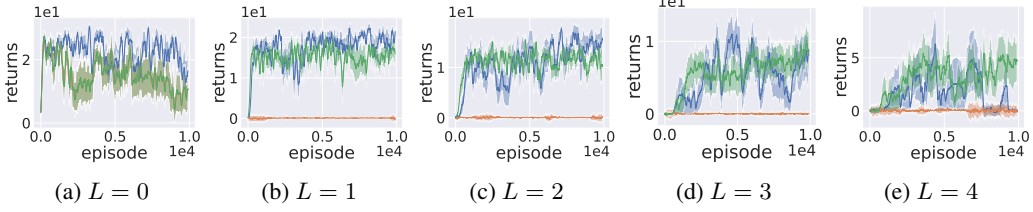

(a) $L = 0$     (b) $L = 1$     (c) $L = 2$     (d) $L = 3$     (e) $L = 4$

Figure 15: Returns in Episodic **Active-TMaze** with PPO with an MLP ($N_{rs} = 10$) for varying maze lengths ($L + 2$). The corridor lengths per episode are fixed to the max length. AS quickly matches the oracle FS($k^*$) in returns, while outperforming FS($\kappa$, orange).

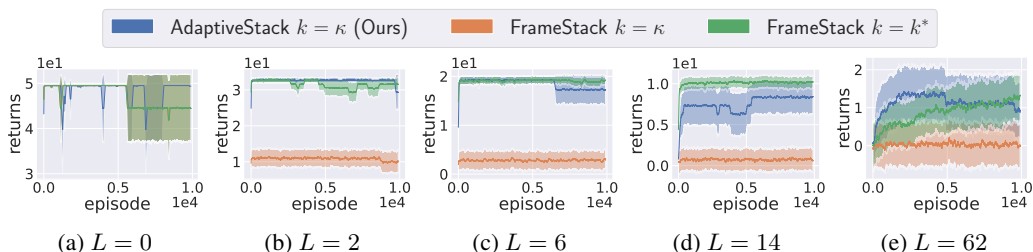

Figure 16: Returns in Episodic **Passive-TMaze** using PPO with an MLP ($N_{rs} = 10$) for varying maze lengths ($L + 2$). AS is comparable to the oracle FS($k^*$) in returns, while outperforming FS($\kappa$).

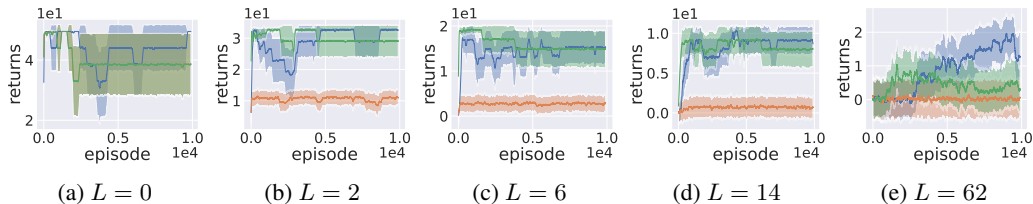

Figure 17: Returns in Episodic **Passive-TMaze** using PPO with an LSTM ($N_{rs} = 10$) for varying maze lengths ($L + 2$). AS is comparble to the oracle FS($k^*$) in returns, while outperforming FS($\kappa$).

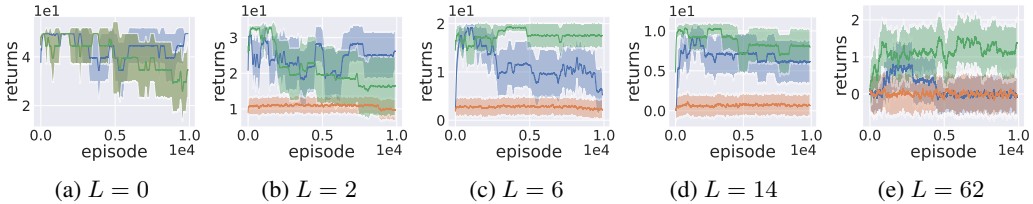

Figure 18: Returns in Episodic **Passive-TMaze** with PPO with an Transformer ($N_{rs} = 10$) for varying maze lengths ($L + 2$). AS matches the oracle FS($k^*$) in returns for smaller mazes but struggles to learn for larger mazes, while still outperforming FS($\kappa$, orange).

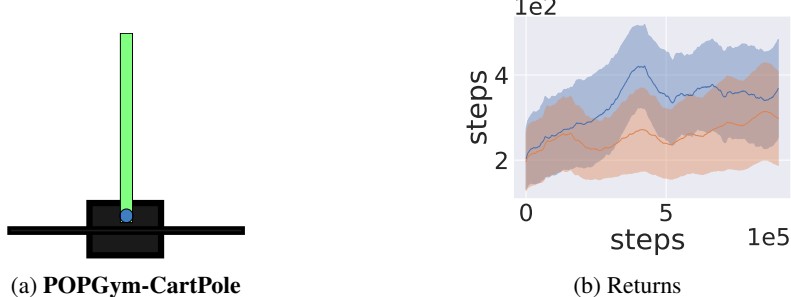

Figure 19: **POPGym-CartPole** results using PPO with an MLP ($N_{rs} = 10$). This is a partially observable environment were the agent needs to keep the pole upright for $T = 600$ steps per episode (with rewards of 1 per step). Only positional information is visible while velocity information is hidden (so only the previous observation must be remembered). Hence $k^* = \kappa = 2$. Consistent with our previous results, we observe that AS achieves performance close to FS($k^*$), even when the task is continuous and $k^* = \kappa$ (which is not our ideal setting where $\kappa \ll k^*$).

### E.3    EVALUATING LEARNED POLICIES

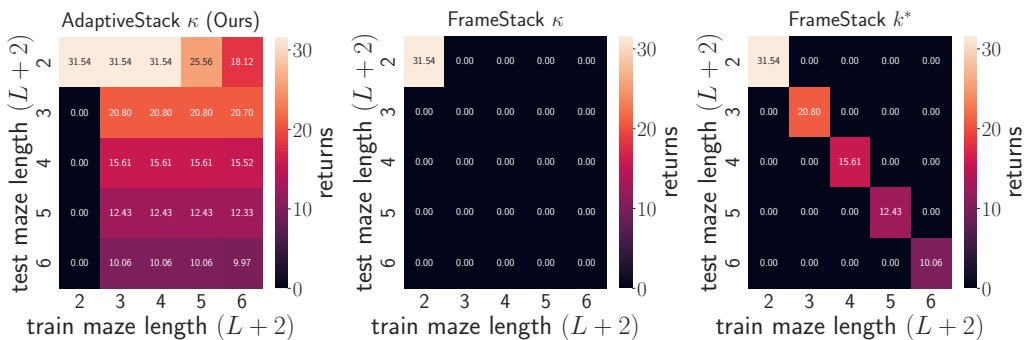

Figure 20: Evaluated returns in the continual **Passive-TMaze** with Q-learning ($N_{rs} = 20$). After training for 1 million steps, each agent is restarted at $s_0$ and tested for 100 additional steps in varying maze lengths. We show results averaged over the 20 training runs. We observe that AS leads to significantly better generalisation than FS($\kappa$) and even the oracle FS($k^*$), since it explicitly learns to remember only the observations that are relevant for decision-making.

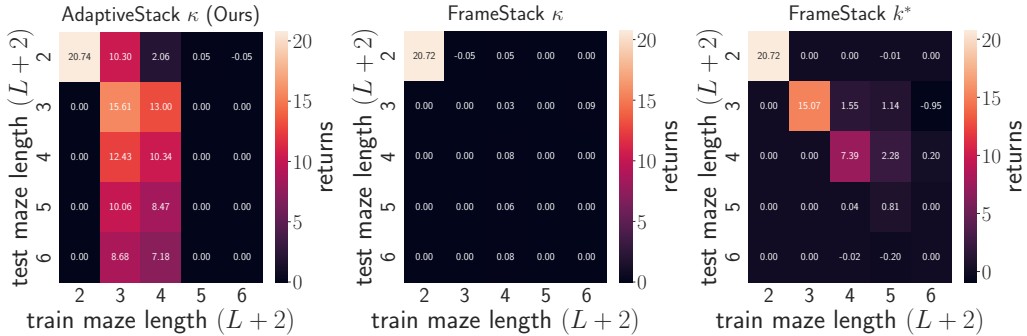

Figure 21: Evaluated returns in the continual **Active-TMaze** with Q-learning ($N_{rs} = 20$). We observe that when an agent using AS is able to successfully reach the correct goals during training, it has significantly better generalisation than even one using the oracle FS($k^*$). Note that policies with success rates in $[0\ 0.5]$ can have returns of 0 since the rewards are non-zero only for goal transitions.

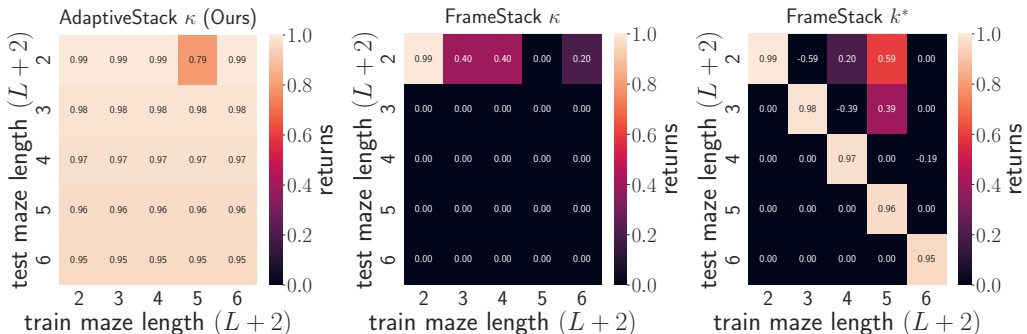

Figure 22: Evaluated returns in the episodic **Passive-TMaze** (with corridor lengths per episode fixed to max length) with PPO and MLP policy ($N_{rs} = 10$). We observe better results for AS and worse results for FS similarly to Figure 20.

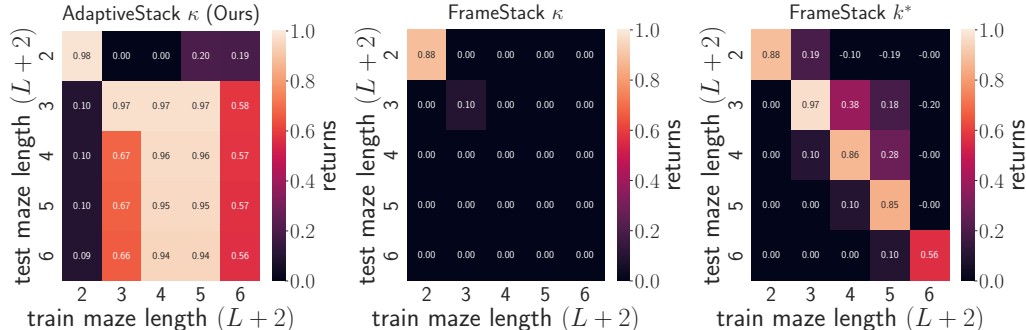

Figure 23: Evaluated returns in the episodic **Active-TMaze** (with corridor lengths per episode fixed to max length) with PPO and MLP policy ($N_{rs} = 10$). We observe similar results as Figures 21 and 22, except for maze lengths of 2. This difference is potentially because maze length 2 has no corridor (▢) observation, which makes it difficult to generalise the correct navigation actions to (and from) it.

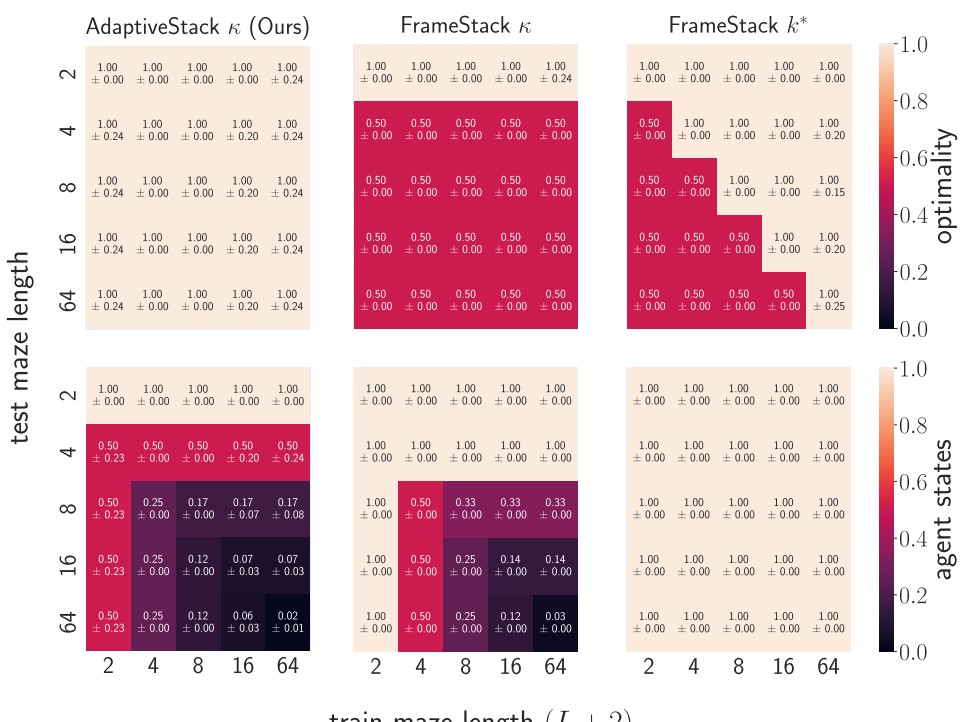

Figure 24: Generalisation and state abstraction in the episodic **Passive-TMaze** with PPO and MLP policy ($N_{rs} = 10$). After training for 1 million steps, each agent is tested for 2 additional episodes (for each goal color) in varying maze lengths (the corridor length in each testing episode is fixed to the max length). For each algorithm, we show the best test results across all 10 trained models, where $\pm$ is their standard deviations. The metrics are (a) **Optimality** (higher is better ↑): normalised difference between evaluated and optimal values (mean over 50 evaluation episodes per training run), (b) **Abstraction (agent states)** (lower is better ↓): normalised difference between the number of observed agent states (memory stacks) during evaluation from a trained policies using $k$ memory and optimal policies using $\kappa$ memory (mean over 50 evaluation episodes per training run). We observe that the random corridor lengths during training leads to consistently good in-distribution generalisation (upper-diagonal), even for FS, in contrast to Figure 22. This is potentially because FS mainly relies on the random corridor lengths during training to generalise. In contrast, AS still generally leads to better out-of-distribution generalisation (lower-diagonal) than even the oracle FS($k^*$) (as the goal cue gets pushed out of the memory stack). This may be because AS leads to far stronger state abstraction than FS, as the percentage of observed agent states (memory stacks) for AS drastically decreases (less observations added to the stack) as the maze length increases.

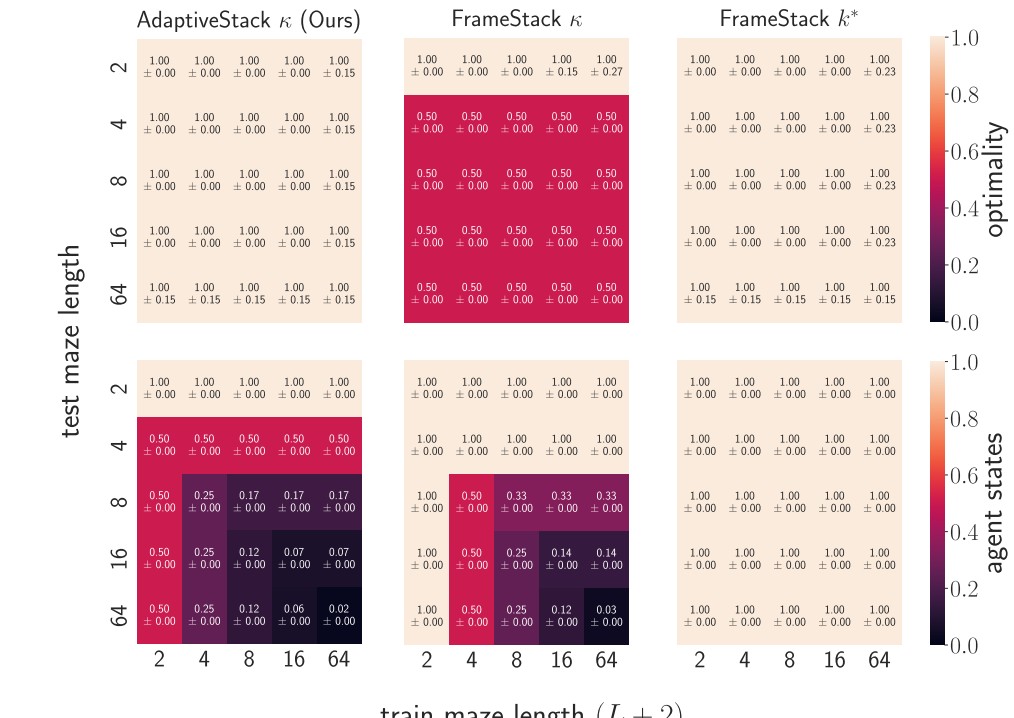

Figure 25: Generalisation and abstraction in the episodic **Passive-TMaze** with PPO and LSTM policy ($N_{rs} = 10$). We observe similar results for AS as the MLP case (Figure 24). However, we also notice that the oracle FS($k^*$) is now able to also generalise out of distribution thanks to the LSTM. This is most likely because it maintains a recurrent hidden state, but this requires backpropagation through time on the full episode rollout during training ($k = k^*$).

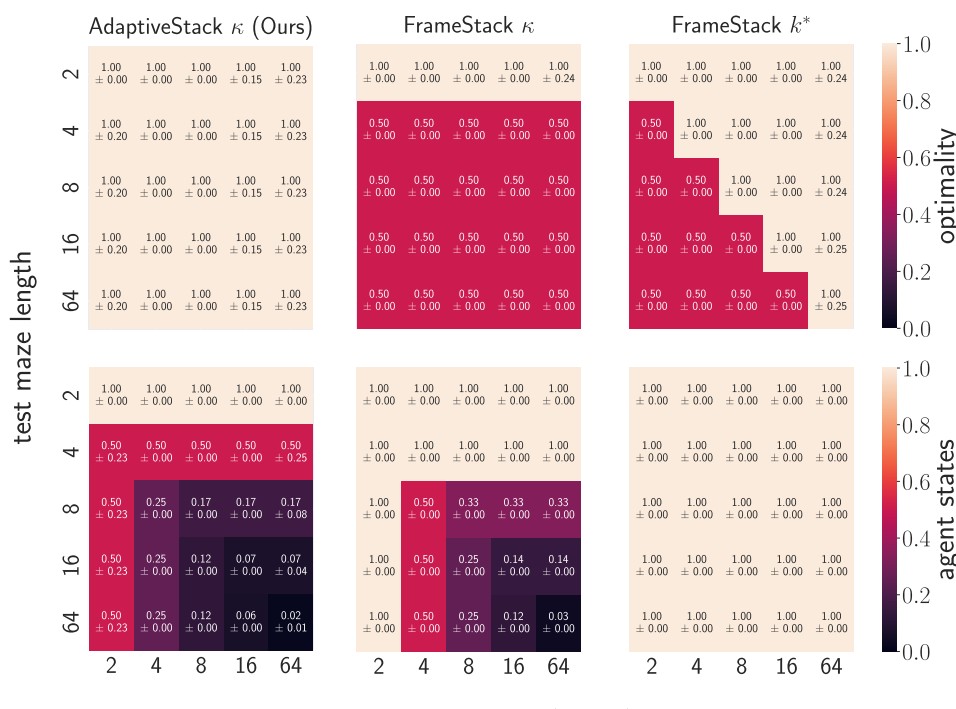

Figure 26: Generalisation and abstraction in the episodic **Passive-TMaze** with PPO and Transformer policy ($N_{rs} = 10$). We observe similar results for Adaptive Stacking as the MLP case (Figure 24).

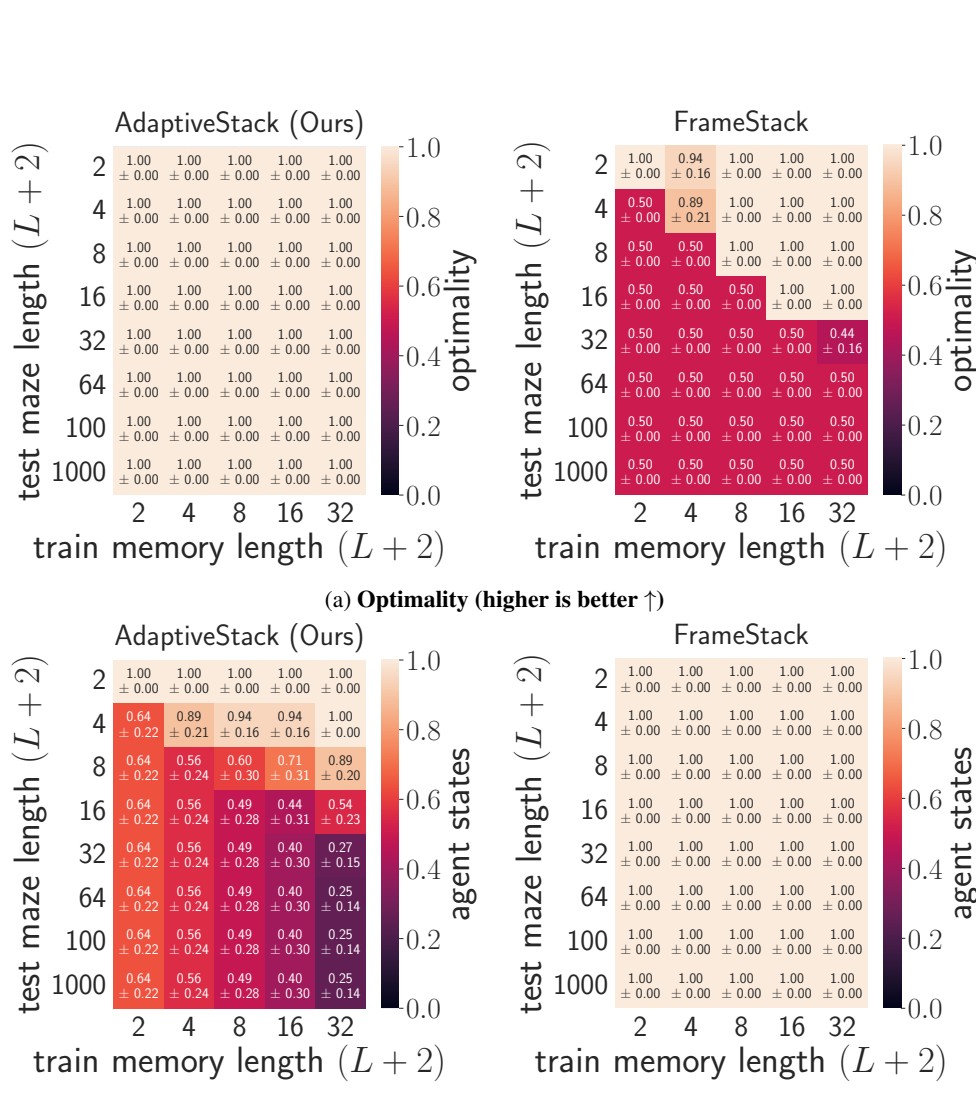

(a) **Optimality (higher is better ↑)**

(b) **Abstraction (lower is better ↓)**

Figure 27: Generalisation and state abstraction in the episodic **Passive-TMaze** with PPO training and a MLP policy ($N_{rs} = 10$). (a) **Optimality** (higher is better ↑): normalised difference between evaluated and optimal values (mean over 50 evaluation episodes per training run), (b) **Abstraction (agent states)** (lower is better ↓): normalised difference between the number of observed agent states (memory stacks) during evaluation from a trained policies using $k$ memory and optimal policies using $\kappa$ memory (mean over 50 evaluation episodes per training run). AS leads to far stronger state abstraction than FS, which explains it's much better generalisation to out of distribution test mazes.

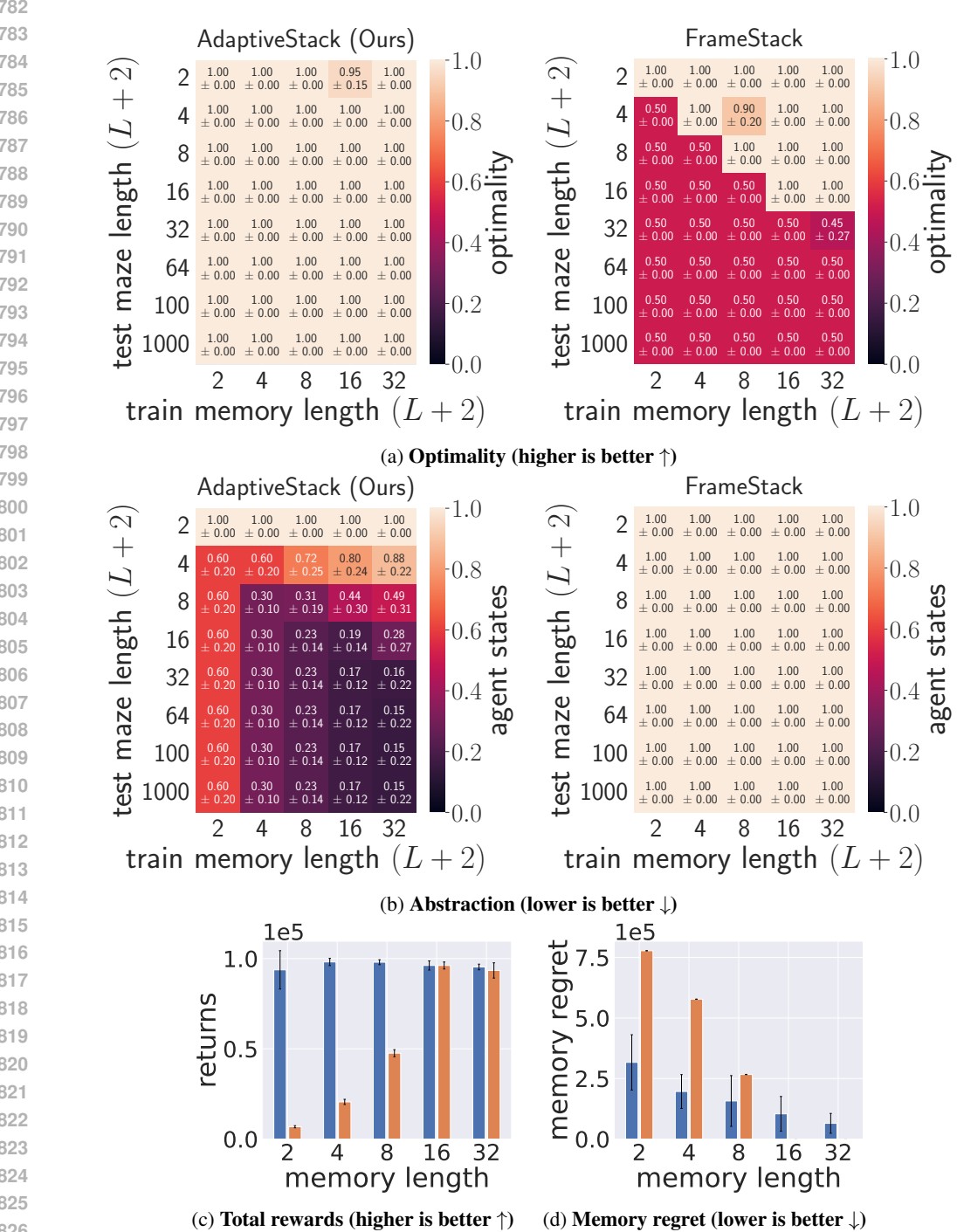

(a) **Optimality (higher is better ↑)**

(b) **Abstraction (lower is better ↓)**

(c) **Total rewards (higher is better ↑)**    (d) **Memory regret (lower is better ↓)**

Figure 28: Generalisation and state abstraction in the episodic **Passive-TMaze** with GRPO (Shao et al., 2024) training and a MLP policy ($N_{rs} = 10$). We implement a simplified version of GRPO in stable-baselines by simply removing the critic from PPO and using only MC estimates. (a) **Optimality** (higher is better ↑): normalised difference between evaluated and optimal values (mean over 50 evaluation episodes per training run), (b) **Abstraction (agent states)** (lower is better ↓): normalised difference between the number of observed agent states (memory stacks) during evaluation from a trained policies using $k$ memory and optimal policies using $\kappa$ memory (mean over 50 evaluation episodes per training run). AS leads to far stronger state abstraction than FS, which explains it's much better generalisation to out of distribution test mazes.

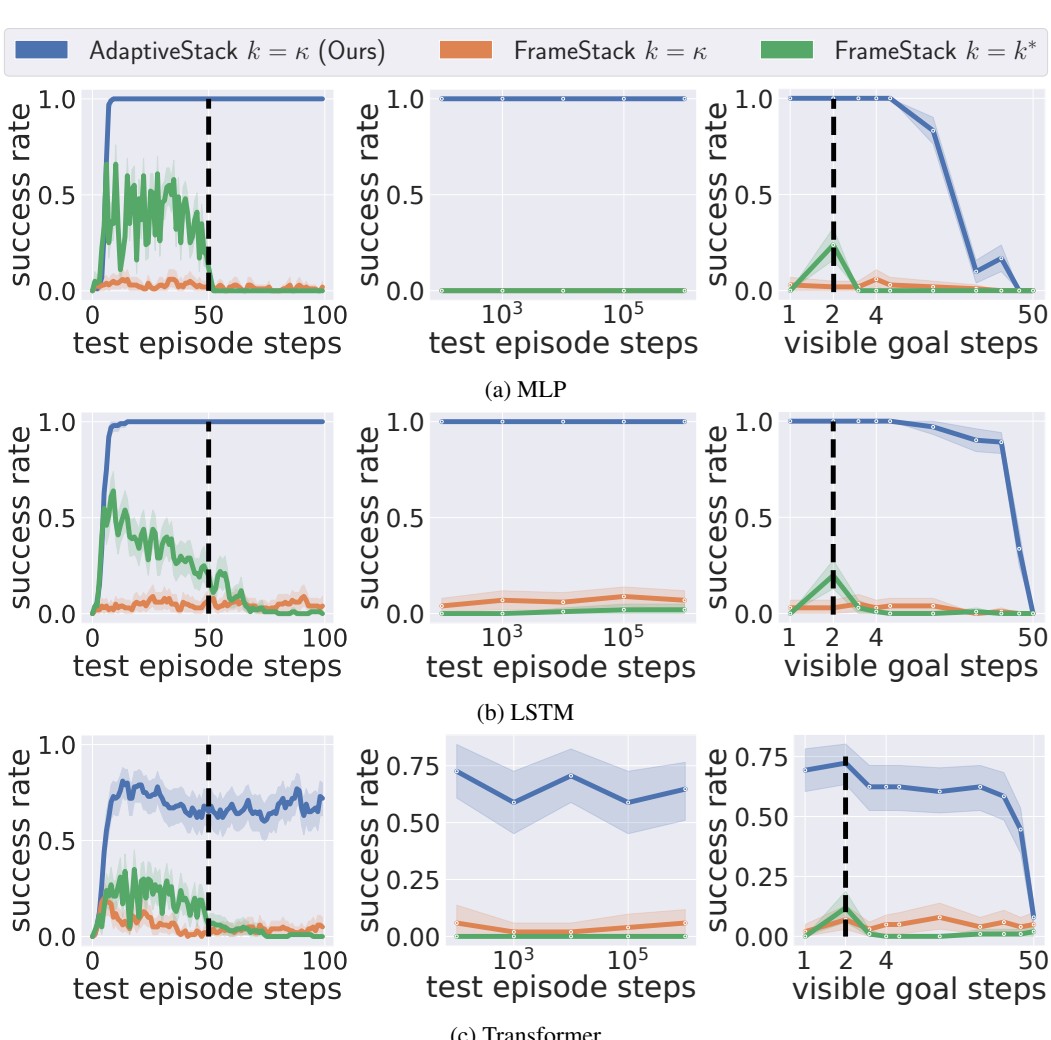

(a) MLP

(b) LSTM

(c) Transformer

Figure 29: Architecture agnostic generalisation results in **FetchReach** ($k^* = \infty$) with PPO ($N_{rs} = 100$). We show the evaluated success rates of the best models found across all training seeds. The black dashed lines on the left indicate the training: **(left)** the number of steps before the goal position changes during training, **(right)** the number of steps the goal is visible for at the beginning of each position change. Consistent with our other results, we observe that AS achieves significantly higher success rates than FS with the same memory stack length $k = 4$, and even higher than the oracle FS($k^*$), demonstrating the difficulty of learning from full histories when training steps are bounded ($10^6$). Interestingly, this generalisation of AS holds even under extreme distribution shifts (**middle** and **right**), demonstrating that it indeed learns to remember only observations relevant for reward maximisation, while discarding irrelevant observations. Finally, it is interesting that both the LSTM and transformer models trained with FS($k^*$) also struggle, quickly degrading to $0\%$ success rates similarly to a basic MLP. This is consistent with the results of Ni et al. (2023), which demonstrated that LSTMs struggle significantly with memory assignment (e.g. remembering the goal position) while transformers struggle significantly with credit assignment (e.g. learning a policy that maximises rewards, especially when short-term memory is also required). Given that we also observe the same behaviour when using an MLP, our results demonstrate that the use of FS is potentially the main driving factor behind performance difference, irrespective of model architecture.

### E.4 COMPARING WITH DEMIR (2023)

As discussed in our related works, Demir (2023) is a Frame Stacking approach and probably the most directly relevant work to our paper in their use of memory actions over a stack of observations. The memory action space they consider is significantly smaller than ours, but the memory architecture is biased in favor of always overwriting the oldest memory (it is still a FIFO memory like Frame Stacking):

1. *push:* which adds an element to the top of the stack. If the stack is full, then it removes the last observation (bottom of the stack) to free up space for the new observation (similarly to FS).

2. *skip:* which does nothing if the stack is full. Otherwise, it always pushes the new observation on top of the stack (similarly to FS). This means that when $k = k^*$ in environments like our **Passive-TMaze**, Demir (2023) is identical to regular FS.

Hence it is quite different than the action space we consider that allows for the observation to be skipped or to remove *any* available slot in memory to push the new observation. Our approach provides the agent with more choice over the maintenance of memory and thus has the potential to more efficiently utilize memory for problems where multiple observations with significant temporal distance must be considered. Our approach also learns a policy to access a memory by maximizing reward whereas Demir (2023) considers intrinsic motivation (IM) to store observations that are more novel. This bias is again intuitively helpful for many problems, but it would be easy to construct counter examples where it is detrimental (such as the famous "noisy TV" scenario). For example, Figures 30-31 shows the results comparing their performance to regular FS and AS in our continual TMazes and **XorMaze**. We observe that their approach struggles especially when using IM, which is likely due to the IM incentivising the wrong behaviours. Hence, we focus our analysis in this paper to regular Frame Stacking, since it has theoretical guarantees, is agnostic to the sequence model and RL algorithm, and is the approach most commonly used in RL with function approximation (Ni et al., 2021).

Finally, it is also important to note that while Demir (2023) compares to RNNs with function approximation, their memory management approach is instantiated as a purely tabular method using Sarsa($\lambda$) and compared against A2C and PPO. In contrast, our work extends stack management to function approximation (irrespective of architecture or RL algorithm) and perhaps most importantly, demonstrates utility for Transformer models, whereas most efficient memory methods for RL so far have been restricted to recurrent processing.

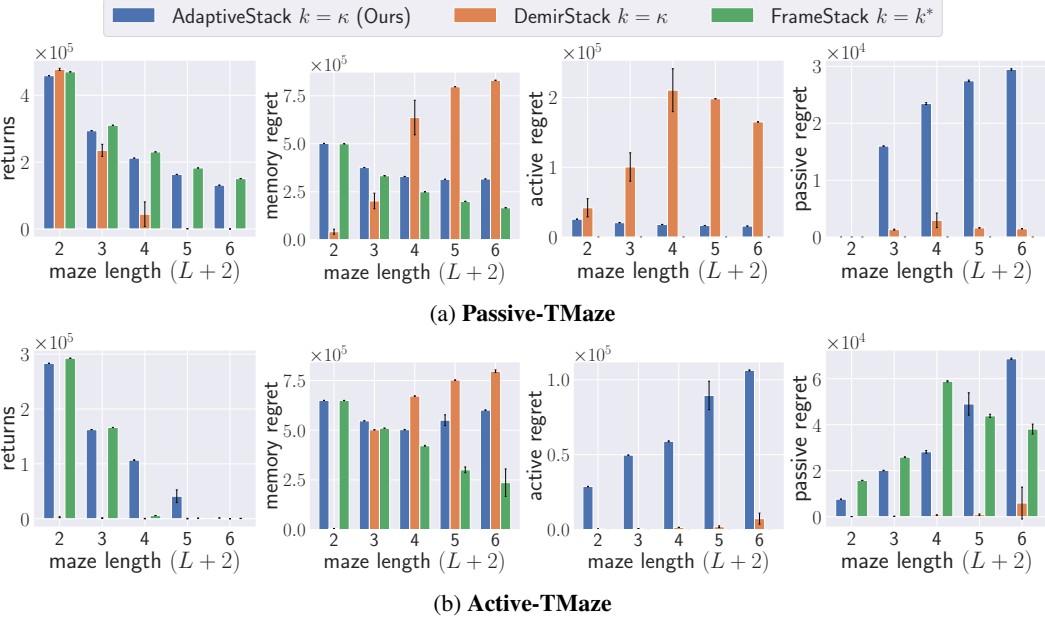

(a) **Passive-TMaze**

(b) **Active-TMaze**

Figure 30: Comparison with Demir (2023) (DemirStack) in the Continual TMazes with Q-learning ($N_{rs} = 5$). Our approach significantly outperforms it while it significantly struggles in the **Active-Tmaze**, which shows the importance of theoretically grounded memory management like AS and FS.

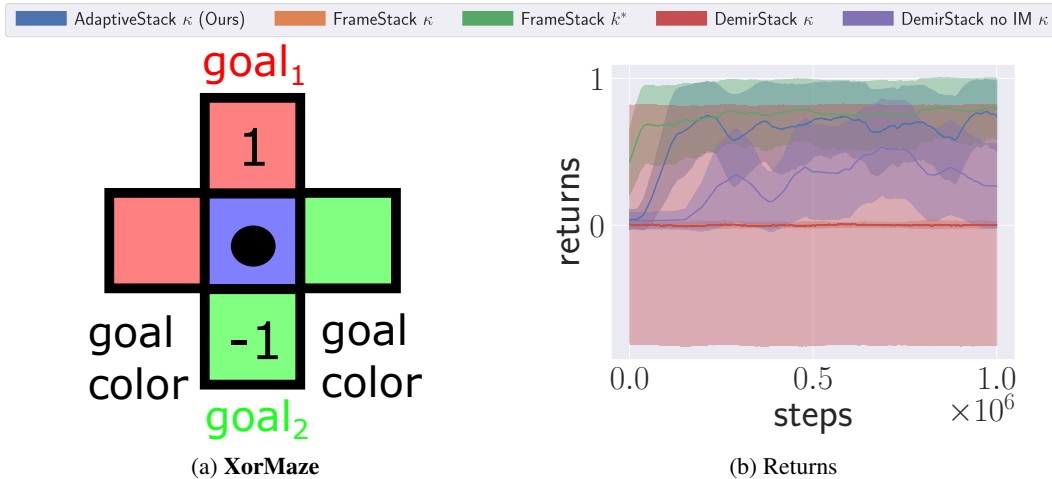

(a) **XorMaze**

(b) Returns

Figure 31: Comparison with Demir (2023) (DemirStack) in the **XorMaze** task using Q-learning ($N_{rs} = 5$). Every episode, the agent must remember both goal cues on the horizontal corners to reach the correct goal ($goal_1$ if both goal cues are the same, otherwise $goal_2$). In addition to this, it must add and remove observations from its stack to be able to know what direction to take each time it is at the junction. This is because it starts at the junction and has to always cross the junction to go to goal cues or final goal locations. We observe that DemirStack ($\kappa$) completely fails with even higher variance than FrameStack ($\kappa$), and struggles to learn without intrinsic motivation (IM). This is because some transitions may require adding a new observation while retaining the last ones. In contrast, AdaptiveStack ($\kappa$) learns to solve the task even with the same limited memory, converging to the same performance as the oracle FrameStack ($k^*$).

## F    COMPUTE AND MEMORY REQUIREMENTS FOR TRANSFORMERS

In this section we analyse the compute and memory efficiency of Adaptive Stacking compared to Frame Stacking. We provide compute and memory requirements for MLP, LSTM, and Transformer models as a function of $k$, the context length processed by the model. In Table 1 of the main text, we further extend these results leveraging the fact that for Frame Stacking to learn the optimal policy $k \geq k^*$ and for Adaptive Stacking to learn the optimal policy $k \geq \kappa$.

### F.1    MLP MODELS

Let us consider an MLP encoder model with weight precision $\mathcal{P}$ bytes per unit, $L$ layers, a hidden size of $h$, an action space size of $|\mathcal{A}|$, an inference batch size of $B_{\text{Inference}} = 1$, and a learning batch size of $B_{\text{Learn}} = B$. For uniformity with our analysis of the sequence models, we assume the input is already provided to the MLP in the form of $k$ embeddings of size $h$, so the total input size is $kh$. We also assume a Relu non-linearity for each layer. Additionally, in the case of the Frame Stacking model, we assume that there is an linear output head with a value for each environment action in $\mathcal{A}$. Furthermore, in the case of the Adaptive Stacking model there is another linear output head with a value for each of the $k$ memory eviction actions. The number of copies of the model $G$ that needs to be stored in memory to compute updates during training depends on the learning optimizer. In the case of SGD, we only need to store a single gradient $G = 1$, whereas for the popular AdamW optimizer $G = 4$.

### F.1.1    PRODUCING A SINGLE ACTION

**Compute of Frame Stacking.** In the first layer of the network $2kh^2 + h$ FLOPs are used for the linear layer due to the matrix multiplication and addition of bias (one multiply + one add per element of output). An additional $h$ FLOPs are used for the Relu non-linearity computations. For the next $L - 1$ layers, $2h^2 + 2h$ FLOPs are used. In the final layer, $2h|\mathcal{A}|$ FLOPs are used. Therefore, the total FLOPs for a single action generation is:

$$|c|_{a \sim \pi_\theta} = 2kh^2 + 2h + (L-1)(2h^2 + 2h) + 2h|\mathcal{A}| \in \Omega(k)$$

**Compute of Adaptive Stacking.** For Adaptive Stacking, we do the same amount of computation the first $L$ layers of the network, one using the standard last layer with output size $|\mathcal{A}|$ and one using a layer of size $k$ representing the memory action. This then brings the total FLOPs for a single action generation to:

$$|c|_{a \sim \pi_\theta} = 2kh^2 + 2h + (L-1)(2h^2 + 2h) + 2h(|\mathcal{A}| + k) \in \Omega(k)$$

**Memory of Frame Stacking.** For MLP inference, we do not need to store intermediate activations after they are used. They are only needed when computing gradients. As such, we lower bound the memory needed for action inference by the number of parameters and precision of the model $|w|_{a \sim \pi_\theta} \geq \mathcal{P}|\theta|$ where $|\theta| = |\theta|_{\text{MLP}} + |\theta|_{\text{stack}}$. The weight matrix in the first layer has $kh^2$ parameters, the weight matrix in the middle $L-1$ layers each have $h^2$ parameters, and the weight matrix in the last layers has $h|\mathcal{A}|$ parameters. The bias vector in the first $L$ layers each have $h$ parameters, and the bias vector in the last layer has $|\mathcal{A}|$ parameters. As such, the number of total number parameters is $|\theta|_{\text{MLP}} = kh^2 + (L-1)h^2 + Lh + (h+1)|\mathcal{A}|$. Additionally, the stack itself must store $|\theta|_{\text{stack}} = \mathcal{P}hk$. So the total RAM requirement of the model can be lower bounded as:

$$|w|_{a \sim \pi_\theta} \geq \mathcal{P}|\theta| = \mathcal{P}k(h^2 + h) + \mathcal{P}(L-1)h^2 + \mathcal{P}Lh + \mathcal{P}(h+1)|\mathcal{A}| \in \Omega(k)$$

**Memory of Adaptive Stacking.** The number of parameters in the Adaptive Stacking approach are the same for the first $L$ layers, with the addition of a final layer with a weight matrix of size $hk$ and a bias vector of size $k$. Additionally, the stack itself has the same number parameters as a function of $k$. So the total RAM requirement of the model can be lower bounded as:

$$|w|_{a \sim \pi_\theta} \geq \mathcal{P}|\theta| = \mathcal{P}k(h^2 + h) + \mathcal{P}(L-1)h^2 + \mathcal{P}Lh + \mathcal{P}(h+1)(|\mathcal{A}| + k) \in \Omega(k)$$

### F.1.2 PRODUCING A TD UPDATE

**Compute of Frame Stacking.** To compute a TD update, we must perform two forward propagations for each item in the batch. The additional forward propagation is for computing the bootstrapping target using a target network that is the same size as the original network. The cost of a backward propagation should match that of a forward propagation, so it is clear that $|c|_{\text{TD}} = 3B|c|_{a \sim \pi_\theta}$. This then brings the total FLOPs for a TD batch update to:

$$|c|_{\text{TD}} = 3B\left(2kh^2 + 2h + (L-1)(2h^2 + 2h) + 2h|\mathcal{A}|\right) \in \Omega(k)$$

**Compute of Adaptive Stacking.** In the case of Adaptive Stacking, we must perform TD updates for both the environment actions and the memory actions. Thus, we again have the relationship that $|c|_{\text{TD}} = 3B|c|_{a \sim \pi_\theta}$. This then implies that the total FLOPs for a TD batch update to:

$$|c|_{\text{TD}} = 3B\left(2kh^2 + 2h + (L-1)(2h^2 + 2h) + 2h(|\mathcal{A}| + k)\right) \in \Omega(k)$$

**Memory of Frame Stacking.** During a TD update, we must also store the target network in memory, which has the same number of parameters as the original MLP. We also must store the activations of the main network now to compute the gradients. Thus, we can lower bound the memory required as $|w|_{\text{TD}} \geq (2 + G)\mathcal{P}|\theta|_{\text{MLP}} + \mathcal{P}B|\theta|_{\text{stack}} + \mathcal{P}BhL$, meaning the total RAM requirement of the model can be lower bounded as:

$$|w|_{\text{TD}} \geq (2 + G)\left(\mathcal{P}kh^2 + \mathcal{P}(L-1)h^2 + \mathcal{P}Lh + \mathcal{P}(h+1)|\mathcal{A}|\right) + \mathcal{P}Bkh + \mathcal{P}BhL \in \Omega(k)$$

**Memory of Adaptive Stacking.** We again have the fact that $|w|_{\text{TD}} \geq (2+G)\mathcal{P}|\theta|_{\text{MLP}} + \mathcal{P}B|\theta|_{\text{stack}} + \mathcal{P}BhL$, but $|\theta|_{\text{MLP}}$ is different for Adaptive Stacking because of the extra final layer for the memory policy. So the total RAM requirement of the model can be lower bounded as:

$$|w|_{\text{TD}} \geq (2 + G)\left(\mathcal{P}kh^2 + \mathcal{P}(L-1)h^2 + \mathcal{P}Lh + \mathcal{P}(h+1)(|\mathcal{A}|+k)\right) + \mathcal{P}Bkh + \mathcal{P}BhL \in \Omega(k)$$

### F.2 LSTM Models

Let us consider an LSTM encoder model with weight precision $\mathcal{P}$ bytes per unit, $L$ layers, a hidden size of $h$, an action space size of $|\mathcal{A}|$, an inference batch size of $B_{\text{Inference}} = 1$, and a learning batch size of $B_{\text{Learn}} = B$. We assume the input is already provided in the form of $k$ embeddings of size $h$. Additionally, in the case of the Frame Stacking model, we assume that there is an linear output head with a value for each environment action in $\mathcal{A}$. Furthermore, in the case of the Adaptive Stacking model there is another linear output head with a value for each of the $k$ memory eviction actions. The number of copies of the model $G$ that needs to be stored in memory to compute updates during training depends on the learning optimizer. In the case of SGD, we only need to store a single gradient $G = 1$, whereas for the popular AdamW optimizer $G = 4$.

While it is well known that RNNs can have inference costs independent of the history length, we note that this only works in pure testing settings and is not relevant to the continual learning setting we explore in this work. The issue is that the historical examples must be re-encoded by the RNN if any update has happened to the network during this sequence.

#### F.2.1 Producing a Single Action

**Compute of Frame Stacking.** Each LSTM cell at a given time step performs operations for 4 gates: the input gate, the forget gate, the output gate, and the candidate cell update. Each gate for each item in the batch for each time-step requires a matrix multiplication with the input, a matrix multiplication with the last hidden state, an additive bias vector, and a cost per hidden unit of applying non-linearities. Thus for each of the $L$ layers we need $8h^2 + 4h + 16h$ FLOPs. For the last linear layer at the last step we need $2h|\mathcal{A}|$ FLOPs. This then brings the total FLOPs for a single action generation to:

$$|c|_{a\sim\pi_\theta} = kL\left(8h^2 + 20h\right) + 2h|\mathcal{A}| \in \Omega(k)$$

**Compute of Adaptive Stacking.** For Adaptive Stacking, we must do two passes through the $L$ layer LSTM and additionally produce a memory action with a final layer head requiring $2hk$ FLOPs. This then brings the total FLOPs for a single action generation to:

$$|c|_{a\sim\pi_\theta} = kL\left(8h^2 + 20h\right) + 2h(|\mathcal{A}| + k) \in \Omega(k)$$

**Memory of Frame Stacking.** As with the MLP network, $|w|_{a\sim\pi_\theta} \geq \mathcal{P}|\theta|$ where the total parameters can be decomposed as $|\theta| = |\theta|_{\text{LSTM}} + |\theta|_{\text{activation}} + |\theta|_{\text{stack}}$. The network consists of 4 gates in each layer, including two matrices with $h^2$ parameters and one bias vector with $h$ parameters. So, there are $8h^2 + 4h$ parameters per layer, and $L(8h^2 + 4h)$ parameters in the $L$ layers. The linear output layer then contains $(h + 1)|\mathcal{A}|$ parameters. The activation memory only needs to be stored at the current step during inference, requiring $\mathcal{P}hL$ bytes of memory. The stack itself requires $\mathcal{P}kh$ bytes of memory. So the total RAM requirement of the model can be lower bounded as:

$$\boxed{|w|_{a\sim\pi_\theta} \geq \mathcal{P}L(8h^2 + 4h) + \mathcal{P}(h+1)|\mathcal{A}| + \mathcal{P}hL + \mathcal{P}kh \in \Omega(k)}$$

**Memory of Adaptive Stacking.** The Adaptive Stacking case only adds the additional output layer for memory actions, which has $(h+1)k$ total parameters. So the total RAM requirement of the model can be lower bounded as:

$$\boxed{|w|_{a\sim\pi_\theta} \geq \mathcal{P}L(8h^2 + 4h) + \mathcal{P}(h+1)(|\mathcal{A}| + k) + \mathcal{P}hL + \mathcal{P}kh \in \Omega(k)}$$

### F.2.2 Producing a TD Update

**Compute of Frame Stacking.** To compute a TD update, we must perform two forward propagations for each item in the batch. The additional forward propagation is for computing the bootstrapping target using a target network that is the same size as the original network. The cost of a backward propagation should match that of a forward propagation, so it is clear that $|c|_{\text{TD}} = 3B|c|_{a\sim\pi_\theta}$. This then brings the total FLOPs for a TD batch update to:

$$\boxed{|c|_{\text{TD}} = 3BkL\left(8h^2 + 20h\right) + 6Bh|\mathcal{A}| \in \Omega(k)}$$

**Compute of Adaptive Stacking.** In the case of Adaptive Stacking, we must perform TD updates for both the environment actions and the memory actions. Thus, we again have the relationship that $|c|_{\text{TD}} = 3B|c|_{a\sim\pi_\theta}$. This then implies that the total FLOPs for a TD batch update to:

$$\boxed{|c|_{\text{TD}} = 3BkL\left(8h^2 + 20h\right) + 6Bh(|\mathcal{A}| + k) \in \Omega(k)}$$

**Memory of Frame Stacking.** During a TD update, we must also store the target network in memory, which has the same number of parameters as the original LSTM. We also must store the activations of the main network for all steps now to compute the gradients. Thus, we can lower bound the memory required as $|w|_{\text{TD}} \geq (2+G)\mathcal{P}|\theta|_{\text{LSTM}} + \mathcal{P}B|\theta|_{\text{stack}} + \mathcal{P}BkhL$, meaning the total RAM requirement of the model can be lower bounded as:

$$\boxed{|w|_{\text{TD}} \geq (2+G)\left(\mathcal{P}L(8h^2 + 4h) + \mathcal{P}(h+1)|\mathcal{A}|\right) + \mathcal{P}BkhL + \mathcal{P}Bkh \in \Omega(k)}$$

**Memory of Adaptive Stacking.** We again have the fact that $|w|_{\text{TD}} \geq (2+G)\mathcal{P}|\theta|_{\text{LSTM}} + \mathcal{P}B|\theta|_{\text{stack}} + \mathcal{P}BkhL$, but $|\theta|_{\text{LSTM}}$ is different for Adaptive Stacking because of the extra final layer for the memory policy. So the total RAM requirement of the model can be lower bounded as:

$$\boxed{|w|_{\text{TD}} \geq (2+G)\left(\mathcal{P}L(8h^2 + 4h) + \mathcal{P}(h+1)(|\mathcal{A}| + k)\right) + \mathcal{P}BkhL + \mathcal{P}Bkh \in \Omega(k)}$$

### F.3 Transformer Models

Let us consider an Transformer model with weight precision $\mathcal{P}$ bytes per unit, $L$ layers, a hidden size of $h$, an action space size of $|\mathcal{A}|$, an inference batch size of $B_{\text{Inference}} = 1$, and a learning batch size of $B_{\text{Learn}} = B$. We assume the input is already provided in the form of $k$ embeddings of size $h$. Additionally, in the case of the Frame Stacking model, we assume that there is an linear output head with a value for each environment action in $\mathcal{A}$. Furthermore, in the case of the Adaptive Stacking model there is another linear output head with a value for each of the $k$ memory eviction actions. The number of copies of the model $G$ that needs to be stored in memory to compute updates during training depends on the learning optimizer. For example, in the case of SGD, we only need to store a single gradient $G = 1$, whereas for the popular AdamW optimizer $G = 4$.

F.3.1 PRODUCING A SINGLE ACTION

**Compute of Frame Stacking.** We consider the analysis of the compute required for a typical Transformer from Narayanan et al. (2021). The compute cost $|c|$ of doing inference of the final hidden state over a batch size of $B_{\text{Inf}}$ over tokenized inputs with a context length of $k$ using a Transformer with $L$ layers and a hidden size of $h$ is $24LB_{\text{Inf}}kh^2 + 4LB_{\text{Inf}}k^2h$ the compute cost of the final logit layer producing values for each action in $\mathcal{A}$ is $2B_{\text{Inf}}h|\mathcal{A}|$ only applied once per sequence. So we can lower bound the compute cost of producing a single action (i.e. $B_{\text{Inf}} = 1$) as:

$$\boxed{|c|_{a\sim\pi_\theta} \geq 24Lh^2k + 4Lhk^2 + 2h|\mathcal{A}| \in \Omega(k^2)}$$

It is a lower bound because we do not include any pre-Transformer layers needed to produce embeddings for the input. We also do not include actions and rewards as part of the interaction history, which would bring the context length to $k' = 3k - 2$. Additionally, we do not include any recomputation costs that make sense to incur when we are bound by memory rather than compute – here we assume we are compute bound.

**Compute of Adaptive Stacking.** For producing a single action with Adaptive Stacking, the new compute overhead comes from the addition of the memory action head that comprises an extra $2hk$ FLOPs. This then brings the total FLOPs for a single action generation to:

$$\boxed{|c|_{a\sim\pi_\theta} \geq 24Lh^2k + 4Lhk^2 + 2h(|\mathcal{A}| + k) \in \Omega(k^2)}$$

**Memory of Frame Stacking.** We now assume that we are memory bound and not compute bound and include the cost of storing the model of parameter size $|\theta|$ at precision $\mathcal{P}$ where $|\theta| = |\theta|_{\text{Transformer}} + |\theta|_{\text{stack}} + |\theta|_{\text{activations}}$. In each Transformer layer, there are $4h^2$ parameters used to compute attention, $8h^2$ parameters used in the feedforward network, and $4h$ parameters used in the layer norm. If biases are used for all linear layers, there are an additional $9h$ parameters – we exclude these for now in the spirit of lower bounds as they do not change the asymptotic result in terms of $k$ either way. The output layer then has $(h + 1)|\mathcal{A}|$ parameters, making $|\theta|_{\text{Transformer}} = L(12h^2 + 4h) + (h + 1)|\mathcal{A}|$. The memory used for the stack itself is $\mathcal{P}kh$. Additionally, the cost of activations $\mathcal{P}khL$ assuming full re-computations at each step. This results in a lower bound on the working memory cost of producing a single action:

$$\boxed{|w|_{a\sim\pi_\theta} \geq \mathcal{P}L(12h^2 + 4h) + \mathcal{P}(h + 1)|\mathcal{A}|) + \mathcal{P}B(L + 1)hk \in \Omega(k)}$$

**Memory of Adaptive Stacking.** The main additional memory overhead of Adaptive Stacking is the output layer for the memory policy, which has $(h + 1)k$ parameters. This results in a lower bound on the working memory cost of producing a single action:

$$\boxed{|w|_{a\sim\pi_\theta} \geq \mathcal{P}L(12h^2 + 4h) + \mathcal{P}(h + 1)(|\mathcal{A}| + k) + \mathcal{P}(L + 1)hk \in \Omega(k)}$$

F.3.2 PRODUCING A TD UPDATE

**Compute of Frame Stacking.** To compute a TD update, we must perform two forward propagations for each item in the batch. The additional forward propagation is for computing the bootstrapping target using a target network that is the same size as the original network. The cost of a backward propagation should match that of a forward propagation, so it is clear that $|c|_{\text{TD}} = 3B|c|_{a\sim\pi_\theta}$. This then brings the total FLOPs for a TD batch update to:

$$\boxed{|c|_{\text{TD}} \geq 3B\left(24Lh^2k + 4Lhk^2 + 2h|\mathcal{A}|\right) \in \Omega(k^2)}$$

**Compute of Adaptive Stacking.** In the case of Adaptive Stacking, we must perform TD updates for both the environment actions and the memory actions. Thus, we again have the relationship that $|c|_{\text{TD}} = 3B|c|_{a\sim\pi_\theta}$. This then implies that the total FLOPs for a TD batch update to:

$$|c|_{\text{TD}} \geq 3B\left(24Lh^2k + 4Lhk^2 + 2h(|\mathcal{A}| + k)\right) \in \Omega(k^2)$$

**Memory of Frame Stacking.** To analyse the working memory requirements $|w|$ of producing a single action for a typical Transformer, we follow Anthony et al. (2023). We now assume that we are memory bound and not compute bound. During a TD update, we must also store the target network in memory, which has the same number of parameters as the original Transformer. Thus, we can lower bound the memory required as $|w|_{\text{TD}} \geq (2 + G)\mathcal{P}|\theta|_{\text{Transformer}} + \mathcal{P}B|\theta|_{\text{stack}} + \mathcal{P}B|\theta|_{\text{activations}}$, meaning the total RAM requirement of the model can be lower bounded as:

$$|w|_{\text{TD}} \geq (2 + G)\left(\mathcal{P}L(12h^2 + 4h) + \mathcal{P}(h + 1)|\mathcal{A}|\right) + \mathcal{P}B(L + 1)hk \in \Omega(k)$$

**Memory of Adaptive Stacking.** We again have the fact that $|w|_{\text{TD}} \geq (2 + G)\mathcal{P}|\theta|_{\text{Transformer}} + \mathcal{P}B|\theta|_{\text{stack}} + \mathcal{P}B|\theta|_{\text{activations}}$, but $|\theta|_{\text{Transformer}}$ is different for Adaptive Stacking because of the extra final layer for the memory policy. So the RAM requirement of the model can be lower bounded as:

$$|w|_{\text{TD}} \geq (2 + G)\left(\mathcal{P}L(12h^2 + 4h) + \mathcal{P}(h + 1)(|\mathcal{A}| + k)\right) + \mathcal{P}B(L + 1)hk \in \Omega(k)$$

