# OpenReview forum: "Learning What to Remember for Non-Markovian Reinforcement Learning"
_ICLR.cc/2026/Conference — Submitted to ICLR 2026_

### Official Review · Reviewer_piza · 2025-10-30

**Soundness:** 2
**Presentation:** 2
**Contribution:** 2
**Rating:** 2
**Confidence:** 4

**Summary:**

The authors concern themselves with a subset of POMDPs where a few distinct observations are sufficient to predict future value. In other words, POMDPs that admit "keyframe"-based approaches. Frame stacking is a simple and popular approach to POMDPs, where agents buffer the last $k$ observations and treat this as a Markov state. Such methods assume that the last $k$ frames are all that is necessary for value estimation, where in many cases this is not true. In this work, the authors propose to learn which $k$ frames to keep, rather than using the most recent $k$ frames. They do this by extending the agent action space to include an "overwrite" action: a discrete action that corresponds to the buffer index where the agent will store the most recent observation.

The authors provide a formal problem definition, introduce their method called "Adaptive Stacking", and then provide various theorems and proofs concerning the optimality of their approach. Finally, the perform various experiments evaluating their method. They report returns, regret, and other memory metrics on a T-maze task where optimal $k$ is known ahead of time. They find that their method outperforms others. Finally, they evaluate their method on more difficult tasks like XorMaze, Rubik's Cube, and POPGym.

**Strengths:**

The authors propose a simple approach that performs quite well. The idea of intelligently selecting a buffer of keyframes is a good idea. Finally, the authors provide promising results across various experiments.

**Weaknesses:**

## Writing and Contribution 2
The main shortcoming of this paper is its writing. The paper is generally cluttered and takes many words to say little. It feels as if the authors purposely introduce a lot of unnecessary math to motivate their **theoretical analysis** contribution.

Almost all theoretical contributions are based on these three facts:
- The authors only concern themselves with $k$th order POMDPs, where the last $k$ observations are always a Markov state: $s_t = x_{t:t-k}$
- Frame stacking $k$ of the most recent frames is therefore sufficient to solve $k$th order POMDPs (again, $s = x_{t:t-k}$)
- The authors' method subsumes a regular frame stacking approach ($s = x_{t:t-k}$)

I do not find value in any of the three theoretical contributions:
- Proposition 1: $V(s_t) = V(x_{t:t-k})$, this is just the definition of $k$th order POMDP
- Theorem 1: If standard framestacking is optimal, Adaptive Stacking can be too since it can learn to represent standard framestacking
- Theorem 2: $V(s_t) > V(s'\_t)$ implies $V(x_{t:t-k}) > V(x'\_{t:t-k})$, again this is just the definition of $k$th order POMDP ($s = x_{t:t-k}$)
- Theorem 3: Standard $Q$ learning converges to a fixed-point optimal policy (we can let $s = x_{t:t-k}$)

Beyond the unncessary analysis, there is much related work in method section (4.3) that should be integrated into related work section (3).

## Buffer Index Selection Methodology
The authors' method increases the dimensionality of the action space by $k$. In $Q$ learning, this means the size of the $Q$ table grows by $k$. For harder problems with large $k$, this can quickly become intractable. Furthermore, the proposed method is not content-aware. It cannot see what data current occupies a specific slot. Contrast this with the Differentiable Neural Computer [1] which both (1) decouples the $Q$ table growth from buffer size and (2) provides content-aware reading and writing via keys/queries. The authors should consider and compare their approach to such indexing methods.

**Questions:**

> selective gating mechanism that retains task-relevant information while filtering out
irrelevant inputs

The connection to computational psychology is tenuous. Humans do compress sensory information, but it is not clear that they use a keyframe based approach. Do they only consider $k$ distinct observations, or simply build a state from tiny bits of information from many observations?

> This equation (2) shows that the agent’s value under compressed memory is an expectation over possible latent histories. When the memory stack discards critical observations, this conditional distribution becomes broader, increasing uncertainty. As such, it is clear that Adaptive Stacking can be seen as a form of state abstraction in which multiple histories are compressed together in the estimate of the value function.

This is vague, uninformative, and frankly incorrect. You should define "compressed memory", how does it differ from "latent history"? Equation 2 does not contain an expectation. You cannot prove that the value distribution becomes "broader". What does it mean for multiple histories to be compressed together?

## References

[1] Graves et al., Hybrid computing using a neural network with dynamic external memory

---

> ### Author Response · Authors · 2025-11-27
>
> We are grateful to the reviewer for their time and effort spent reviewing our paper. We believe we have identified fundamental misunderstandings at the core of the reviewers' concerns regarding our theoretical contributions. Hence, with the following clarifications we hope we can gain their vote of confidence and improve their outlook on our paper -- particularly with respect to the significance of our contributions.
>
> > The authors only concern themselves with $k$th order POMDPs, where the last $k$ observations are always a Markov state: $s_t = x_{t:t-k}$
>
> This is incorrect. **We do not define $s_t$ as the Markov state $x_{t:t-k}$.** We define $s_t$ as the memory stack (line 219), and also refer to it as the memory or agent state (line 221) following recent language and notation in RL [1,2]. Hence $s_t$ is not Markov since it can exclude observations from the history, as demonstrated with the TMaze example in page 5 (moved from the Appendix and updated for space and clarity).
>
> However, we understand why the reviewer got confused, as it is our fault for using recent notation and language from the RL community that has not yet had time to proliferate to the broader ICLR community. We are happy to modify this notation to $m_t$ throughout to refer to the memory stack if the reviewer prefers.
>
> **All of the other concerns in “Writing and Contribution 2” seem to stem from this one misunderstanding.**
>
>
> > They do this by extending the agent action space to include an "overwrite" action: a discrete action that corresponds to the buffer index where the agent will store the most recent observation.
>
> This is incorrect. The most recent observation is always *pushed* to the top of our memory stack (lines 220-221). This is also illustrated with the TMaze example in page 5.
>
> > For harder problems with large $k$, this can quickly become intractable. Furthermore, the proposed method is not content-aware. It cannot see what data currently occupies a specific slot. Contrast this with the Differentiable Neural Computer [1] which both (1) decouples the
> $Q$ table growth from buffer size and (2) provides content-aware reading and writing via keys/queries. The authors should consider and compare their approach to such indexing methods.
>
> This is incorrect. **The policy $\pi(a_t,i_t|s_t)$ is context-aware as it is defined over the memory stack (the context) $s_t$.** This was explicitly stated on line 230. Hence when the agent chooses to remove the observation at position $i_t$ in the stack $s_t$, it sees that observation.
>
> To the best of our understanding, DNC can be considered more like a soft memory mechanism akin to LSTM. I.e. It’s an architectural approach. Even the DNC paper compares against LSTMs as their baseline. We compare Adaptive Stacking with Frame Stacking irrespective of the architecture (MLP, LSTM, and Transformer). **See Figure 1 and 9.  Figure 1 also highlights how the focus of this paper is not on specific architectures, but rather on how they update the sequence/context/stack with which they do the gradient updates for their policy.**
>
> We also hope the memory scaling experiment (Figure 5.b) demonstrates that large $k$ this is not much of a concern, as we see no performance degradation as $k$ increases. In fact, for a Transformer (even linear transformers), the input also scales at least linearly with $k$.
>
> > The connection to computational psychology is tenuous. Humans do compress sensory information, but it is not clear that they use a keyframe based approach. Do they only consider $k$ distinct observations, or simply build a state from tiny bits of information from many observations?
>
> It is unclear to us how the statements “they only consider $k$ distinct observations” and “simply build a state from tiny bits of information from many observations” are different. Can the reviewer please clarify? Although this may be our fault for not being clear enough about the relationship between a memory stack and the agent in our paper. To be precise, we see the memory stack as the context/sequence which the agent uses to build/update its state representation (e.g. using a sequence model). This is could be the full history (which means using Frame Stacking with $k\^\*=\infty$) or a subset of observations from this history. We hope the updated Figure 1 better frames this relationship to clarify this concern.
>
> [1] Shi Dong, Benjamin Van Roy, and Zhengyuan Zhou. Simple agent, complex environment: Efficient reinforcement learning with agent states. Journal of Machine Learning Research, 23(255):1–54, 2022.
>
> [2] Abel, David, André Barreto, Hado van Hasselt, Benjamin Van Roy, Doina Precup, and Satinder Singh. "On the convergence of bounded agents." arXiv preprint arXiv:2307.11044 (2023).

---

> > ### Author Response · Authors · 2025-11-27
> >
> > > This is vague, uninformative, and frankly incorrect. You should define "compressed memory", how does it differ from "latent history"? Equation 2 does not contain an expectation. You cannot prove that the value distribution becomes "broader". What does it mean for multiple histories to be compressed together?
> >
> > We apologise for the lack of clarity. We hope all of the clarifications given so far have given the reviewer enough insight to address this concern (particularly the TMaze example on page 5).
> >
> > **By "compressed memory" we mean $s_t$, and by “latent history” we mean $x_{t:t-k}$.** While we do not use the expectation symbol in that equation, that sum of probabilities times values is the definition of an expectation. We are happy to include the expectation symbol if the reviewer prefers that (similarly to equation 2 on page 14).
> >
> >
> > Finally, we would really appreciate it if the reviewer can critically evaluate our rebuttal, and look forward to engaging with them for further clarifications.

---

> > > ### Comment · Reviewer_piza · 2025-11-28
> > >
> > > >> The authors only concern themselves with $k$-th order POMDPs, where the last observations are always a Markov state: $s_t = x_{t:t-k}$
> > > >>> This is incorrect. We do not define $s_t$ as the Markov state $x_{t:t-k}$ .
> > >
> > > Perhaps I am still not understanding. On lines 115-120 you write
> > > > The environment is $k_*$-order Markovian (i.e. a $k_*$-order Markov Decision Process
> > > (Puterman, 2014)), meaning that $k_* \in N$ is the smallest number such that the probability function
> > > $Pr(x_{t+1},r_{t+1}|x_{t:t−k∗},a_t)$ is stationary regardless of the agent’s policy, where $x_{t:t−k_*}$ includes the
> > > last $k_*$ observations. If $k_* = 1$, then this is a standard Markov Decision Processes (MDP).
> > >
> > > Let's ignore the action for now. Then by definition, isn't $x_{t:t−k_*}$ a Markov state? You can define a Markov state in other ways too, but $x_{t:t−k_*}$ is still a Markov state because it satisfies the definition of $k_*$ POMDP. If an adaptive stack is Markov, it is a subset of $x_{t:t−k_*}$.
> > >
> > > My concern is that the theoretical contributions are based on two things:
> > > 1. By definition, $x_{t:t−k_*}$ is a Markov state in a $k_*$ POMDP
> > > 2. Adaptive stacking can represent $x_{t:t−k_*}$ with a stack size $k_*$ by retaining everything (frame stacking)
> > >
> > > This seems trivial to me. All your theorems are still true if you replace adaptive stacking with frame stacking, both with buffer size $k_*$, right? Would it suffice to say that $k_*$-adaptive stacking can degenerate into $k_*$-framestacking, and therefore can theoretically do everything frame stacking could do?
> > >
> > > I read the newly added text surrounding remark 1 about state aliasing. The claim is that are scenarios where state aliasing can lead to incorrect value functions, but still result in optimal policies, as shown in [1]. But I do not see how this is unique to adaptive stacking.
> > >
> > > > The most recent observation is always pushed... This is incorrect. The policy is context-aware...
> > >
> > > Thank you for correcting me. The action is not the push index, it is the pop index. And the policy is indeed context aware. I apologize for my mistakes, and my points on the DNC no longer stand. However, my last point still remains: the action space is coupled with $k_*$, and increasing the action space is expensive from an environment exploration point of view.
> > >
> > > > It is unclear to us how the statements “they only consider $k$ distinct observations” and “simply build a state from tiny bits of information from many observations” are different.
> > >
> > > I apologize for the ambiguous nature of this comment. I was referring to lines 87-98. My argument is that humans have very large observation spaces, and I consider it unlikely that we maintain a buffer of complete sensor information (full images, sounds, touches, etc) as in adaptive stacking. It is much more reasonable for humans to update some low-dimensional state representation as new sensory information comes in. Therefore, I find the claimed connection between adaptive stacking and human psychology weaker than other approaches to memory.
> > >
> > >
> > > [1] Singh et al, Learning Without State Estimation in Partially Observable
> > > Markovian Decision Processes

---

> > > > ### Author Response · Authors · 2025-12-02
> > > >
> > > > Thank you to the reviewer for taking the time to reply to our rebuttal. It is very unfortunate that we did not have an opportunity to continue to engage with them after the Open Review data leak as their reply has made it clear that their confusion stems from a very basic and fundamental misunderstanding of our notation and setting.
> > > >
> > > > **They appear to be missing the extremely important difference between the stack length $k$ and the degree of non-Markovness $k\^\*$, which we first formally established in the paragraph titled "The Problem" in lines 136 to 146, and re-iterated throughout the paper (from the abstract, to the intro, to the theory and experiments).** Missing this important detail would make it impossible to truly appreciate our theoretical results and experiments. Our analysis is not interesting for a stack size of $k\^\*$ (which we have been very straightforward about) as in this setting even Frame Stacking can arrive at the optimal policy. Instead, **we are interested in settings where the stack size $k$ is much less than $k\^\*$.** Indeed, we consider multiple environments where $k\^\*$ is entirely unbounded depending on the policy. The reason for establishing the notion of $k\^\*$ is to establish the relation between non-Markovian environments and Markovian environments for pedagogical reasons. When we have a stack of size $k << k\^\*$, it is not obvious that it is possible to still learn the optimal policy, which is why our results are highly non-trivial and novel.
> > > >
> > > > Hence, yes, the last $k\^\*$ observations is a Markov state and the last $k < k\^\*$ is not. Yes, with a stack size of $k\^\*$ Adaptive Stacking may learn the Frame Stacking solution. This however has nothing to do with our analysis in the paper which explicitly focusses on the non-Markovian setting where $k < k\^\*$.
> > > >
> > > > > However, my last point still remains: the action space is coupled with $k\^\*$, ...
> > > >
> > > > No, the action space is coupled with $k$ and not $k\^\*$, which is why this causes insignificant overhead in our experiments.
> > > >
> > > > > Therefore, I find the claimed connection between adaptive stacking and human psychology weaker than other approaches to memory.
> > > >
> > > > The reviewer's concern regarding biological plausibility equally applies to Frame Stacking, which is our main baseline and the most prominent approach in the literature for non-Markovian settings (e.g. when training LSTM and Transformer memory models). Importantly, we do not claim that our approach is a cognitive nor mechanistic model of human memory, nor do we seek one. We rather seek an approach to learning in challenging non-Markovian environments that integrates seamlessly with popular neural network architectures.

---

### Official Review · Reviewer_DARz · 2025-10-31

**Soundness:** 3
**Presentation:** 2
**Contribution:** 2
**Rating:** 4
**Confidence:** 4

**Summary:**

This paper introduces Adaptive Stacking, a method for efficient memory use in RL under non-Markovian conditions. Unlike Frame Stacking, which keeps a fixed window of recent observations, Adaptive Stacking learns which past observations to retain or discard through a memory policy. The authors prove convergence under both Monte Carlo and TD learning and view the method as a structured state abstraction. Experiments on TMaze, XorMaze, VelocityOnlyCartPoleHard POPGym task , and Rubik's Cube show that it matches or surpasses Frame Stacking while using much less memory and computation.

**Strengths:**

1. **Interesting idea in promising area of memory-intensive RL:** Adaptive Stacking lets the agent decide what to keep in memory instead of using a fixed-size history window. This is a conceptually simple but powerful shift from passive to active memory control.
2. **Theoretical analysis:** Theoretical analysis is rare among works on memory in RL.
3. **Solid experimental results:** The experiments show consistent improvements over Frame Stacking across multiple environments. AS matches oracle performance with far smaller memory and achieves lower "memory regret",  demonstrating its ability to retain only useful observations.

**Weaknesses:**

1. **Insufficient exposition of the method in the main text:** The central algorithm (Algorithm 1) is specified in the Appendix without any detailed description either in the main text or in the Appendix (which is optional reading). The main text provides only a fairly general rule for updating the frame stack (Equation 1) and a general diagram (Figure 1) without describing the details. To convey the proposed idea more clearly, more attention should be paid to the method in the main text (for example, by moving part of the theoretical section to the Appendix).
2. **Reproducibility**: The authors did not attach the code to their submission and did not provide a more detailed description of the operations used in Algorithm 1. For example, how are the pop() and push() operations defined? Personally, after reading the text, the principle of the method remains unclear to me.
3. **Limited experimental diversity:** The paper considers only relatively simple low-dimensional diagnostic tasks, while full-fledged 2D (e.g., Minigrid tasks) or 3D (e.g., ViZDoom-Two-Colors) tasks are not considered. Additionally, no continuous-control or high-dimensional sensory domains are evaluated, so the claimed scalability to "big worlds" (L068) remains speculative. The results on Rubik’s Cube and single POPGym task, while interesting, are not sufficient to demonstrate generality.
4. **Clarity and presentation:** The exposition is extremely dense, with excessive theoretical framing relative to algorithmic intuition. The connection between theoretical abstraction (state compression) and the empirical AS implementation could be made explicit. Figure 1 and Appendix, Figure 6 illustrate the concept, but there is little narrative about how the agent learns which memory to drop. Overall, the readability of the work needs to be improved.
5. **Limited Related Work:** The paper considers only agents without memory, RNN-based agents, and Transformer-based agents, but in the field of RL there is a whole class of algorithms designed to solve memory-intensive tasks (for example, using linear attention [1], memory tokens [2], SSMs [3], etc.).

**Questions:**

1. How are push() and pop() operations defined?
2. The paper states that the proposed method is general and does not depend on the chosen RL policy, although experiments are only presented for q-learning and PPO. Where can I find the pseudocode for adaptive stacking+PPO? Is it possible to get a general idea of the pseudocode for an arbitrary RL policy to support the claims in the paper?
3. What is the empirical overhead of learning over both $a_t$​ and ​ $i_t$? For large $k$, does the discrete selection make exploration unstable?
4. How will the proposed method work on tasks that require rewriting information to memory or retaining information about multiple events (for example, if we are dealing with a multidimensional T-Maze, where we may have more than one cue-turn pair, or if some previously important information becomes obsolete over time in the environment and at a certain point we need to remember other information)?
5. How does the proposed method behave on classic MDP tasks where we are dealing with states rather than observations? Are there any performance drops or additional computational overheads?
6. If we consider a simple k-order MDP (e.g., Atari Pong/Breakout), are there any confusing observations in the Adaptive Stack that interfere with the internal estimation of the direction/speed of the projectiles?
7. In tasks violating Assumption 4.1, how does AS behave empirically? Does it degrade gracefully, or can it catastrophically forget relevant histories?

Certain elements of the work would benefit from additional clarification or evidence, which I look forward to seeing addressed in the rebuttal.

[1] Pramanik, Subhojeet, et al. "AGaLiTe: Approximate Gated Linear Transformers for Online Reinforcement Learning." arXiv preprint arXiv:2310.15719 (2023).

[2] Cherepanov, Egor, et al. "Recurrent action transformer with memory." arXiv preprint arXiv:2306.09459 (2023).

[3] Lu, Chris, et al. "Structured state space models for in-context reinforcement learning." Advances in Neural Information Processing Systems 36 (2023): 47016-47031.

---

> ### Author Response · Authors · 2025-11-27
>
> We are grateful to the reviewer for their time and effort spent reviewing our paper. With the following clarifications, we hope we can gain their vote of confidence and actually improve their outlook on our paper -- particularly with respect to the significance of its contributions.
>
> # W1,W2,Q1,Q2
>
> We have updated the RL loop in Figure 1 (page 1) with more details to better frame our approach. Precisely, we position Adaptive Stacking as a drop-in replacement for Frame stacking. Hence similarly to Frame Stacking, Adaptive Stacking is implemented simply as a wrapper over the environment when training with stable-baselines (we have attached our code as supplementary material). **We also include new results showing Adaptive Stacking generalizing to a maze of length 1 million irrespective of architecture while Frame Stacking with the same amount of memory fails.**
>
> We have also moved the TMaze example from the Appendix to page 5, and horizontally flipped the stacks in all our figures to make the sliding window of Frame Stacking even clearer. Pop and push use their standard precise definitions in computer science.  pop($s_t$, $i_t$) removes the ith element from the stack s, and push($s_t$, $x_t$) adds $x_t$ to the top of the stack.
>
> # W3
>
> > The paper considers only relatively simple low-dimensional diagnostic tasks, while full-fledged 2D (e.g., Minigrid tasks) or 3D (e.g., ViZDoom-Two-Colors) tasks are not considered.
>
> Thank you to the reviewer for the suggestions. Please note that the suggested domains like VizDoom-two-colors and MemoryGym respectively require tens of millions ([9] Fig. 4.b) and hundreds of millions ([7] Figs. 6-9) of timesteps to train. However the compute budget we set for our experiments is 1 million training steps (even MiniGrid-Memory is not able to be sufficiently trained within this budget for the baselines).
>
> > so the claimed scalability to "big worlds" (L068) remains speculative
>
> These 2D or 3D domains require so many training steps because feature extraction from vision in RL is hard, even though their degree of non-Markovness ($k\^\*$) is low, significantly lower than ours (with millions and even infinite $k\^\*$, see Table 2 page 21). This makes sense for these works since they focus on architectural improvements: given a sequence/context/stack, how to learn a “good” memory state representation. However we are focused on how that sequence/context/stack itself is updated, a question which becomes extremely important when $k\^\*$ gets impractically large (e.g. infinite). The key insight of our work is that most prior work uses Frame Stacking (a FIFO memory) to update the sequence/context/stack (which is principled and theoretically sound by definition), but leads to compute, memory, and generalization tradeoffs for large $k\^\*$ (in addition to needing oracle knowledge of $k\^\*$). Hence by “big worlds” we are referring to domains with extremely large $k\^\*$ (even infinite) relative to the agent’s sequence/context/stack length $k$. **Please note how this was already emphasized in the original submission in the first paragraph of the introduction (lines 67-69), in the problem statement (page 3), and even in the abstract and throughout the paper**.
>
> > Additionally, no continuous-control or high-dimensional sensory domains are evaluated
>
> The reviewer’s suggestion inspired us to also add an experiment on the Mujoco FetchReach task [8], modified to be partially observable (Figures 3.d and 6.b). Here the goal position changes every 50 steps and is only observable for the first 2 steps after each change. We also hide the joint velocities for all timesteps. **Our results demonstrate that the Adaptive Stacking agent achieves $10^5$ times higher number of successes during training, and is even able to remember the goal position after 1 million steps in an episode** during testing. To the best of our knowledge this is the first work in RL demonstrating memorization over that order of magnitude, while the highest in prior works is only over **10^3** orders of magnitude (Ni et al., 2023) (also see Table 2 page 21).
>
> # Q3
>
> The memory scaling results (Figure 3.a) shows that there’s little to no impact on sample efficiency of learning over $a_t$ and $i_t$ as $k$ gets larger. In fact, we observe that the generally high training variance of Adaptive Stacking reduces as $k$ gets larger, leading to equivalent training performance to Frame Stacking when $k \geq k\^\*=18$. A deeper analysis of the learned policies shows that this may be because of the increasingly more compact state abstraction learned by Adaptive Stacking (Figure 27.b).

---

> > ### Author Response · Authors · 2025-11-27
> >
> > # Q4
> >
> > If we understood the reviewer’s point correctly, the continual TMazes and the XORMaze demonstrate exactly this scenario (see full descriptions in Appendix D.1 pages 23-24).
> > In the continual TMazes, there’s no episode termination. So when the agent reaches a goal, it receives the corresponding reward, the goal cue is then randomly changed and the agent transitions automatically to the start location without resetting the memory stack. Hence it must learn to remember the goal cue to reach the correct goal location, then choose to discard it when it transitions back to the starting location.
> > In the XORMaze, the agent must remember both goal cues on the horizontal corners to reach the correct goal (goal1 if both goal cues are the same, otherwise goal2). In addition to this, it must add and remove observations from its stack to be able to know what direction to take each time it is at the junction. This is because it starts at the junction and has to always cross the junction to go to goal cues or final goal locations.
> >
> > # Q5
> >
> > On classic MDP tasks, $k=k\^\*=\kappa=1$ making Frame Stacking and Adaptive Stacking equivalent and the same as the MDP (since the top of the stack is always the current observation $x_t$). If the reviewer was instead wondering about what happens when we still choose to use $k>1$ in such a setting, we observed in the unmodified fully observable FetchReach that Frame Stacking and Adaptive Stacking get the same performance as when $k=1$. This is likely because Adaptive Stacking just learns to ignore the memory actions. We are happy to run this experiment over multiple seeds and include them in the camera ready.
> >
> > # Q6
> >
> > Given the reviewer’s example using direction/speed, we hope that the POPGym Cartpole (Figure 19. Page 29) and Mujoco FetchReach tasks clarify this point. In both tasks, the velocity information is occluded from the observation, hence the agent must infer direction/speed from just positional information. Our results show that the Adaptive Stacking agent is able to do this successfully. This is because, just like Frame Stacking, the Adaptive Stacking maintains temporal order.
> >
> > # Q7
> >
> > The partially observable FetchReach task is an example violating Assumption 4.1. Unfortunately, Adaptive Stacking still performed really well here, so we are still unable to give an example showing performance degradation. This may suggest that Assumption 4.1 is a sufficient but not necessary condition for Adaptive Stacking (the approach may be more general than what we are comfortable claiming in this paper), which is an interesting investigation for future works.
> >
> >
> > Finally, we would really appreciate it if the reviewers can critically evaluate our rebuttal, and look forward to engaging with them for further clarifications.

---

### Official Review · Reviewer_8iuy · 2025-10-31

**Soundness:** 2
**Presentation:** 2
**Contribution:** 3
**Rating:** 4
**Confidence:** 4

**Summary:**

This paper explores the use of adaptive stacking memory, a discrete external memory mechanism controlled by a reinforcement learning agent, to address partially observable decision-making problems. The central idea is to allow the agent to selectively store observations by choosing memory operations as part of its action space.

The authors provide a theoretical analysis, establishing conditions under which both Monte Carlo and Temporal Difference learning methods can successfully learn to control this memory structure in partially observable environments.

Empirical results demonstrate that agents can, in practice, learn to use adaptive stacking to solve a range of partially observable tasks. The method is also compared to soft memory architectures such as LSTMs and Transformers. A key finding is that adaptive stacking achieves comparable performance while, in principle, requiring significantly fewer computational resources—suggesting it may be a more efficient alternative in certain settings.

**Strengths:**

The idea of equipping reinforcement learning (RL) agents with external memory that they can manipulate through actions has a long history [1–5]. However, to the best of my knowledge, this is the first work to formally establish conditions under which such an approach can be guaranteed to work. While I find the empirical results less compelling—since similar conclusions have been reported in prior work [3–5], to some extent—the theoretical contributions represent a meaningful advancement. These results offer valuable insights into the broader research effort on memory-augmented RL.

Although I remain skeptical that hard memory mechanisms alone will be sufficient to solve partially observable problems in the long term (as discussed in the weaknesses section), I consider this paper an important milestone in the development of memory-augmented RL methods.

In fact, I encourage the authors to consider generalizing their theoretical results to encompass a broader class of external memory architectures [4]. Such an extension could significantly enhance the impact of the work. From my understanding, this generalization may be relatively straightforward to achieve and would further strengthen the paper’s contributions.

---

**References**

[1] Littman, M. L. (1993). An optimization-based categorization of reinforcement learning environments. From animals to animats, 2, 262-270.

[2] Peshkin, L., Meuleau, N., & Kaelbling, L. (2001). Learning policies with external memory. arXiv preprint cs/0103003.

[3] Zhang, M., McCarthy, Z., Finn, C., Levine, S., & Abbeel, P. (2016, May). Learning deep neural network policies with continuous memory states. In 2016 IEEE international conference on robotics and automation (ICRA) (pp. 520-527). IEEE.

[4] Icarte, R. T., Valenzano, R., Klassen, T. Q., Christoffersen, P., Farahmand, A. M., & McIlraith, S. A. (2020). The act of remembering: A study in partially observable reinforcement learning. arXiv preprint arXiv:2010.01753.

[5] Demir, A. (2023). Learning what to memorize: Using intrinsic motivation to form useful memory in partially observable reinforcement learning. Applied Intelligence, 53(16), 19074-19092.

**Weaknesses:**

**Presentation and Relation to Prior Work**

The paper is somewhat challenging to follow due to its frequent references to the appendix. For instance, some figures discussed in the main body (e.g., Figure 7a) are located in the appendix, requiring readers to continually switch back and forth to fully grasp the content. This disrupts the reading experience and undermines the clarity and coherence of the presentation.

Additionally, the paper shares notable conceptual similarities with the work of Icarte et al. (2020). I encourage the authors to explicitly discuss these connections, clearly articulating both the similarities and the distinctions, to better situate their contribution within the existing literature.

---

**Limitations**

The central idea of using a "hard memory"—where the agent explicitly decides what to store based on its actions—presents a fundamental limitation. It is inherently difficult to determine, a priori, whether a particular observation should be stored in external memory. For example, suppose an agent enters a room and sees a picture. Should it store this observation? At that moment, there is no way to know whether this information will be relevant in the future. If, hypothetically, recalling the picture becomes crucial 1000 steps later, the agent must retain it for that entire duration. However, under a random exploration policy, the probability of retaining that memory diminishes rapidly, as each step introduces a chance of overwriting it. This poses a significant challenge for reinforcement learning agents, which rely on exploration to learn optimal behavior.

Another concern arises from allowing the agent to manipulate its external memory through actions, as this opens the door for the agent to "fool itself" into believing it has reached a favorable state when it has not. In the XorMaze domain, for example—assuming the agent observes only the color of its current location—the optimal policy would involve storing the color of the first room (e.g., green), then navigating to the second room and storing that color as well. If both rooms are green, the resulting memory state [green, green] would correctly lead the agent to the 'equal color' goal and a reward. However, due to the nature of temporal-difference (TD) learning, such an optimal policy is not stable. Instead, the agent may learn to store the first room’s color twice, thereby fabricating the high-value memory state [green, green] without actually visiting both rooms. This non-Markovian shortcut is locally optimal under one-step TD learning, as it transitions the agent from a low-value to a high-value state. As a result, learning to control external memory becomes complex and unstable—even in the tabular setting—raising concerns about the robustness and reliability of the learned policies.

A more intuitive example involves an agent tasked with cleaning a room and then plugging itself in to receive a reward. If the agent stores a picture of a clean room, it may mistakenly believe the room is clean, even if it hasn't performed the cleaning task. Since the external memory is the only source of state information, the agent cannot distinguish between a memory of a clean room and the actual clean state. This ambiguity can lead to unstable learning in most POMDPs.

I appreciate the authors' efforts to introduce assumptions (e.g., Assumption 4.1) that create settings where the proposed approach can succeed. However, the need for such assumptions underscores the core issue. It may simply be unwise to allow RL agents to directly control their memory.

In contrast, alternative approaches based on soft (differentiable) memory architectures—such as LSTMs or Transformers—do not suffer from the same issues. For example, when training an LSTM, the entire trajectory is taken into account, with gradients backpropagated from the end to the beginning. This enables the model to recognize temporally extended patterns that contribute to high returns by leveraging the full trajectory—giving it a clear advantage over hard memories. In adaptive stacking, the agent must decide in the moment whether storing a piece of information will be useful later, without knowing future outcomes. By contrast, LSTMs benefit from backward credit assignment, where the learning signal propagates from future rewards to earlier decisions. While LSTMs are known to suffer from vanishing gradients, this issue is largely mitigated in Transformer-based architectures, which further strengthens the case for using differentiable memory mechanisms in RL.

In short, there are good reasons to believe that soft memory mechanisms are better suited for handling partial observability than hard memory. That said, hybrid approaches combining both paradigms may offer promising directions for future work.

---
**Convergence of Temporal Difference Learning**

Theorem 3 appears incomplete, as it does not reference Assumption 4.1. The more complete version—Theorem 5 in the appendix—states that Q-learning converges to an optimal policy "if policies in $\mathcal{M}_t$​ are value-consistent." Does this imply that every policy must be value-consistent? If so, the assumption seems overly strong and likely only holds in fully observable settings. For instance, in any partially observable domain, a policy that always erases memory would presumably not be value-consistent.

In fact, from my reading, the domains in Appendix C appear to be value-consistent for some policies, but not for all. If that interpretation is correct, Theorem 5 would not apply to those domains. Am I misunderstanding something? Ideally, the theorem would only require the optimal policy to be value-consistent. However, the proof seems to rely on the stronger assumption that all policies are value-consistent—which, as noted, is quite restrictive.

Additionally, the theorem assumes a “fixed exploratory policy.” It would be helpful if the authors clarified what this entails. Is it a fixed behavior policy that is also value-consistent? If so, the theorem might apply more broadly—but only under the assumption that the behavior policy already makes effective use of memory. In that case, the environment becomes effectively Markovian, and the result becomes less interesting, as it sidesteps the core challenge of learning how to use external memory effectively in the first place.


---

**Experiments**

The experimental section is limited to only four domains (T-Maze, XorMaze, Rubik’s Cube, and POPGym Cartpole), despite the fact that Appendix C lists many more domains where the proposed method should, in principle, be applicable. This narrow selection raises concerns about the generality of the empirical findings.

Moreover, the comparison with alternative memory architectures—such as LSTMs or Transformers—is conducted only on the simplest domain (the passive T-Maze). This makes it difficult to assess the relative strengths of Adaptive Stacking in more complex settings. I also suspect that the LSTM baseline, which uses only the current observation, could perform significantly better with more extensive hyperparameter tuning. In my experience, increasing the batch size and using multiple agents can substantially improve PPO-LSTM performance in partially observable environments.

Additionally, it is unclear why Adaptive Stacking outperforms other memory mechanisms such as those proposed by Icarte et al. (2020) and Demir (2023), especially given that those approaches use much simpler memory action spaces (e.g., push or no-push). In contrast, Adaptive Stacking requires more complex decisions, such as selecting which memory entry to erase. According to Figure 27, its performance is substantially better, which seems surprising. One possible explanation is that the intrinsic motivation mechanism is influencing the results in unexpected ways. If that is the case, it might have been more informative to evaluate the method without intrinsic rewards.

Finally, a central claim of the paper is that Adaptive Stacking is more efficient than alternative methods. However, the experimental section does not report computational time or memory usage. While this is a minor concern, including such metrics would strengthen the argument for the method’s practical advantages.

---

**References**

[1] Icarte, R. T., Valenzano, R., Klassen, T. Q., Christoffersen, P., Farahmand, A. M., & McIlraith, S. A. (2020). The act of remembering: A study in partially observable reinforcement learning. arXiv preprint arXiv:2010.01753.

[2] Demir, A. (2023). Learning what to memorize: Using intrinsic motivation to form useful memory in partially observable reinforcement learning. Applied Intelligence, 53(16), 19074-19092.

**Questions:**

1. **Hard vs. Soft Memory Architectures**: Could the authors elaborate on the long-term benefits and trade-offs of using hard (discrete, agent-controlled) versus soft (differentiable) memory mechanisms for solving partially observable problems? As I mentioned earlier, I am inclined to believe that hard memory may be less effective in the long run, but I may be overlooking important advantages.
2. **Clarification on Theorem 5**: Could you clarify the assumptions underlying Theorem 5? Specifically, does it require that the agent follows a fixed policy that is already value-consistent (i.e., one that makes effective use of memory)? Or does it assume that all policies in the policy class are value-consistent? Or both?
3. **Scope of Theorem 5**: Does Theorem 5 imply that Q-learning will, in principle, converge to an optimal policy for all the domains listed in Appendix C, assuming arbitrary Q-value initialization and sufficient exploration (e.g., via $\epsilon$-greedy policies in the limit)?
4. **Performance of Adaptive Stacking**: Could the authors provide insight into why Adaptive Stacking outperforms the memory mechanisms proposed by Demir (2023)? Given that those approaches use simpler memory action spaces (e.g., push/no-push), it is surprising that Adaptive Stacking, which involves more complex memory control decisions, performs significantly better (as shown in Figure 27). One possible explanation might be the influence of the intrinsic motivation mechanism, but if that is the case, it would be helpful to see results without it.
5. **Computational Efficiency**: One of the paper’s key claims is that Adaptive Stacking is more computationally efficient than soft memory architectures. Could the authors provide empirical evidence to support this claim, such as runtime or memory usage comparisons?

---

> ### Author Response · Authors · 2025-11-27
>
> We are grateful to the reviewer for their time and effort spent reviewing our paper. We really appreciate the deep dive and the positive comments. Given that, we believe that most of the reviewers' concerns are because of a couple of misunderstandings, which we hope are addressed below.
>
> Before addressing the questions, we would like to start by clarifying the framing of our paper in case that is a source of misunderstanding (as the following point is reflected throughout the review)
>
> > In short, there are good reasons to believe that soft memory mechanisms are better suited for handling partial observability than hard memory. That said, hybrid approaches combining both paradigms may offer promising directions for future work.
>
> In this paper, we actually frame most prior works with memory as hybrid approaches. E.g. A “soft memory” model like an LSTM uses a “hard memory” like Frame Stacking for backpropagation through time. **We have updated Figure 1 to illustrate this better, and also include new results showing Adaptive Stacking generalising to a maze of length 1 million.** To be precise, **we see a memory stack as the context/sequence which the agent uses to build/update its state representation ( e.g. with MLP,  LSTM, Transformers, etc).** This memory stack could be the full history (which is equivalent to Frame Stacking with $k=k\^\*$ in our terminology) as is done in most prior work, or a subset of observations from the history (e.g. using Frame Stacking or Adaptive Stacking with $k<k\^\*$). Hence we see Frame stacking as the natural baseline for Adaptive Stacking, irrespective of the soft model being used. For example, for an LSTM with $k<k\^\*$, instead of removing the last observation in the history buffer when it is full, we let the agent decide which one to remove.
>
> # Q1
>
> The most important advantage is in continual settings (no episodes) and settings where the degree of non-Markovness ($k\^\*$) is impractically large (e.g. TMaze length of 1 million)  or even infinite (e.g. arbitrarily long maze lengths), as it becomes impossible or impractical for the entire trajectory to be taken into account (with gradients backpropagated from the end to the beginning). In such cases, hard memory becomes a necessity, not a choice (the agent must choose a subset k of the history). This is especially true for agents with bounded memory and compute [1]. This is why we frame this paper based on the “big world hypothesis”, where in our case “big word” means $k\^\*>>k$. Our experimental domains are also motivated by this setting, which is why they have orders of magnitude higher $k\^\*$ than all prior works (Table 2 page 21).
>
> The second main advantage is we may not know $k\^\*$.  If the environment is continual, and we know $k\^\*$ and that it is finite, soft memory methods can still use the last $k\^\*$ window of the trajectory. If we do not know $k\^\*$, then this becomes impossible.  For example the continual Passive-TMaze, where $k\^\*=L+2$ but there are no episodes (Figure 4.a).
>
> The third main advantage is generalization. Even if the entire trajectory can be taken into account (with gradients backpropagated from the end to the beginning), the soft memory agent struggles to generalize to significantly longer trajectory lengths as is known from the literature (e.g. [2]). See Figures 1 demonstrating this across all three architectures. We see that even the FrameStack Transformer agent trained on full episode rollouts fails to generalise to a maze of length 1 million (as the goal cue gets outside of its context window). Only the LSTM architecture is able to generalize thanks to the recurrent hidden state it maintains, but this requires backpropagation through time on the full episode rollout during training and hence scales poorly with compute and memory [Table 1]. It also leads to sample inefficiency in significantly harder tasks like our FetchReach one, where all architectures (MLP,LSTM, and Transformer) struggle to learn and generalise (Figures 6 and 29).
>
> # Q2
>
> The fixed exploratory policy is just an assumption needed to use Theorem 2 of Singh et al.. Hence we do not require that fixed exploratory policy to be value-consistent, since Singh et al. does not assume value-consistency to show that the Q-values converge to a fix point.
> However, we do assume that all policies in the policy class are value-consistent. For Q-learning this is the set of all deterministic policies.
>
> # Q3
> Yes, for the domains we marked as value consistent.
>
> > In fact, from my reading, the domains in Appendix C appear to be value-consistent for some policies, but not for all.
>
> We apologize if we made a mistake. Can the reviewer please highlight which domains we said are value consistent but which are actually not?

---

> > ### Author Response · Authors · 2025-11-27
> >
> > # Q4
> >
> > Indeed, the poor performance of Demir (2023) may be because of the use of intrinsic motivation. We are happy to add TMaze results without using intrinsic motivation in the camera-ready or by the end of rebuttal period time-allowing. For now, to combine the reviewers request here with that of reviewer FrDC (to reduce compute), we have added a comparison in the XorMaze task (Figure 31, page 37).
> >
> > We observe that DemirStack ($\kappa$) completely fails with even higher variance than FrameStack ($\kappa$), and struggles to learn without intrinsic motivation (IM). This is because some transitions may require adding a new observation while retaining the last ones. In contrast, AdaptiveStack ($\kappa$) learns to solve the task even with the same limited memory, converging to the same performance as the oracle FrameStack ($k\^\*$).
> >
> >
> > [1] Abel, David, André Barreto, Hado van Hasselt, Benjamin Van Roy, Doina Precup, and Satinder Singh. "On the convergence of bounded agents." arXiv preprint arXiv:2307.11044 (2023).
> >
> > [2] Cherepanov, E., Staroverov, A., Yudin, D., Kovalev, A. K., & Panov, A. I. (2023). Recurrent action transformer with memory. arXiv preprint arXiv:2306.09459.

---

### Official Review · Reviewer_FrdC · 2025-11-01

**Soundness:** 3
**Presentation:** 3
**Contribution:** 2
**Rating:** 6
**Confidence:** 3

**Summary:**

This submission proposes Adaptive Stacking (AS), a novel meta-learning algorithm that enables RL agents to learn which observations to retain in a fixed-size memory state rather than using standard frame stacking's FIFO policy. This is relevant for environments where only a relatively small subset of distant past observations are important for optimal decision-making. The authors provide convergence guarantees under a value-consistency assumption, analyzes computational benefits, and demonstrates the approach on T-Maze variants and other memory tasks.

**Strengths:**

1. The theoretical derivation on convergence is well-constructed with formal definitions, assumptions and proofs. I appreciate the detailed discussion on value consistency assumption.
2. The proposed AS framework is architecture-agnostic, demonstrating applicability to MLPs, LSTMs, and Transformers. The authors demonstrate practical benefits of AS over FS on multiple memory-based tasks.
3. Generally clear writing with good motivation, with comprehensive appendix, detailed proofs and experiments.
4. Detailed analysis and discussion on improvements in compute and memory (Table 1 and Appendix F).

**Weaknesses:**

1. The comparison with Demir (2023) is limited. Can you provide more comprehensive comparisons and justify when your action space design is preferable? It would be helpful to see more detailed comparison on all benchmarks against Demir 2023, which uses push/skip operations. Currently, only single comparison on TMazes is presented in the appendix (Figure 27).
2. In addition, since a lot of the tasks considered are short memory (e.g. maze length 2-6), why aren’t RNN-based agents that could potentially learn full belief state estimation (e.g. RL2, or methods with predictive representation such as VariBAD) included as baseline methods to compare against? I understand they might struggle with very long memory, but since the memory length requirements in your tasks are not super long, these methods should be included to fully demonstrate when memory management is better than belief state learning.
3. Most experiments are T-Maze variants (Figures 2, 3, 8-26), while POPGym shows marginal improvements with high variance, and Rubik's cube is interesting but under-analyzed. Since in the appendix a lot of tasks are said to satisfy the value consistency assumption, it would be more convincing to demonstrate the performance of AS on more diverse tasks.
4. Currently there seems to be little discussions on practical consideration of choosing κ. Can you provide more details on how to choose κ, what happens when κ is chosen too small, and how does performance vary as κ changes?

**Questions:**

1. The generalization results (Figures 19-26) seem promising. Could you provide more discussion on why AS generalizes better?
2. In the RNN-based agent section under related work, since the tasks you considered are POMDPs more generally, literature on methods such as RL2 (Duan et al, 2016; Wang et al, 2016) and methods extending RL2 with predictive representations such as VariBAD (Zintgraf et al, 2021) should also be cited.

---

> ### Author Response · Authors · 2025-11-27
>
> We are grateful to the reviewer for their time and effort spent reviewing our paper. With the following clarifications, we hope we can gain their vote of confidence and actually improve their outlook on our paper and the significance of its contributions.
>
> Before addressing the questions, we would like to start by clarifying the framing of our paper, as that seems to be a source of misunderstanding reflected throughout the review.
>
> We believe we may not have been clear enough about the relationship between a memory stack and the agent in our paper. **We have updated Figure 1 to illustrate this better, and also include new results showing Adaptive Stacking generalising to a maze of length 1 million. To be precise, we see a memory stack as the context/sequence which the agent uses to build/update its state representation (e.g. with MLP,  LSTM, Transformers, etc).** This memory stack could be the full history (which means using Frame Stacking with $k=k\^\*$) as is done in most prior work, or a subset of observations from the history (e.g. using Frame Stacking or Adaptive Stacking with $k<k\^\*$). A sequence model like an LSTM uses a memory stack like Frame Stacking for backpropagation through time. Hence we see Frame Stacking as the natural baseline for Adaptive Stacking, irrespective of the sequence model being used. For example, for an LSTM with $k<k\^\*$, instead of removing the last observation in the history buffer when it is full, we let the agent decide which one to remove.
>
>
>
>
> # W1
>
>
> We appreciate the reviewer’s comment here, as it highlights that our discussion of Demir (2023) may not have been clear or detailed enough. We have added a comparison in the XorMaze task (Figure 31, page 37), and expanded the discussion in Appendix E.4 (page 36). We provide a brief summary here:
>
>
> Through the lens of our theoretical analysis, we see Demir (2023) as still being a Frame Stacking approach, since their push action always removes the last observation from memory (bottom of the stack) if they need to add a new observation when the memory is full. However, that not only loses the theoretical guarantees of standard Frame Stacking due to their skip actions, but it also loses our theoretical guarantees because of their push actions. This is because some transitions may require adding a new observation while retaining the last ones.
>
>
> Given that, our aim in providing that single comparison experiment was to decide if Demir (2023) is still a reasonable baseline in addition to Frame Stacking. However, notice that they use intrinsic motivation which is unclear how to most straightforwardly extend to function approximation. In addition, while Demir (2023) compares to RNNs with function approximation, their memory management approach is instantiated as a purely tabular method using Sarsa($\lambda$) and compared against A2C and PPO. Their codebase also only contains tabular Sarsa($\lambda$). Hence we hope the reviewer can sympathise with our inability to provide apples to apples comparisons beyond tabular settings.
>
>
> In contrast, one of the main contributions of our work is to extend stack management to function approximation irrespective of architecture or RL algorithm, and to also provide rigorous theoretical analysis of when such approaches are expected to be beneficial with convergence guarantees. As Reviewer 8iuy highlighted in their strengths, “to the best of my knowledge, this is the first work to formally establish conditions under which such an approach can be guaranteed to work”.
>
>
>
>
> # W2
>
>
> We indeed compared Adaptive Stacking with Frame Stacking when using RNNs for both. For the RNN architecture, we used an LSTM since it is one of the most popular ones in prior works (Ni et al., 2023). In general, we focused our function approximation experiments with PPO using an MLP, LSTM, and Transformers for the same reason. We hope that these at least demonstrate the expected performance of swapping Frame Stacking with Adaptive Stacking in other RNN-based and Transformers-based agents.

---

> > ### Author Response · Authors · 2025-11-27
> >
> > # W3
> >
> >
> > Thank you to the reviewer for the suggestions. Please note that some of those domains like **MemoryGym require hundreds of millions ([7] Figs. 6-9) of timesteps to train. However the compute budget we set for our experiments is 1 million training steps** (even MiniGrid-Memory is not able to train sufficiently within this budget for the baselines).
> >
> > In addition, the degree of Non-Markovness ($k\^\*$) in all of those domains is low, significantly lower than ours (with millions and even infinite $k\^\*$, see Table 2 page 21). Their long training time is instead as a result of the difficulty of feature extraction, which is not the focus of our work. The key insight of our work is that most prior works use Frame Stacking (a FIFO memory) to update the sequence/context/stack (which is principled and theoretically sound by definition), but leads to compute, memory, and generalisation tradeoffs for large $k\^\*$ (in addition to needing oracle knowledge of $k\^\*$). This is why we frame this paper based on the “big world hypothesis”, where in our case “big word” means $k\^\*>>k$.
> >
> > However the reviewer’s suggestion inspired us to also add an experiment on the **Mujoco FetchReach task [8]**, modified to be partially observable (Figures 3.d and 6.b). Here the goal position changes every 50 steps and is only observable for the first 2 steps after each change. We also hide the joint velocities for all timesteps. Our results demonstrate that the Adaptive Stacking agent achieves orders of magnitude higher successes during training, and is even able to remember the goal position after **1 million steps in an episode during testing**. To the best of our knowledge this is the first work in RL demonstrating memorisation over that order of magnitude, while the highest in prior works is only over 10^3 orders of magnitude (Ni et al., 2023) (also see Table 2 page 21).
> >
> > # W4
> >
> > Please note that the aim of those experiments was to show that even when our approach only has the smallest amount of memory possible (that still admits an optimal policy), our approach is able to find that optimal policy by learning to manage its extremely small memory. This is in contrast to Frame Stacking which does require one to know the oracle amount of memory $k\^\*$ (which is generally the full episode trajectories in prior works). Of course, as the memory length grows it becomes easier for any agent to remember important observations, as shown by the memory scaling results (Figure 5.b). This shows that in practice, without knowing $k\^\*$, we can simply set the memory length $k$ to whatever is practical based on memory and compute limitations, and our agent will learn to make the best of it -- even if it is lower than $k\^\*$ (the amount of memory needed by Frame Stacking).
> >
> > # Q1
> >
> > We believe it is because of the more compact state abstractions learned by Adaptive Stacking ( as demonstrated in Figures 27-28). We discuss this idea of state abstraction extensively in Section 4.3, and hope the TMaze example on page 5 (moved from the appendix) helps make that discussion even clearer.
> >
> > # Q2
> >
> > We are happy to add these papers to our related work section.

---

### Official Review · Reviewer_kFSE · 2025-11-01

**Soundness:** 2
**Presentation:** 3
**Contribution:** 2
**Rating:** 2
**Confidence:** 4

**Summary:**

A new framework, Adaptive Stacking, is proposed, allowing selective retention of observations in a fixed-size working memory.

**Strengths:**

1. A new framework for working with non-Markovian environments, which is a very important area of research.
2. The proposed framework is theoretically well-founded.
3. The paper is well-written and easy to read.

**Weaknesses:**

1. The survey of related work overlooks important studies on recurrent models (FFM [1], SHM [2]) and transformer-based models addressing the limited context problem (HCAM [3], AGaLiTe [4]), including approaches that incorporate recurrence into transformers (RATE [5]). Additionally, work [6] investigates the impact of context length on successful memory-dependent task performance.
2. The experiments are conducted on only a small number of vector-based environments, and for TMaze, a very short corridor length is used. Testing on a larger variety of environments (both vector-based and visual) is necessary, such as MiniGrid-Memory, MemoryGym [7], or VizDoom-two-colors [5].
3. Results are missing on how the proposed framework performs in cases where using a frame stack is unnecessary, such as in MuJoCo [8] or similar environments. This is important to demonstrate that the proposed mechanism does not degrade performance in Markovian environments.
4. There is no comparison with strong baselines, such as FFM, SHM, RATE, and AGaLiTe.

I am willing to reconsider my evaluation if the mentioned shortcomings are addressed.

**References:**
1. Morad, Steven, et al. "Reinforcement learning with fast and forgetful memory." Advances in Neural Information Processing Systems 36 (2023): 72008-72029.
2. Le, Hung, et al. "Stable Hadamard Memory: Revitalizing Memory-Augmented Agents for Reinforcement Learning." arXiv preprint arXiv:2410.10132 (2024).
3. Lampinen, Andrew, et al. "Towards mental time travel: a hierarchical memory for reinforcement learning agents." Advances in Neural Information Processing Systems 34 (2021): 28182-28195.
4. Pramanik, Subhojeet, et al. "AGaLiTe: Approximate Gated Linear Transformers for Online Reinforcement Learning." arXiv preprint arXiv:2310.15719 (2023).
5. Cherepanov, Egor, et al. "Recurrent action transformer with memory." arXiv preprint arXiv:2306.09459 (2023).
6. Cherepanov, Egor, et al. "Unraveling the Complexity of Memory in RL Agents: an Approach for Classification and Evaluation." arXiv preprint arXiv:2412.06531 (2024).
7. Pleines, Marco, et al. "Memory Gym: Towards Endless Tasks to Benchmark Memory Capabilities of Agents." Journal of Machine Learning Research 26.6 (2025): 1-40.
8. Fu, Justin, et al. "D4rl: Datasets for deep data-driven reinforcement learning." arXiv preprint arXiv:2004.07219 (2020).

**Questions:**

1. How does the proposed approach perform compared to strong baselines?
2. Can it be adapted, for example, to FFM or SHM, and would this improve the original models’ performance?
3. Does the approach generalize to visual observations?
4. How does the approach perform in Markovian tasks?
5. How does the approach behave, or how does performance change, if the corridor length in TMaze is increased substantially, e.g., to 100 or 1000?

---

> ### Author Response · Authors · 2025-11-27
>
> We are grateful to the reviewer for the time and effort spent reviewing our paper. We believe we have identified the fundamental misunderstandings at the core of the reviewers' concerns (which required minimal revisions to our manuscript for clarity and additional results). Hence, with the following clarifications we hope we can gain their vote of confidence and actually improve their outlook on our paper -- particularly with respect to the significance of its contributions.
>
> ## W1,W4,Q1,Q2
> We believe this concern from the reviewer shows that we were not clear enough about the relationship between a memory stack and the agent in our paper. **We have updated Figure 1 to illustrate this better, and also include new results showing Adaptive Stacking generalising to a maze of length 1 million. To be precise, we see the memory stack as the context/sequence which the agent uses to build/update its state representation (e.g. using an LSTM, or a Transformer, or other architectures [1-5])**. This memory stack could be the full history (which is in our terminology equivalent to using Frame Stacking with $k=k\^\*$) as is done in most prior work [6], or a subset of observations from the history (e.g. using Frame Stacking with $k<k\^\*$).
>
> Hence the focus of this paper is not on specific architectures, but rather on how they update the sequence/context/stack with which they do the gradient updates for their policy.  To the best of our knowledge, most prior works irrespective of the architectures still propose using Frame Stacking with $k=k\^\*$ (e.g. using k = T = the maximum episode length). Hence the baselines in this work are not architectures nor RL algorithms, but Frame Stacking itself (i.e. using the same architectures and RL algorithms). We focused on PPO using MLPs, LSTMs, and Transformers since they are the most popular baselines in prior works (Ni et al., 2023) and we can train them within our compute budget. We hope these at least demonstrate the expected performance of swapping Frame Stacking with Adaptive Stacking in other architectures (e.g. FFM, SHM, etc).
>
> ## W2,W3,Q3,Q4
>
> Thank you to the reviewer for the suggestions. Please note that the suggested domains like VizDoom-two-colors and MemoryGym respectively require tens of millions ([9] Fig. 4.b) and hundreds of millions ([7] Figs. 6-9) of timesteps to train. However the compute budget we set for our experiments is 1 million training steps (even MiniGrid-Memory is not able to train sufficiently within this budget for the baselines).
>
> These domains require so many training steps because feature extraction from vision in RL is hard, even though their degree of non-Markovness ($k\^\*$) is low -- significantly lower than ours (with millions and even infinite $k\^\*$, see Table 2 page 21). This makes sense for these works since they focus on architectural improvements: given a sequence/context/stack, how to learn a “good” memory state representation. However we are focused on how that sequence/context/stack itself is updated, a question which becomes extremely important when $k\^\*$ gets impractically large (e.g. infinite). The key insight of our paper is that all prior works use Frame Stacking (a FIFO memory) to update the sequence/context/stack (which is principled and theoretically sound by definition), but leads to compute, memory, and generalisation tradeoffs for large $k\^\*$ (in addition to needing oracle knowledge of $k\^\*$).
>
> However, the reviewer’s suggestion inspired us to also add an experiment on the **Mujoco FetchReach task [8]**, modified to be partially observable (Figures 3.d and 6.b). Here the goal position changes every 50 steps and is only observable for the first 2 steps after each change. We also hide the joint velocities for all timesteps. **Our results demonstrate that the Adaptive Stacking agent achieves orders of magnitude more successes during training, and is even able to remember the goal position after 1 million steps in an episode during testing**. To the best of our knowledge this is the first work in RL demonstrating memorization over that order of magnitude, while the highest in prior works is only over **10\^3** orders of magnitude (Ni et al., 2023) (also see Table 2 page 21).
>
> Finally, if the domain is Markovian, then $k=k\^\*=\kappa=1$ making Frame Stacking and Adaptive Stacking equivalent and the same as the MDP (since the top of the stack is always the current observation $x\_t$). If the reviewer was instead wondering about what happens when we still choose to use $k>1$ in such a setting, we observed in the unmodified fully observable FetchReach that Frame Stacking and Adaptive Stacking get the same performance as when k=1. This is likely because Adaptive Stacking just learns to ignore the memory actions. We are happy to run this experiment over multiple seeds and include them in the camera ready.

---

> > ### Author Response · Authors · 2025-11-27
> >
> > # Q5
> >
> > Thank you to the reviewer for the suggestion. We have added maze lengths 100 and 1000 to our evaluations in Figure 27. We also evaluated on a maze length of 1 million in Figure 1. Our results show that Adaptive Stacking is still able to generalise to these much longer mazes.
> >
> > Finally, we would really appreciate it if the reviewers can critically evaluate our rebuttal, and look forward to engaging with them for further clarifications.
> >
> > [9] Sorokin, Artyom, et al. "Explain my surprise: Learning efficient long-term memory by predicting uncertain outcomes." Advances in Neural Information Processing Systems 35 (2022): 36875-36888.

---

### Author Response · Authors · 2025-11-27
**Clarified paper positioning and added scaling results for a TMaze and FetchRobot of length one Million per episode**

We are really grateful to the reviewers for their time and effort spent reviewing our paper. We believe we have identified the fundamental sources of the reviewers' concerns **(which required minimal revisions for clarity and additional results)**. With these clarifications and additional results, we hope we can gain the reviewers' vote of confidence and improve their outlook on our paper and the significance of its contributions.

We have made three main revisions (highlighted in red in the main paper) to address the concerns that were common across most reviewers:

# 1- Paper positioning:

We have updated **Figure 1 (page 1)** and the related works (page 4) to illustrate this better.
* It shows that the focus of this paper is not on specific architectures (e.g. LSTM, or a Transformer, or other architectures), but rather on how they update the sequence/context/stack with which they do the gradient updates for their policy.
* To the best of our knowledge, most prior works irrespective of the architectures they propose, still use Frame Stacking (full episode history as the context or a sliding context window).
* **Hence the baselines in this work are not architectures nor RL algorithms, but Frame Stacking itself.**

# 2- Theoretical results:

Some reviewers did not appreciate the purpose or significance of our theoretical results, which is a main contribution of this work.
* We have moved the counter example from the Appendix to the main paper (page 5).
* It shows that when the agent chooses to remove observations from its memory stack (its context/episode history), it makes the problem non-Markovian. This significantly changes the agent's value function (expectation of future rewards), and invalidates most current convergence and optimality guarantees for RL agents.
* Hence, why we go to great lengths to rigorously prove existence and convergence guarantees for our approach. As **Reviewer 8iuy** highlighted in their strengths, **“to the best of my knowledge, this is the first work to formally establish conditions under which such an approach can be guaranteed to work”.**

# 3- Experimental results

Some reviewers were unconvinced by the scale of our experiments and choice of baselines, and suggested more visually complex domains like VizDoom-two-colors and MemoryGym.
* Please note that the suggested domains require tens of millions ([1] Fig. 4.b) and hundreds of millions ([2] Figs. 6-9) of timesteps to train. However the compute budget we set for all of our experiments is 1 million training steps.
* These domains require so many training steps because feature extraction from vision in RL is hard, even though **their degree of non-Markovness ($k\^\*$) is low -- significantly lower than ours with millions and even infinite $k\^\*$, see Table 2 page 21**.
* This is why we frame this paper based on the “big world hypothesis”, where in our case “big world” means $k\ll k\^\*$.
* Hence, we include new results showing Adaptive Stacking generalising to a maze of length *1 million*, irrespective of architecture (Figure 1 page 1, Figures 24-28). Similarly, we also add an experiment on the **Mujoco FetchReach task [8]**, modified to require both short-term and long-term memory, and show that it can **still remember the goal position after 1 million steps in an episode**, irrespective of architecture (Figure 3.d page 8, Figure 6.b page 9, and Figure 29 page 35).

Hence, we would really appreciate it if the reviewers can critically evaluate our rebuttal, and look forward to engaging with them for further clarifications.


[1] Pleines, Marco, et al. "Memory Gym: Towards Endless Tasks to Benchmark Memory Capabilities of Agents." Journal of Machine Learning Research 26.6 (2025): 1-40.

[2] Sorokin, Artyom, et al. "Explain my surprise: Learning efficient long-term memory by predicting uncertain outcomes." Advances in Neural Information Processing Systems 35 (2022): 36875-36888.

---

> ### Author Response · Authors · 2025-12-02
>
> We understand that the Open Review data leak has compromised the review process and disadvantaged many authors. We feel particulary vulnerable because several of our reviews were indicative of truly fundamental misunderstandings, and in some cases, just uncareful reading of the paper. Our rebuttal outlines all of this and the considerable effort we devoted to additional experiments. It is quite unfortunate that we will not get to engage with the reviewers directly where we believe we would have cleared up misunderstandings and strongly rebutted the less favorable reviews.

---

### Meta-Review · Area_Chair_bcAH · 2026-01-07

**Summary:**

The paper proposes Adaptive Stacking (AS), a meta-algorithm designed to address non-Markovian environments where the history required for optimal decision-making ($k^*$) is large, but the subset of relevant observations ($\kappa$) is small. Unlike standard Frame Stacking (FS), which relies on a fixed-size FIFO buffer, AS learns a policy to selectively push and pop observations into a fixed-size memory stack. The authors provide theoretical convergence guarantees for AS under Monte Carlo and TD learning (contingent on a "Value-Consistency" assumption). They validate the method on T-Maze variants, XorMaze, Rubik's Cube, and a modified FetchReach task, demonstrating that AS can solve tasks requiring long-term memory (up to $10^6$ steps) with significantly smaller memory buffers than FS, and generalizes better to unseen horizons.

Synthesizing the reviews, the primary tension in this submission lies between its theoretical novelty and its empirical scope. While the theoretical analysis (establishing convergence for learned, discrete memory operations) is novel and rigorous, the empirical validation relies heavily on diagnostic, low-dimensional vector domains (T-Maze, Rubik's). I share the reviewers' concern that the exclusion of visually complex, high-dimensional benchmarks (e.g., MemoryGym, VizDoom), which are standard in modern memory-based RL literature, limits the assessment of the method's robustness in "real-world" scenarios. Furthermore, while the authors argue AS is architecture-agnostic, the lack of direct comparison against state-of-the-art memory-specific architectures (beyond vanilla LSTMs/Transformers) makes it difficult to benchmark its practical utility against existing specialized solutions. Finally, the "Value-Consistency" assumption required for TD convergence appears restrictive, though the empirical results on FetchReach (which violates this assumption) suggest some robustness.

I recommend Rejection, though this is a borderline decision. The theoretical contribution is distinct and the demonstration of generalization to 1-million-step horizons is impressive. However, the experimental evaluation feels disjointed from the complexity expected at ICLR. The refusal to evaluate on established, complex visual benchmarks (citing compute constraints) leaves a gap in proving the method's scalability beyond "toy" domains. While the paper handles the "Big $k^*$" problem well theoretically, the empirical evidence is not quite strong enough to secure an acceptance without reservations.

**Reviewer Concerns:**

### Concerns Addressed by Rebuttal:
* Reviewer DARz  raised concerns about the method description and the reproducibility. They noted the algorithm was buried in the appendix and code was missing. The authors updated Figure 1, moved the motivating example to the main text, and provided the code.
* Reviewer FrdC requested a comparison to similar "push/skip" memory methods (Demir, 2023). The authors added the XorMaze comparison, showing AS outperforms Demir’s method, which struggled with variance.
* Confusion regarding $k$ vs $k^\star$: Reviewer piza fundamentally misunderstood the paper's premise, arguing the theorems were trivial because they conflated the stack size $k$ with the environment's non-Markovian order $k^\star$. The authors' rebuttal successfully clarified that the contribution lies specifically in the setting where $k \ll k^\star$.
* Finally, reviewers kFSE and FrdC questioned scalability. The authors added a modified FetchReach experiment demonstrating the ability to retain goal information over 1 million steps, a significant improvement over standard baselines.

### Outstanding / Partially Unaddressed Concerns:
* Multiple reviewers (kFSE, FrdC, DARz) requested evaluations on visually complex environments like VizDoom or MemoryGym. The authors declined these requests citing compute constraints and a focus on memory management over feature extraction. This remains a significant weakness.
* Regarding missing SOTA baselines, reviewer kFSE requested comparisons to recent memory architectures (FFM, SHM, RATE). The authors argued AS is a wrapper for architectures and compared against FS using LSTMs/Transformers. While logically sound, the absence of direct comparison to state-of-the-art memory-specific architectures makes it difficult to contextualize the performance.

**Reviewer Scores:**

I estimated the scores based on how the rebuttal likely impacted the reviewer's original stance, adjusted to the allowed even numbers.

* **Reviewer kFSE. Original score: 2. Predicted score: 2**. While the rebuttal clarified the method's positioning, the reviewer's primary request for visual/complex benchmarks was not met.

* **Reviewer FrdC. Original score: 6. Predicted score: 6**. The reviewer was already positive. The addition of FetchReach and the Demir comparison likely solidifies their support.

* **Reviewer 8iuy. Original score: 4. Predicted score: 6**. This reviewer appreciated the theoretical "firsts" but remained skeptical of the hard-memory paradigm vs. soft memory. The FetchReach results might push them toward a 6, but a conservative estimate would be 4 given their philosophical objections.

* **Reviewer DARz. Original score: 4. Predicted score: 4**. The presentation/code issues were resolved, but the limited experimental diversity (lack of visual domains) likely prevents a full endorsement.

* **Reviewer piza. Original score: 2. Predicted score: 2**. This reviewer had a fundamental misunderstanding of the notation. While the rebuttal was decisive, reviewers with such fundamental misunderstandings rarely flip completely to acceptance. I have heavily discounted this score in my decision-making.

---

### Decision · Program_Chairs · 2026-01-26

Reject